# Open-vocabulary Multimodal Emotion Recognition: Dataset, Metric, and Benchmark

## Abstract

Multimodal Emotion Recognition (MER) is an important research topic. This paper advocates for a transformative paradigm in MER. The rationale behind our work is that current approaches often rely on a limited set of basic emotion labels, which do not adequately represent the rich spectrum of human emotions. These traditional and overly simplistic emotion categories fail to capture the inherent complexity and subtlety of human emotional experiences, leading to limited generalizability and practicality. Therefore, we propose a new MER paradigm called Open-vocabulary MER (OV-MER), which encompasses a broader range of emotion labels to reflect the richness of human emotions. This paradigm relaxes the label space, allowing for the prediction of arbitrary numbers and categories of emotions. To support this transition, we provide a comprehensive solution that includes a newly constructed database based on LLM and human collaborative annotations, along with corresponding metrics and a series of benchmarks. We hope this work advances emotion recognition from basic emotions to more nuanced emotions, contributing to the development of emotional AI.

## 1 Introduction

Research on emotions has a history spanning two centuries. As early as the 19th century, Charles Darwin conducted pioneering research about the evolutionary origins and possible purposes of emotions, explaining the emotional expressions of humans and animals (Darwin, 1872). In 1884, William James revealed the process of emotion generation, noting that stimuli trigger activities in the autonomic nervous system, which in turn produces an emotional experience in the brain (James, 1884). With the rapid development of AI, emotions have garnered increasing attention. Minsky (1988) highlighted the importance of emotions: *The question is not whether intelligent machines can have any emotions, but whether machines can be intelligent without any emotions*.

Humans convey emotions through various modalities, which gives rise to the task of Multimodal Emotion Recognition (MER) (Lian et al., 2024b). The basis of MER is how to model emotions. Currently, emotion models can be broadly classified into two categories: dimensional models and discrete models. The former uses multi-dimensional space to describe emotions, with the arousal-valence dimension being widely adopted; valence describes the pleasantness of a stimulus, while arousal indicates the intensity of emotion provoked by a stimulus (Warriner et al., 2013). However, these definitions require specialized psychological knowledge, making them abstract and difficult for the ordinary person to understand. This can lead to discrepancies between different annotators, posing challenges for subsequent applications.

Discrete models align more closely with human understanding of emotions. Ekman (1992) proposed the basic emotion theory, suggesting that there are six basic emotions: *anger*, *disgust*, *fear*, *happiness*, *sadness*, and *surprise*. This theory is widely used in MER, where researchers typically limit the label space to these basic emotions and use multiple annotators to select the most likely label through majority voting. We refer to this task as One-hot MER (OH-MER). Considering that emotions can be compound, researchers further propose Multi-label MER (ML-MER), allowing each sample to have multiple labels (Li et al., 2017). However, both OH-MER and ML-MER generally have limited label spaces and numbers. Nevertheless, Plutchik (2001) pointed out that humans can express approximately 34,000 different emotions. Therefore, restricting the label space and number will overlook some nuanced emotions.

(a) Task comparison        (b) Label comparison

Figure 1: Comparison. (a) **Task Comparison**: We compare the differences among three tasks (one-hot MER, multi-label MER, and open-vocabulary MER) across three aspects (label space, label number, and annotation manner); (b) **Label Comparison**: We provide an example to visualize the one-hot and OV labels. Since the original video contains real people, we use DemoAI to remove personal information to address copyright concerns. This paper uses emotion-related descriptions as a bridge to extract OV labels. We observe that OV labels contain richer emotions.

In this paper, we extend traditional MER to Open-vocabulary MER (OV-MER), where we allow the prediction of emotions across any number and categories. Figure 1(a) provides comparisons between different tasks. To facilitate further research, we build an initial dataset, define evaluation metrics, and develop solutions: (1) **For the dataset**, we propose a human-LLM collaboration strategy. Compared to human-only annotation, our strategy can leverage LLM to enhance the label richness (see Section 2); (2) **For the metrics**, since there is no fixed label space, the model may predict closely related but differently expressed emotions. For example, the ground truth is *joyful* and the model predicts *happy*. To provide more reliable evaluation results, we first group similar emotions and specifically design metrics for this task (see Section 3); (3) **For the solutions**, traditional classifiers rely on fixed label spaces. However, OV-MER does not restrict the label space or number of labels, necessitating the definition of new solutions (see Section 4).

A natural question arises: why is open vocabulary so important for MER? The reason is that OV can generate more nuanced emotions, leading to more accurate and reliable MER. As illustrated in Figure 1(b), labeling an emotion solely as *surprise* is not sufficiently informative—we cannot ascertain whether the *surprise* is positive or negative, which limits its practicality, such as in human-computer interaction (HCI) applications. In contrast, our OV-MER provides emotions like *surprise* along with additional descriptors such as *nervous* and *dissatisfied*, offering a more comprehensive and insightful understanding of the emotional state. Therefore, OV-MER facilitates the transition from basic to nuanced emotion recognition, advancing the development of emotion AI. Appendix A provides more detailed motivation. In summary, we make the following key contributions:

- **Paradigm**. We propose a new paradigm in MER called OV-MER. This paradigm transitions from traditional MER to a framework that enables the prediction of any number and category of emotions, thereby advancing emotion AI toward real-world applicability by capturing the full spectrum of human emotions.

- **Groundwork**. We lay the groundwork for OV-MER by constructing datasets, defining evaluation metrics, and proposing effective solutions. Our dataset enhances label richness through human-LLM collaboration. Meanwhile, we introduce new evaluation metrics that leverage emotional relevance to achieve more reliable results.

- **Benchmark**. We establish zero-shot benchmarks for OV-MER by conducting extensive experiments and providing detailed analysis. This task can serve as an important evaluation benchmark for multimodal LLMs (MLLMs), challenging their ability to integrate multimodal clues and capture subtle temporal variations in emotional expression.

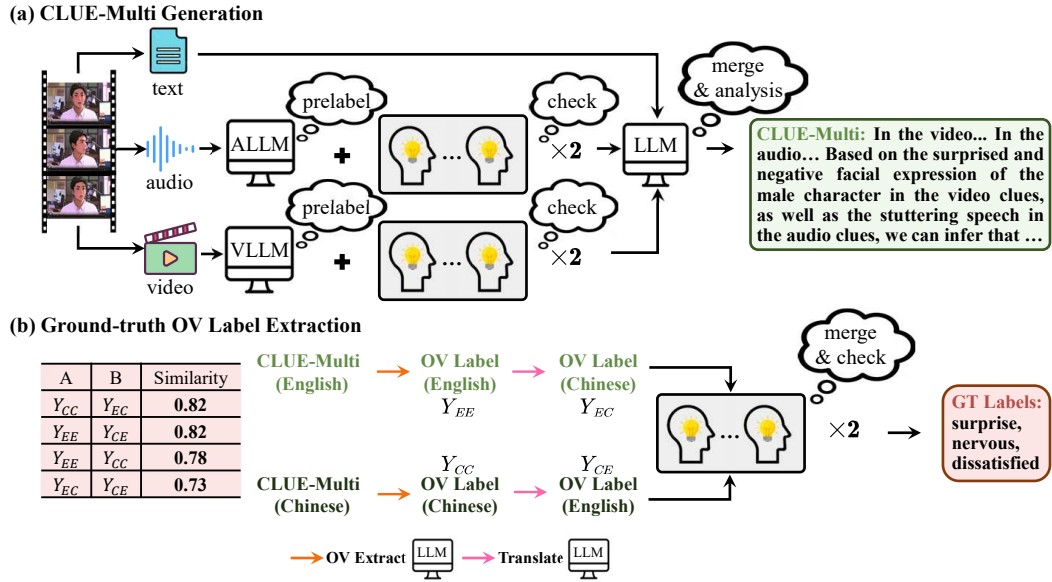

Figure 2: Dataset construction. (a) **CLUE-Multi Generation:** For audio and video, we use ALLM and VLLM to extract initial clues, followed by two rounds of manual checks to eliminate errors and duplicates while adding missing content. Each round involves multiple annotators, with no overlap between annotators in the two rounds. Finally, we merge the checked clues with text to generate CLUE-Multi. (b) **Ground-truth OV Label Extraction:** There are certain differences in the labels extracted from different languages. To eliminate language influence and achieve consensus labels, we merge these labels and conduct manual checks.

## 2    THE OV-MERD DATASET CONSTRUCTION

Although the concept of OV-MER is intuitive and holds great promise, its practical implementation faces significant challenges. The main difficulty lies in the broad and subtle range of human emotions, making comprehensive labeling a complex task. Traditional annotation methods are limited by their predefined emotion categories, which are often insufficient for the needs of OV-MER. In Figure 2, we propose a human-LLM collaboration strategy that consists of two steps: CLUE-Multi generation and emotion label extraction. Ultimately, we create a dataset called OV-MERD, which offers a richer set of emotions compared to existing datasets (see Table 1). This dataset is an extension of MER2023 (Lian et al., 2023). Specifically, MER2023 is collected from movies and TV series, and we randomly select a portion of data for further annotation to build our OV-MERD.

### 2.1    CLUE-MULTI GENERATION

During the annotation process, we observe that descriptions generated through human-LLM collaboration are more detailed. This indicates that when annotations are solely performed by humans, they tend to focus on major clues while neglecting minor ones (see Section 5 for more analysis). In this section, we provide a detailed overview of our human-LLM collaboration strategy.

**Pre-annotation.**    Initially, we attempt to annotate visual and acoustic clues directly. However, the descriptions obtained in this way cannot cover all information. Therefore, we explore using other models for pre-annotation. (1) For video, given the strong visual understanding capabilities of GPT-4V ("gpt-4-vision-preview"), we use it as video LLM (VLLM) for pre-annotation. Since GPT-4V only supports image input, we uniformly sample three frames from each video and input them into GPT-4V. We discuss the reasons for sampling three frames in Appendix I. (2) For audio, we use the open-source SALMONN (Tang et al., 2023) as audio LLM (ALLM) for pre-annotation, as GPT-4V does not support audio input.

Table 1: Dataset comparison. See Appendix G for a comprehensive comparison.

| Dataset | Modality | Annotation Type | # Categories | # Labels per Sample |
|---|---|---|---|---|
| MOUD (Pérez-Rosas et al., 2013) | A,V,T | Dimensional Emotion | 1 | 1 |
| CMU-MOSI (Zadeh et al., 2017) | A,V,T | Dimensional Emotion | 1 | 1 |
| CH-SIMS (Yu et al., 2020) | A,V,T | Dimensional Emotion | 1 | 1 |
| CH-SIMS v2 (Liu et al., 2022a) | A,V,T | Dimensional Emotion | 1 | 1 |
| SEMAINE (McKeown et al., 2011) | A,V,T | Dimensional Emotion | 5 | 1 |
| MSP-IMPROV (Busso et al., 2016) | A,V,T | Discrete Emotion | 4 | 1 |
| IEMOCAP (Busso et al., 2008) | A,V,T | Discrete Emotion | 10 | 1 |
| MELD (Poria et al., 2019) | A,V,T | Discrete Emotion | 7 | 1 |
| MER2023 (Lian et al., 2023) | A,V,T | Discrete Emotion | 6 | 1 |
| MER2024 (Lian et al., 2024a) | A,V,T | Discrete Emotion | 6 | 1 |
| **OV-MERD (Ours)** | **A,V,T** | **Discrete Emotion** | **248** (arbitrary label) | **1∼9, most 2∼4** (arbitrary number) |

**Manual Check.** As part of our quality assurance procedures, we perform a detailed examination of the pre-annotated results. For visual clues, GPT-4V may generate hallucinated responses, i.e., clues that do not actually exist. Additionally, there are repeated expressions and some temporal association clues are missing. Therefore, we hire annotators to eliminate errors and duplicates, as well as add missing content. For acoustic clues, ALLM struggles to capture emotion-related paralinguistic features. The main reason is that current ALLM primarily focuses on tasks like ASR or audio event detection (Tang et al., 2023), with less emphasis on paralinguistic information. Hence, we hire multiple annotators to focus on the speaker's intonation and other emotion-related paralinguistic clues. To reduce subjective bias, we conduct two rounds of manual checks, each involving different annotators. These annotators are experts in affective computing and are familiar with the definitions of emotions. Ultimately, these checked clues can accurately reflect the video content. Appendix J provides the annotation guideline and layout of the annotation platform.

**CLUE-Multi Generation.** Subsequently, we leverage the reasoning capabilities of LLM to merge all clues. Specifically, we use GPT-3.5 ("gpt-3.5-turbo-16k-0613") as the LLM and ask it to merge textual, acoustic, and visual clues to infer emotional states. The output of this process is an emotion-related description, denoted as *CLUE-Multi* (see Figure 2). In Appendix E, we further discuss the details of this merging process and the reasons behind it. Overall, the above annotation pipeline reflects the collaboration between humans and LLMs. In Section 5, we compare human-only annotation with this strategy, observing that the latter can generate richer descriptions.

## 2.2 Ground-truth OV Label Extraction

**Label Extraction.** After that, we use the LLM to extract emotion labels from *CLUE-Multi*. This process relies on GPT-3.5, which we request to identify emotional states based on the provided descriptions without restricting the label space. See Appendix E for more details.

**Language Impact.** We further explore the language impact. In Figure 2, we first extract OV labels from English and Chinese descriptions, obtaining $Y_{EE}$ and $Y_{CC}$. Then, we translate them into the other language, yielding $Y_{EC}$ and $Y_{CE}$. After that, we measure the similarity between different sets and report results in Figure 2. In Appendix K, we detail our experimental design and similarity metric. We observe that the labels extracted from different languages exhibit some differences. For example, the similarity score between $Y_{EE}$ and $Y_{CE}$ is 0.82, which may be due to the varying definitions of emotions in different languages. To eliminate language influence and achieve consensus labels, we merge the labels extracted from both languages and conduct manual checks. These checked labels are regarded as the ground truth.

## 2.3 OV-MERD Dataset

Finally, we construct a dataset called OV-MERD. This dataset is an extension of MER2023 (Lian et al., 2023), from which we randomly select a portion of samples for further annotation. Table 1 compares OV-MERD with existing datasets. We observe that our OV-MERD dataset contains 248 emotion categories and most samples have 2 to 4 labels, far exceeding those in current datasets. In Appendix F, we observe that OV-MERD encompasses a broader range of emotions, including some that have been rarely discussed in previous research, such as *shy*, *nervous*, and *grateful*.

## 3 EVALUATION METRIC

Defining evaluation metrics for OV-MER presents significant challenges: (1) **OV-MER supports predicting emotions of any category.** Thus, the model may predict closely related but differently expressed emotions. To provide more reliable evaluation results, we first group the emotions based on their similarities. (2) **OV-MER allows for the prediction of an arbitrary number of labels.** Thus, traditional evaluation metrics designed for a fixed number of labels may not be applicable. In this section, we propose set-based evaluation metrics specifically tailored for this task.

### 3.1 GROUPING

We propose two grouping strategies: one based on GPT and the other based on the emotion wheel (EW) Plutchik (1980). In the following experiments, we default to using GPT-based grouping.

**GPT-based Grouping.** The most direct approach is to use GPT-3.5 to group all labels based on their similarity: *Please assume the role of an expert in the field of emotions. We provide a set of emotions. Please group the emotions, with each group containing emotions with the same meaning. Directly output the results. The output format should be a list containing multiple lists.* However, the evaluation results may be affected by the API version. For example, if OpenAI deprecates an old API, the results based on that API will become difficult to reproduce. Additionally, this process is costly (see Appendix P). Therefore, we attempt to find a replacement for GPT-based grouping.

**EW-based Grouping.** EW is a psychological model that categorizes emotions in a structured manner. The inner part shows core emotions while moving to the outer part reveals more nuanced emotions. Therefore, EW naturally provides emotion grouping information. Since there is no consensus on EW, we select five representative wheels W1~W5. See Appendix R for details.

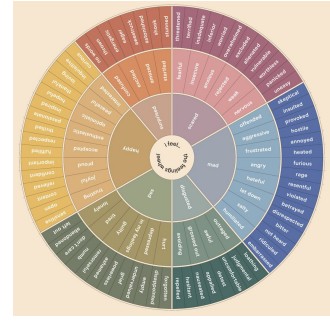

Take W1 as an example (see Figure 3). Before calculating the metrics, we define some symbols. We group the labels by their levels from the innermost to the outermost as $L_{w_1}^1$, $L_{w_1}^2$, and $L_{w_1}^3$. Specifically, $L_{w_1}^1 = \{mad, \cdots, scared\}$. Next, we define a function $m_{w_1}^{i \rightarrow j}(\cdot)$ that maps the labels in $L_{w_1}^i$ to the corresponding labels in $L_{w_1}^j$. From inner to outer ($i < j$), $m_{w_1}^{i \rightarrow j}(\cdot)$ is a many-to-one mapping; from outer to inner ($i > j$), $m_{w_1}^{i \rightarrow j}(\cdot)$ is a one-to-many mapping. For example, $m_{w_1}^{2 \rightarrow 1}(angry) = \{mad\}$ and

Figure 3: Emotion wheel W1.

$m_{w_1}^{1 \rightarrow 2}(mad) = \{offended, \cdots, humiliated\}$. We collect all the labels from these emotion wheels and represent them as *EW*, i.e., $\{L_{w_i}^j, 1 \leq i \leq 5, 1 \leq j \leq 3\}$. We denote the labels in *EW* as $y_w$.

Considering that the emotional categories in *EW* are still limited, we perform some label expansion operations. Specifically, we repeatedly call GPT-3.5, asking it to generate synonyms for each label. The prompt used is as follows: *Please retrieve the synonyms for the following words and output them in a table format.* Then, we generate *EW-S*, i.e., $\{f(y_w) = \{y_f^1, ..., y_f^n\}, y_w \in \text{EW}\}$, where $f(\cdot)$ is a function that maps each label $y_w$ to its synonym $y_f$. We also define its inverse function $f'(\cdot)$, which maps different synonyms $y_f$ back to their base label $y_w$.

To eliminate the influence of word forms (e.g., *happy* and *happiness*), we further ask GPT-3.5 multiple times to generate different forms for each label. The prompt used is as follows: *Please output different forms of the following word in a list format.* After that, we obtain *EW-SF*, i.e., $\{g(y_f) = \{y_g^1, ..., y_g^m\}, y_f \in \text{EW-S}\}$, where $g(\cdot)$ is a function that maps each label $y_f$ to its different forms $y_g$. We also define its inverse function $g'(\cdot)$, which maps different labels $y_g$ back to their base form $y_f$. Finally, we define different types of metrics:

(1) **M1**. We use $g'(\cdot)$ to map all labels to their corresponding $y_f$.

(2) **M2**. We use $f'(g'(\cdot))$ to map all labels to their corresponding $y_w$.

(3) **M3**. We use the emotion wheel during metric calculation. Specifically, we first use $f'(g'(\cdot))$ to map all labels to their corresponding $y_w$. Then, we define two grouping functions, L1 and L2. For

L1, we map all labels to their corresponding $L_{w_i}^1$:

$$
\begin{cases}
y_w, & \text{if } y_w \in L_{w_i}^1 \\
m_{w_i}^{2\to1}(y_w), & \text{if } y_w \in L_{w_i}^2 \\
m_{w_i}^{2\to1}(m_{w_i}^{3\to2}(y_w)), & \text{if } y_w \in L_{w_i}^3
\end{cases}
\tag{1}
$$

For L2, we map all labels to their corresponding $L_{w_i}^2$:

$$
\begin{cases}
\text{select one label in } m_{w_i}^{1\to2}(y_w), & \text{if } y_w \in L_{w_i}^1 \\
y_w, & \text{if } y_w \in L_{w_i}^2 \\
m_{w_i}^{3\to2}(y_w), & \text{if } y_w \in L_{w_i}^3
\end{cases}
\tag{2}
$$

## 3.2 METRIC DEFINITION

Then, we convert the above emotion grouping information into a function $G(\cdot)$, which can map each label to its group ID. Specifically, suppose $\{y_i\}_{i=1}^M$ and $\{\hat{y}_i\}_{i=1}^N$ are the ground truth and predictions, where $M$ and $N$ are the number of labels. We first map each label into its group ID:

$$
\mathcal{Y} = \{G(x) | x \in \{y_i\}_{i=1}^M\}, \hat{\mathcal{Y}} = \{G(x) | x \in \{\hat{y}_i\}_{i=1}^N\}.
\tag{3}
$$

Finally, we design set-based metrics for performance evaluation. Specifically, Precision$_\text{S}$ indicates the number of correctly predicted labels; Recall$_\text{S}$ indicates whether the prediction covers all ground truth; and F$_\text{S}$ is the harmonic mean of two metrics, which is used for the final ranking. It is important to note that changing the label order in $\mathcal{Y}$ and $\hat{\mathcal{Y}}$ does not result in any change in performance.

$$
\text{Precision}_\text{S} = \frac{|\mathcal{Y} \cap \hat{\mathcal{Y}}|}{|\hat{\mathcal{Y}}|}, \ \text{Recall}_\text{S} = \frac{|\mathcal{Y} \cap \hat{\mathcal{Y}}|}{|\mathcal{Y}|}, \ \text{F}_\text{S} = 2 \times \frac{\text{Precision}_\text{S} \times \text{Recall}_\text{S}}{\text{Precision}_\text{S} + \text{Recall}_\text{S}}.
\tag{4}
$$

## 4 BASELINES FOR OV-MER

### 4.1 CLUE GENERATION

Figure 2 illustrates the generation process of CLUE-Multi, where we combine text with manually checked visual and acoustic clues. In this section, we further introduce the following variants.

**CLUE-A/T/V.** To reveal the modality impact, we propose three variants of CLUE-Multi: *CLUE-Audio*, *CLUE-Text*, and *CLUE-Video*. Their generation process is illustrated in Figure 4. (1) *CLUE-Audio*: We observe that ALLM cannot fully leverage the text, and using an additional LLM to emphasize the text can further improve performance, which is also verified in Section 5. Therefore, we merge the checked acoustic clues with text using an additional LLM; (2) *CLUE-Text*: We only use the text to infer emotional states; (3) *CLUE-Video*: Since the visual content does not contain audio and text, we only use the checked visual clues. See Appendix M for more examples.

**CLUE-MLLM.** MLLMs can address various multimodal tasks. Since emotion recognition relies on temporal information, we choose models that support at least video or audio. To generate *CLUE-MLLM*, we first use ALLM or VLLM to extract emotion-related descriptions, and then combine these descriptions with text using LLM. Compared with CLUE-Multi, this process does not use manually checked clues. Appendix N provides model cards and relevant prompts. This paper aims to build a zero-shot benchmark for OV-MER, without the training process. All models are implemented in PyTorch, and all inference processes are carried out using a 32G NVIDIA Tesla V100 GPU.

### 4.2 METRIC CALCULATION

As shown in Figure 2, there are certain differences in the labels extracted from different languages. Therefore, we report the results for both English and Chinese descriptions. In Figure 4, for the Chinese branch, we first extract OV labels and then translate them into English; for the English branch, we directly extract OV labels. Finally, we compute the evaluation metrics with the ground truth. It is worth noting that the OV labels extracted from the monolingual CLUE-Multi differ from the ground truth. Our ground truth combines the labels extracted from different languages and undergoes further manual checks (see Figure 2).

Figure 4: Baselines. (a) **Preliminary:** We begin by defining some preliminary symbols. (b) **CLUE Generation:** CLUE-Video and CLUE-Audio use manually-checked clues; CLUE-Text relies solely on text; CLUE-MLLM does not involve manual checks and directly uses the outputs from ALLM or VLLM. (c) **Metric Calculation:** We rely on CLUE to predict emotion labels. Due to variations in labels extracted from different languages, we report results across different languages.

Table 2: Baseline results. Figure 4 shows the metric calculation process.

| Model | L | V | A | English | | | Chinese | | |
|---|---|---|---|---|---|---|---|---|---|
| | | | | $F_S \uparrow$ | $Precision_S \uparrow$ | $Recall_S \uparrow$ | $F_S \uparrow$ | $Precision_S \uparrow$ | $Recall_S \uparrow$ |
| Heuristic Baseline | | | | | | | | | |
| Random | ✕ | ✕ | ✕ | $17.42_{\pm 0.01}$ | $24.85_{\pm 0.15}$ | $13.42_{\pm 0.04}$ | $16.59_{\pm 0.00}$ | $24.70_{\pm 0.00}$ | $12.48_{\pm 0.00}$ |
| CLUE-MLLM | | | | | | | | | |
| Qwen-Audio | √ | ✕ | √ | $38.13_{\pm 0.05}$ | $49.42_{\pm 0.18}$ | $31.04_{\pm 0.00}$ | $41.14_{\pm 0.07}$ | $53.71_{\pm 0.00}$ | $33.34_{\pm 0.09}$ |
| OneLLM | √ | ✕ | √ | $42.84_{\pm 0.06}$ | $45.92_{\pm 0.05}$ | $40.15_{\pm 0.06}$ | $46.17_{\pm 0.02}$ | $52.07_{\pm 0.06}$ | $41.47_{\pm 0.08}$ |
| Otter | √ | √ | ✕ | $43.51_{\pm 0.09}$ | $50.71_{\pm 0.10}$ | $38.09_{\pm 0.09}$ | $46.22_{\pm 0.01}$ | $52.65_{\pm 0.16}$ | $41.18_{\pm 0.08}$ |
| Video-LLaMA | √ | √ | ✕ | $44.73_{\pm 0.14}$ | $44.14_{\pm 0.13}$ | $45.34_{\pm 0.15}$ | $47.26_{\pm 0.03}$ | $47.98_{\pm 0.07}$ | $46.56_{\pm 0.01}$ |
| VideoChat | √ | √ | ✕ | $45.53_{\pm 0.11}$ | $42.90_{\pm 0.27}$ | $48.49_{\pm 0.10}$ | $45.57_{\pm 0.03}$ | $47.20_{\pm 0.12}$ | $44.05_{\pm 0.05}$ |
| SECap | √ | ✕ | √ | $45.72_{\pm 0.09}$ | $54.52_{\pm 0.15}$ | $39.37_{\pm 0.05}$ | $45.57_{\pm 0.13}$ | $55.55_{\pm 0.23}$ | $38.64_{\pm 0.08}$ |
| PandaGPT | √ | √ | √ | $45.89_{\pm 0.20}$ | $50.03_{\pm 0.01}$ | $42.38_{\pm 0.33}$ | $47.33_{\pm 0.04}$ | $53.01_{\pm 0.08}$ | $42.75_{\pm 0.11}$ |
| Video-LLaVA | √ | √ | ✕ | $47.07_{\pm 0.16}$ | $48.58_{\pm 0.02}$ | $45.66_{\pm 0.29}$ | $49.21_{\pm 0.06}$ | $53.95_{\pm 0.03}$ | $45.23_{\pm 0.13}$ |
| SALMONN | √ | ✕ | √ | $47.96_{\pm 0.04}$ | $50.20_{\pm 0.04}$ | $45.92_{\pm 0.04}$ | $48.24_{\pm 0.03}$ | $52.24_{\pm 0.00}$ | $44.82_{\pm 0.05}$ |
| VideoChat2 | √ | √ | ✕ | $49.07_{\pm 0.26}$ | $54.72_{\pm 0.41}$ | $44.47_{\pm 0.15}$ | $48.86_{\pm 0.05}$ | $57.12_{\pm 0.08}$ | $42.68_{\pm 0.04}$ |
| Video-ChatGPT | √ | √ | ✕ | $50.52_{\pm 0.06}$ | $54.03_{\pm 0.04}$ | $47.44_{\pm 0.07}$ | $54.73_{\pm 0.00}$ | $\mathbf{61.15}_{\pm 0.10}$ | $49.52_{\pm 0.06}$ |
| OneLLM | √ | √ | ✕ | $50.52_{\pm 0.07}$ | $\mathbf{55.93}_{\pm 0.09}$ | $46.06_{\pm 0.06}$ | $51.44_{\pm 0.08}$ | $56.43_{\pm 0.04}$ | $47.26_{\pm 0.11}$ |
| LLaMA-VID | √ | √ | ✕ | $51.25_{\pm 0.09}$ | $52.71_{\pm 0.16}$ | $49.87_{\pm 0.06}$ | $52.01_{\pm 0.02}$ | $57.30_{\pm 0.00}$ | $47.61_{\pm 0.03}$ |
| mPLUG-Owl | √ | √ | ✕ | $52.73_{\pm 0.13}$ | $54.54_{\pm 0.13}$ | $51.04_{\pm 0.13}$ | $50.95_{\pm 0.06}$ | $56.40_{\pm 0.11}$ | $46.47_{\pm 0.18}$ |
| Chat-UniVi | √ | √ | ✕ | $53.08_{\pm 0.01}$ | $53.68_{\pm 0.00}$ | $52.50_{\pm 0.02}$ | $53.86_{\pm 0.02}$ | $58.54_{\pm 0.01}$ | $49.86_{\pm 0.03}$ |
| GPT-4V | √ | √ | ✕ | $\mathbf{55.51}_{\pm 0.05}$ | $48.52_{\pm 0.07}$ | $\mathbf{64.86}_{\pm 0.00}$ | $\mathbf{57.21}_{\pm 0.01}$ | $54.61_{\pm 0.02}$ | $\mathbf{60.07}_{\pm 0.01}$ |
| CLUE-M/A/T/V | | | | | | | | | |
| CLUE-Text | √ | ✕ | ✕ | $46.00_{\pm 0.06}$ | $54.41_{\pm 0.15}$ | $39.84_{\pm 0.01}$ | $43.11_{\pm 0.25}$ | $50.69_{\pm 0.26}$ | $37.50_{\pm 0.23}$ |
| CLUE-Video | ✕ | √ | ✕ | $60.55_{\pm 0.13}$ | $63.29_{\pm 0.08}$ | $58.05_{\pm 0.16}$ | $61.73_{\pm 0.10}$ | $66.47_{\pm 0.13}$ | $57.62_{\pm 0.08}$ |
| CLUE-Audio | √ | ✕ | √ | $65.35_{\pm 0.04}$ | $67.54_{\pm 0.08}$ | $63.30_{\pm 0.00}$ | $68.56_{\pm 0.07}$ | $70.10_{\pm 0.06}$ | $67.07_{\pm 0.08}$ |
| CLUE-Multi | √ | √ | √ | $\mathbf{80.05}_{\pm 0.24}$ | $\mathbf{80.03}_{\pm 0.37}$ | $\mathbf{80.07}_{\pm 0.10}$ | $\mathbf{85.16}_{\pm 0.03}$ | $\mathbf{87.09}_{\pm 0.00}$ | $\mathbf{83.31}_{\pm 0.05}$ |

## 5 RESULTS AND DISCUSSION

In this section, we default to using GPT-based grouping and employ GPT-3.5 ("gpt-3.5-turbo-16k-0613") as LLM. We generally report evaluation results in both languages, but if no specific language is mentioned, we default to reporting results for the English branch. To mitigate the impact of randomness, we conduct each experiment twice and report the average scores and standard deviations.

**Main Results on CLUE-M/A/T/V.** For CLUE-M/A/T/V, most baselines use manually checked clues, which serve as performance upper bounds of different modality combinations. In Table 2, we observe that CLUE-Multi performs the best, highlighting the importance of multimodal information in emotion recognition. In contrast, CLUE-Text performs the worst. This is because our OV-MERD dataset is derived from MER2023, where the contribution of text is smaller compared to audio and video (Lian et al., 2024b). Relying solely on text makes it difficult to recognize emotions accurately.

Table 3: Performance of combinations of different MLLMs.

| Model | English | | | Chinese | | |
|---|---|---|---|---|---|---|
| | $F_S$ ↑ | $\text{Precision}_S$ ↑ | $\text{Recall}_S$ ↑ | $F_S$ ↑ | $\text{Precision}_S$ ↑ | $\text{Recall}_S$ ↑ |
| Audio + Text | | | | | | |
| SECap | $45.72_{\pm0.09}$ | $54.52_{\pm0.15}$ | $39.37_{\pm0.05}$ | $45.57_{\pm0.13}$ | $55.55_{\pm0.23}$ | $38.64_{\pm0.08}$ |
| SALMONN | $47.96_{\pm0.04}$ | $50.20_{\pm0.04}$ | $45.92_{\pm0.04}$ | $48.24_{\pm0.03}$ | $52.24_{\pm0.00}$ | $44.82_{\pm0.05}$ |
| Video + Text | | | | | | |
| Video-ChatGPT | $50.52_{\pm0.06}$ | $54.03_{\pm0.04}$ | $47.44_{\pm0.07}$ | $54.73_{\pm0.00}$ | $61.15_{\pm0.10}$ | $49.52_{\pm0.06}$ |
| mPLUG-Owl | $52.73_{\pm0.13}$ | $54.54_{\pm0.13}$ | $51.04_{\pm0.13}$ | $50.95_{\pm0.06}$ | $56.40_{\pm0.11}$ | $46.47_{\pm0.18}$ |
| Chat-UniVi | $53.08_{\pm0.01}$ | $53.68_{\pm0.00}$ | $52.50_{\pm0.02}$ | $53.86_{\pm0.02}$ | $58.54_{\pm0.01}$ | $49.86_{\pm0.03}$ |
| Audio + Video + Text | | | | | | |
| SECap + mPLUG-Owl | $56.69_{\pm0.03}$ | $50.05_{\pm0.23}$ | $65.38_{\pm0.33}$ | $54.99_{\pm0.23}$ | $51.65_{\pm0.27}$ | $58.79_{\pm0.16}$ |
| SECap + Video-ChatGPT | $56.90_{\pm0.08}$ | $52.03_{\pm0.04}$ | $62.79_{\pm0.14}$ | $56.49_{\pm0.02}$ | $\mathbf{56.50}_{\pm0.01}$ | $56.48_{\pm0.05}$ |
| SECap + Chat-UniVi | $57.34_{\pm0.16}$ | $48.85_{\pm0.29}$ | $\mathbf{69.41}_{\pm0.13}$ | $56.19_{\pm0.13}$ | $52.38_{\pm0.07}$ | $60.59_{\pm0.22}$ |
| SALMONN + Video-ChatGPT | $58.19_{\pm0.23}$ | $\mathbf{53.16}_{\pm0.17}$ | $64.26_{\pm0.31}$ | $55.05_{\pm0.16}$ | $53.44_{\pm0.14}$ | $56.76_{\pm0.19}$ |
| SALMONN + Chat-UniVi | $58.43_{\pm0.06}$ | $51.62_{\pm0.00}$ | $67.31_{\pm0.15}$ | $\mathbf{56.93}_{\pm0.06}$ | $51.65_{\pm0.06}$ | $\mathbf{63.42}_{\pm0.06}$ |
| SALMONN + mPLUG-Owl | $\mathbf{58.70}_{\pm0.04}$ | $51.77_{\pm0.01}$ | $67.76_{\pm0.11}$ | $55.62_{\pm0.21}$ | $51.74_{\pm0.19}$ | $60.14_{\pm0.23}$ |

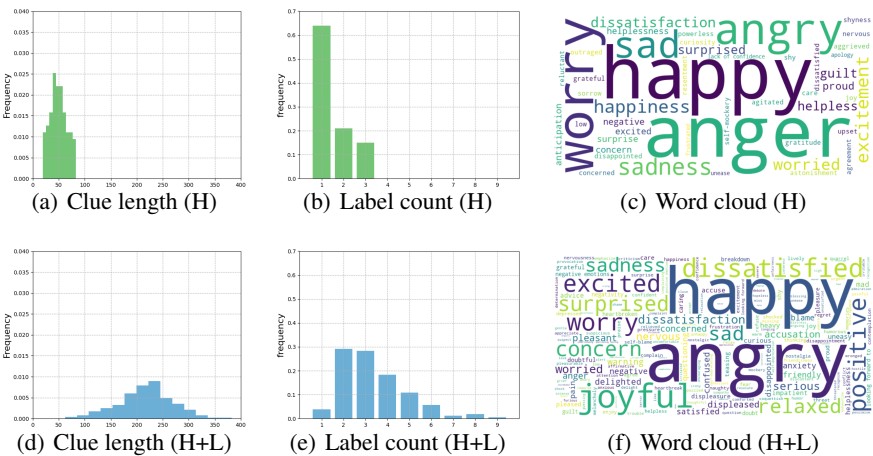

(a) Clue length (H)  (b) Label count (H)  (c) Word cloud (H)

(d) Clue length (H+L)  (e) Label count (H+L)  (f) Word cloud (H+L)

Figure 5: Human-only annotation (H) vs. Human-LLM (H+L) collaboration.

**Main Results on CLUE-MLLM.** Table 2 presents the results of CLUE-MLLM. Additionally, we introduce a heuristic baseline called *Random*, where we randomly select a label from basic emotions. This baseline reflects the lower bound of performance. We observe that MLLM generally outperforms *Random*, indicating that MLLM can partially address OV-MER. However, the performance of MLLM remains unsatisfactory, highlighting the limitations of existing MLLMs and the challenges of OV-MER. Furthermore, models that perform well in Chinese often perform well in English. These results suggest that the impact of language differences on rankings is limited.

**Effectiveness of Multimodal Fusion.** In Table 3, we select the best-performing ALLMs and VLLMs and explore whether their combinations can lead to better performance. To fuse different modalities, we input prelabeled acoustic and visual clues into LLM and leverage its reasoning capabilities for multimodal integration (see Figure 4). This approach is consistent with the fusion method used during our dataset construction process (see Section 2). We observe that multimodal results generally outperform ALLM-only or VLLM-only results. The reason lies in that emotions are conveyed through various modalities. By integrating different modalities, we can obtain a more comprehensive understanding of emotions, leading to better performance in OV-MER.

**Human-only vs. Human-LLM Collaboration.** To verify the effectiveness of our human-LLM collaborative strategy, we additionally introduce a baseline using human-only annotation. In Figure 5, we compare two strategies from three aspects: the length distribution of generated clues, the distribution of label counts, and the word cloud. We observe that through human-LLM collaboration, we can obtain longer descriptions, generate a broader range of emotions, and provide more diverse labels for each sample. These results show that human-only annotation generally focuses on primary emotions while neglecting minor ones. With the pre-annotation and semantic reasoning capabilities

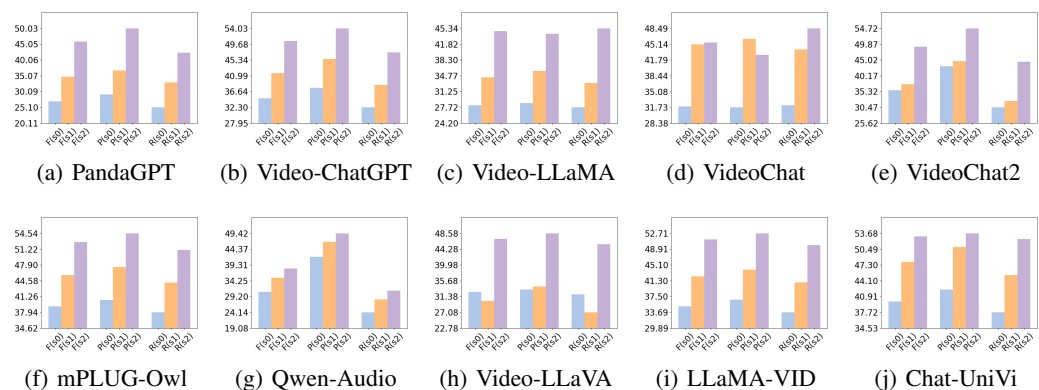

Figure 6: Performance comparison of different strategies for generating CLUE-MLLM.

of LLMs, we can obtain richer emotional labels. These results validate the effectiveness of our human-LLM collaborative strategy. Meanwhile, these results suggest that the LLM-driven approach does not lead to a narrow or biased interpretation of emotions, but rather helps uncover more subtle emotional nuances. Additional experiments are provided in the Appendix O.

**Ablation Study on CLUE-MLLM.** In this section, we reveal the impact of different CLUE-MLLM generation strategies. Figure 7 introduces three methods: 1) **S0** does not use text and inputs the video into MLLM; 2) **S1** inputs both text and video into MLLM; 3) **S2** first uses MLLM to extract descriptions and then combines with text using another LLM, same with the strategy in Figure 4. In Figure 6, S1 and S2 generally outperform S0, indicating the importance of the text content in OV-MER. Moreover, S2 typically performs better than S1. The reason is that inputting video and text into the MLLM simultaneously increases the task difficulty, and current MLLMs may struggle to handle complex prompts. S2 divides this process into two steps, reducing task complexity and achieving better performance. Therefore, we adopt S2 as the default strategy. More results are provided in Appendix N.

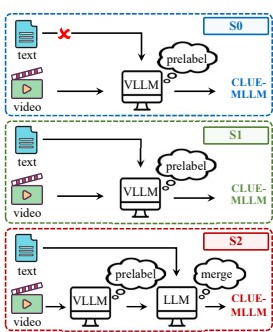

Figure 7: Ablation study.

**GPT-based vs. Matching-based Metrics.** In Table 2, CLUE-Multi performs the best. This leads to a hypothesis: Do sentences that are more "similar" to CLUE-Multi yield better emotion recognition performance? The most common way to measure "similarity" is through matching-based metrics, with $BLEU_1$, $BLEU_4$, METEOR, and $ROUGE_1$ being the most widely used. Therefore, we use CLUE-MLLM as input and calculate both GPT-based and matching-based metrics (see Figure 8(a)), followed by calculating their PCC scores (see Figure 8(b)). From the experimental results, we have some interesting observations. First, the same metric across different languages typically exhibits high correlations. However, the correlation between GPT-based and

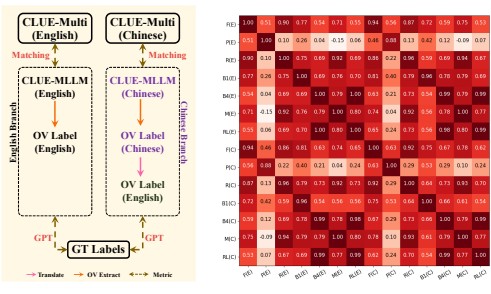

(a) Metric Calculation     (b) Correlation Analysis

Figure 8: GPT- vs. Matching-based metrics.

matching-based metrics is not strong. For example, the highest PCC score between "F(E)" and matching-based metrics is 0.77. These results highlight the limitations of using matching-based metrics to evaluate the OV-MER task. The main reason is that matching-based metrics focus on low-level word-level matches, while emotion understanding is a relatively complex and high-level perceptual task. In Appendix Q, we provide more examples for clarification.

Table 4: GPT-based vs. EW-based grouping. We calculate the PCC score to reveal their correlation.

| Model | GPT | M1 | M2 | M3-W1 | | M3-W2 | |
|---|---|---|---|---|---|---|---|
| | | | | L1 | L2 | L1 | L2 |
| Qwen-Audio | $38.13_{\pm 0.05}$ | $20.49_{\pm 0.01}$ | $23.37_{\pm 0.01}$ | $43.85_{\pm 0.03}$ | $26.60_{\pm 0.01}$ | $41.52_{\pm 0.28}$ | $26.68_{\pm 0.01}$ |
| Otter | $43.51_{\pm 0.09}$ | $22.21_{\pm 0.06}$ | $27.99_{\pm 0.02}$ | $49.75_{\pm 0.11}$ | $33.50_{\pm 0.06}$ | $49.93_{\pm 0.11}$ | $33.04_{\pm 0.06}$ |
| Video-LLaMA | $44.73_{\pm 0.14}$ | $23.56_{\pm 0.08}$ | $28.39_{\pm 0.17}$ | $52.90_{\pm 0.12}$ | $36.08_{\pm 0.13}$ | $53.60_{\pm 0.04}$ | $35.33_{\pm 0.08}$ |
| VideoChat | $45.53_{\pm 0.11}$ | $22.15_{\pm 0.00}$ | $26.24_{\pm 0.08}$ | $47.79_{\pm 0.07}$ | $32.64_{\pm 0.07}$ | $47.76_{\pm 0.11}$ | $32.14_{\pm 0.04}$ |
| SECap | $45.72_{\pm 0.09}$ | $26.52_{\pm 0.01}$ | $32.88_{\pm 0.03}$ | $52.26_{\pm 0.03}$ | $37.55_{\pm 0.03}$ | $52.11_{\pm 0.03}$ | $37.71_{\pm 0.03}$ |
| Video-LLaVA | $47.07_{\pm 0.16}$ | $25.47_{\pm 0.12}$ | $30.73_{\pm 0.11}$ | $54.65_{\pm 0.10}$ | $37.65_{\pm 0.24}$ | $54.54_{\pm 0.02}$ | $38.25_{\pm 0.22}$ |
| SALMONN | $47.96_{\pm 0.04}$ | $23.57_{\pm 0.02}$ | $28.83_{\pm 0.03}$ | $54.90_{\pm 0.15}$ | $38.93_{\pm 0.15}$ | $54.29_{\pm 0.06}$ | $37.79_{\pm 0.07}$ |
| VideoChat2 | $49.07_{\pm 0.26}$ | $26.92_{\pm 0.09}$ | $31.40_{\pm 0.10}$ | $52.38_{\pm 0.13}$ | $36.44_{\pm 0.11}$ | $53.56_{\pm 0.13}$ | $36.91_{\pm 0.11}$ |
| Video-ChatGPT | $50.52_{\pm 0.06}$ | $28.99_{\pm 0.04}$ | $34.05_{\pm 0.05}$ | $57.66_{\pm 0.04}$ | $41.48_{\pm 0.09}$ | $57.37_{\pm 0.00}$ | $40.95_{\pm 0.08}$ |
| LLaMA-VID | $51.25_{\pm 0.09}$ | $28.28_{\pm 0.04}$ | $32.85_{\pm 0.03}$ | $56.59_{\pm 0.04}$ | $41.22_{\pm 0.02}$ | $57.49_{\pm 0.03}$ | $40.39_{\pm 0.04}$ |
| mPLUG-Owl | $52.73_{\pm 0.13}$ | $27.47_{\pm 0.17}$ | $32.47_{\pm 0.19}$ | $57.60_{\pm 0.23}$ | $41.32_{\pm 0.04}$ | $56.32_{\pm 0.26}$ | $40.83_{\pm 0.07}$ |
| Chat-UniVi | $53.08_{\pm 0.01}$ | $28.89_{\pm 0.02}$ | $33.23_{\pm 0.08}$ | $57.00_{\pm 0.06}$ | $42.25_{\pm 0.04}$ | $57.50_{\pm 0.03}$ | $42.43_{\pm 0.03}$ |
| PCC score | — | 0.887 | 0.857 | 0.911 | 0.940 | 0.913 | 0.942 |

| Model | M3-W3 | | M3-W4 | | M3-W5 | | M-avg |
|---|---|---|---|---|---|---|---|
| | L1 | L2 | L1 | L2 | L1 | L2 | |
| Qwen-Audio | $39.46_{\pm 0.28}$ | $30.65_{\pm 0.01}$ | $36.64_{\pm 0.03}$ | $27.33_{\pm 0.01}$ | $35.89_{\pm 0.08}$ | $29.66_{\pm 0.01}$ | 31.84 |
| Otter | $51.03_{\pm 0.04}$ | $37.12_{\pm 0.00}$ | $47.54_{\pm 0.00}$ | $34.77_{\pm 0.00}$ | $50.51_{\pm 0.03}$ | $35.54_{\pm 0.00}$ | 39.41 |
| Video-LLaMA | $47.50_{\pm 0.20}$ | $36.50_{\pm 0.25}$ | $52.97_{\pm 0.09}$ | $35.78_{\pm 0.14}$ | $46.39_{\pm 0.12}$ | $34.77_{\pm 0.23}$ | 40.31 |
| VideoChat | $46.78_{\pm 0.11}$ | $34.37_{\pm 0.03}$ | $49.53_{\pm 0.15}$ | $32.82_{\pm 0.01}$ | $45.93_{\pm 0.18}$ | $32.85_{\pm 0.04}$ | 37.58 |
| SECap | $50.77_{\pm 0.03}$ | $40.49_{\pm 0.03}$ | $50.43_{\pm 0.03}$ | $38.21_{\pm 0.03}$ | $49.97_{\pm 0.03}$ | $40.25_{\pm 0.03}$ | 42.43 |
| Video-LLaVA | $52.29_{\pm 0.05}$ | $40.58_{\pm 0.15}$ | $52.45_{\pm 0.06}$ | $39.91_{\pm 0.13}$ | $52.97_{\pm 0.10}$ | $39.69_{\pm 0.10}$ | 43.27 |
| SALMONN | $56.25_{\pm 0.01}$ | $43.01_{\pm 0.02}$ | $50.53_{\pm 0.09}$ | $38.54_{\pm 0.03}$ | $53.65_{\pm 0.04}$ | $42.09_{\pm 0.02}$ | 43.53 |
| VideoChat2 | $52.14_{\pm 0.23}$ | $40.57_{\pm 0.14}$ | $50.63_{\pm 0.19}$ | $39.64_{\pm 0.18}$ | $51.37_{\pm 0.14}$ | $39.89_{\pm 0.15}$ | 42.65 |
| Video-ChatGPT | $55.50_{\pm 0.13}$ | $44.15_{\pm 0.18}$ | $55.24_{\pm 0.02}$ | $42.42_{\pm 0.05}$ | $52.93_{\pm 0.05}$ | $41.54_{\pm 0.14}$ | 46.02 |
| LLaMA-VID | $55.12_{\pm 0.05}$ | $44.06_{\pm 0.01}$ | $56.62_{\pm 0.15}$ | $42.42_{\pm 0.03}$ | $53.03_{\pm 0.08}$ | $41.65_{\pm 0.04}$ | 45.81 |
| mPLUG-Owl | $55.67_{\pm 0.19}$ | $43.71_{\pm 0.13}$ | $55.06_{\pm 0.17}$ | $40.67_{\pm 0.19}$ | $54.44_{\pm 0.13}$ | $42.00_{\pm 0.18}$ | 45.63 |
| Chat-UniVi | $56.80_{\pm 0.01}$ | $45.66_{\pm 0.05}$ | $55.86_{\pm 0.07}$ | $41.97_{\pm 0.09}$ | $55.81_{\pm 0.02}$ | $43.61_{\pm 0.05}$ | 46.75 |
| PCC score | 0.904 | 0.927 | 0.899 | 0.922 | 0.885 | 0.894 | 0.942 |

**GPT-based vs. EW-based Grouping.** This paper proposes two grouping strategies: GPT-based grouping and EW-based grouping. In this section, we explore the relationship between them and analyze whether the EW-based method can replace the GPT-based method. We calculate the scores for different types of EW-based grouping strategies, with the average score denoted as M-avg. Table 4 reports $F_S$, as this metric is used for the final ranking, and we also compute the PCC scores between different metrics. We observe that the PCC score between M-avg and GPT is relatively high, indicating that EW-based metrics can serve as alternatives to GPT-based metrics.

## 6 LIMITATIONS

Firstly, the main contribution of this paper is the definition of a new task and the conduct of foundational research. In the future, we plan to design more effective frameworks to solve OV-MER. Specifically, we plan to incorporate more emotion-related instruction datasets to finetune MLLMs, thereby enhancing their emotion recognition ability. Meanwhile, as mentioned in our paper, how to integrate subtitle information and fuse multimodal inputs also plays a crucial role in the final performance. We will also consider these aspects in the framework design. Secondly, we have evaluated some representative MLLMs, but not all models are covered. In the future, we will expand the scope of evaluation to cover more emerging MLLMs to enrich the benchmark. Thirdly, this paper does not involve cultural differences. Specifically, our original data is in Chinese, and the annotators we hired are also native Chinese speakers. In the future, we will also try to extend our method to other cultures and further analyze cultural differences.

## 7 CONCLUSION

This paper extends traditional MER to OV-MER, allowing for the prediction of arbitrary numbers and types of emotions. To facilitate further research, we construct an initial dataset, define evaluation metrics, and propose solutions. We observe that current MLLMs struggle to achieve satisfactory results, as this task requires consideration of multimodal clues and subtle temporal changes, placing higher demands on MLLMs. Additionally, EW-based metrics can replace GPT-based metrics, thus reducing evaluation costs while ensuring reproducibility. This paper advances current research from basic to nuanced emotion recognition, which is crucial for building emotion AI.

ETHICS STATEMENT

The raw data of the OV-MERD dataset comes from the MER2023 dataset, from which we evenly select some samples with further annotation. Therefore, we do not collect new data; we just re-annotate existing data. This annotation process has received consent from the dataset owners and has passed our internal review. During the annotation process, we generously pay each annotator approximately ¥3,000 (around $280), which is considered high. After proofreading our annotation results, we find that the annotations focus on the multimodal clues present in the videos, without any discriminatory annotations. Additionally, we restrict the use of the OV-MERD dataset to non-commercial purposes under the CC BY-NC 4.0 license. This license clearly outlines the correct and responsible use of our dataset. Details of our license are provided in the supplementary materials. In summary, this paper does not involve any ethical issues.

REPRODUCIBILITY STATEMENT

This paper provides the source code and intermediate results in the supplementary materials. Since we build a dataset for OV-MER, we also offer a complete description of the data construction process in Section 2 and Appendix. In summary, we have made every effort to ensure its reproducibility.

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

## A  DETAILED MOTIVATION

**Limitations of Dimensional Models.**  Although dimensional models can be used to distinguish different emotions, they are abstract and difficult for the general public to understand. For instance, the VAD model is one of the most widely used dimensional models. Given a VAD score (V=3.3, A=2.9, D=4.7), it is challenging for people to accurately interpret the corresponding emotion. According to previous research (Verma & Tiwary, 2017), this VAD score represents *sad*. Furthermore, the VAD model attempts to simplify emotions into three linear dimensions. However, emotions are inherently complex and often involve combinations of multiple emotional states, which cannot be fully captured by these three dimensions. Additionally, discrete models align more closely with how humans intuitively understand emotions, making it easier to achieve consensus across different annotators. Therefore, this paper uses discrete models to describe emotions.

**OV-MER vs. Existing Emotion Models.**  OV-MER combines the advantages of both discrete and dimensional models. Compared to one-hot discrete models, OV-MER uses denser discrete labels to capture more subtle emotions, mimicking the continuous attributes of dimensional models. Compared to dimensional models, OV-MER uses discrete labels, making it closer to the way humans understand emotions. Therefore, our OV-MER provides a bridge connecting discrete and dimensional models.

**Video Emotion vs. Facial Emotion.**  Video emotion is more complex than facial emotion. This is because, in videos, we need to capture subtle changes in the temporal dimension and integrate multimodal clues. Take Figure 1(b) as an example. In the temporal dimension, we need to infer a person's *nervousness* based on his stuttering; in the multimodal dimension, we need to combine information from different modalities to gain a more comprehensive understanding of emotion. Due to the complexity of video emotion, using a single label is limiting, and more discrete labels are required to better describe video emotion. This is also the motivation behind our OV-MER task.

**Label Importance.**  In OV-MER, we do not assign different levels of importance to each label. Every emotion holds equal significance, and neglecting anyone can impact the performance of downstream tasks. For example, if a human-computer interaction system only captures basic emotions while overlooking nuanced ones, it may fail to generate appropriate responses.

## B  RELATED WORK

**Multimodal Emotion Recognition.**  MER has rapidly developed in recent years (Wu et al., 2014). Current research mainly focuses on building more efficient architectures to achieve higher accuracy on benchmark datasets (Sun et al., 2024). For example, Zadeh et al. (2017) proposed a tensor fusion network that addressed the MER task by leveraging interactions among unimodal, bimodal, and trimodal inputs. Tsai et al. (2019) introduced a Transformer-based model that learned implicit alignment between different modalities and achieved promising results. Lian et al. (2024b) further established MERBench, involving various features, fusion strategies, and datasets. In emotion recognition, benchmark datasets usually limit the label space to basic emotions and use majority voting to determine the most likely one or more labels (Lian et al., 2023; Li et al., 2017). However, emotional categories extend far beyond basic emotions. Restricting the label space will inevitably overlook some nuanced emotions. To address this issue, we extend traditional MER to OV-MER, which allows for the prediction of any number and categories of emotions.

**Open Vocabulary Learning.**  Its main goal is to identify categories beyond the annotated label space (Wu et al., 2024), which has been applied in various fields, such as object detection (Zareian et al., 2021), segmentation (Ghiasi et al., 2022), and scene understanding (Li et al., 2021). For example, the object detection dataset COCO (Lin et al., 2014) contains 80 categories, while objects in the real world are nearly infinite, highlighting the importance of open vocabulary learning. This paper makes the first attempt to address MER in an open-vocabulary manner. Compared to other tasks, MER is more difficult as it requires considering multimodal clues and subtle temporal variations.

## C   MORE EXAMPLES

In Figures 9~11, we provide more examples to visualize the difference between one-hot and OV labels. This paper uses emotion-related descriptions as a bridge for OV label extraction. Since the original video contains real people, we use DemoAI to remove personal information to address copyright concerns. Our OV-MERD dataset is derived from the MER2023 dataset (Lian et al., 2023) with further annotations. Therefore, for the original data, please download the MER2023 dataset.

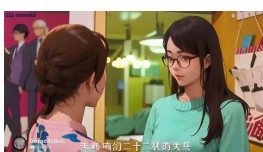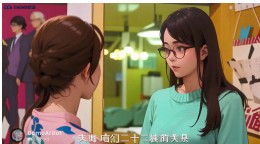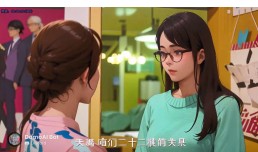

**Subtitle:** Goodness, the relationships on our 22nd floor are getting more and more complicated. It seems like I'll have to speak less in the future.

**One-hot Label:** *surprise*

**Description:** In the video, there are two ladies in the frame. The lady on the left has her back to the camera, while the lady on the right is facing the camera. We mainly analyze the emotional state of the lady on the right. At the beginning of the video, her facial expression is not positive, and she seems a bit nervous and uneasy. In the following scenes, her eyebrows are raised, indicating that she is emphasizing something. At the end of the video, she lowers her head and looks down, not looking at the other person while speaking. This behavior may stem from inner pressure, anxiety, uneasiness, or frustration, causing her to tend to hide her emotions or avoid communication with others. In the audio, the volume is low, the tone is low, and there is a sense of pressure in the voice. In the text, the subtitle says, "Goodness, the relationships on our 22nd floor are getting more and more complicated. It seems like I'll have to speak less in the future." Based on the tense and uneasy facial expression, the emphasized movement of raising eyebrows, and the behavior of lowering her head and looking down, it can be inferred that she may be feeling pressure, anxiety, uneasiness, or frustration. Additionally, based on the audio clue describing low volume, low tone, and a sense of pressure in the voice, it can be further confirmed that her emotional state is likely negative. Therefore, this sentence expresses her exclamation about the increasing complexity of the relationship on the 22nd floor and her decision to speak less in the future, implying her uneasiness and pressure.
**Description → OV labels:** *pressure, negative, uneasy, surprised*

Figure 9: Example1.

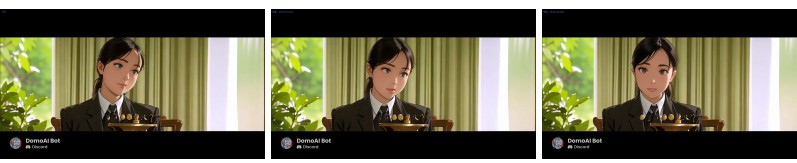

**Subtitle:** Reading requires literacy! In movies, even the illiterate can enjoy themselves, but what really competes with you is cross-talk.

**One-hot Label:** *happy*

**Description:** In the video, at the beginning, we see a woman sitting indoors with bright windows and green plants in the background. Her facial expression is calm, with her head tilted to the right and her mouth slightly open, as if she is engaged in a lighthearted conversation. As time goes by, her smile becomes more pronounced, and she slightly shakes her head to the left, glancing to the left as if referring to someone or something out of view. Towards the end of the video, she stops smiling, leans forward, and opens her mouth slightly, seemingly emphasizing something to the other person. In the audio, the character's tone is relatively steady, laced with humor. However, in the first sentence, there are noticeable emphases: "reading" has an upward intonation, while "literacy" has a downward intonation, creating a contrast. The subtitles read: "Reading requires literacy! In movies, even the illiterate can enjoy themselves, but what really competes with you is cross-talk." This sentence might be the woman's comment or reaction to someone nearby. Given the audio cues of a steady tone with humor and the woman's smile and glance to the left, we can infer that her words carry a mocking or sarcastic tone. Thus, this sentence likely expresses the woman's mockery of the other person's lack of understanding or recognition of something, conveyed in a humorous manner.
**Description → OV labels:** *humor, mockery, irony*

Figure 10: Example2.

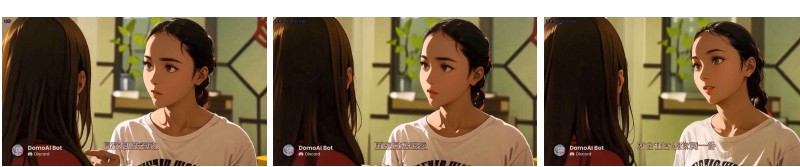

**Subtitle:** Luyuan will never fall into your hands, because he has found a high-paying job.

**One-hot Label:** *angry*

**Description:** In the video, the frame includes two women. The woman on the left has her back to the camera, while the woman on the right is facing the camera. We mainly analyze the emotional state of the woman on the right. At the beginning of the video, she slightly furrows her brows, opens her mouth, and her facial expression appears angry, as if she is engaged in a heated argument with the other person. At the same time, she points her finger at the other person and makes a motion towards them, which may indicate that she is accusing the other person or emphasizing her own viewpoint. Overall, she may be going through a debate or intense conversation, and her emotional state may be one of excitement and anger. She seems to be accusing and expressing her dissatisfaction. In the audio, the tone is aggressive and the character's emotions are more excited. Combined with the text content, the tone seems to carry a sense of threat. In the text, the subtitle says, "Luyuan will never fall into your hands, because he has found a high-paying job." This sentence may be an accusation or threat from the woman on the right to the woman on the left. Based on the angry and angry emotions displayed by the woman on the right in the video clues, as well as her pointing finger and motion towards the other person, it can be inferred that she is accusing the other person or emphasizing her own viewpoint. At the same time, based on the aggressive tone and excited emotions described in the audio clues, as well as the mention in the subtitle that Luyuan has found a high-paying job, it can be inferred that this sentence may carry a sense of threat, and the woman on the right may be threatening the woman on the left not to interfere with or harm Luyuan. Therefore, this sentence expresses the woman on the right's anger and threatening emotions.
**Description → OV labels:** *warning, angry, threat*

Figure 11: Example3.

# D SUMMARY OF ABBREVIATIONS

In Table 5, we summarize the main abbreviations and their meanings.

Table 5: Summary of main abbreviations.

| Category | Abbreviation | Explanation |
|---|---|---|
| Label | OH label | One-hot label, the most likely label in a limited set of basic emotions. |
| | OV labels | Open-vocabulary labels, a set of labels in an unlimited label space. |
| Task | MER | Multimodal Emotion Recognition, which aims to recognize the one-hot emotion label. |
| | OV-MER | Open-vocabulary MER, which aims to identify the OV emotion labels. |
| Dataset | OV-MERD | This is a dataset we built for the OV-MER task. |
| Metric | EW | Emotion Wheel. |
| | M1, M2, M3 | Different grouping strategies based on the emotion wheel. |
| | $Precision_S$, $Recall_S$, $F_S$ | Metrics defined for OV-MER. |
| Model | LLM | Large Language Model. Large-scale models and only process text. |
| | ALLM | Audio LLM. Different from LLM, it can also process audio input. |
| | VLLM | Video LLM. Different from LLM, it can also process video input. |
| | MLLM | Multimodal LLM. Unlike LLM, it can process at least one more modality (e.g., audio or video). Thus, MLLM includes ALLM and VLLM. |
| Description | CLUE-Multi | It uses the checked acoustic and visual clues to generate descriptions. |
| | CLUE-Audio | Different from CLUE-Multi, it only uses checked acoustic clues. |
| | CLUE-Video | Different from CLUE-Multi, it only uses checked visual clues. |
| | CLUE-Text | It only relies on text to generate descriptions. |
| | CLUE-A/T/V | Any of CLUE-Audio, CLUE-Text, and CLUE-Video. |
| | CLUE-M/A/T/V | Any of CLUE-Multi, CLUE-Audio, CLUE-Text, and CLUE-Video. |
| | CLUE-MLLM | It uses the output from MLLM without any manual checking process. |
| | S0, S1, S2 | Different CLUE-MLLM generation strategies. |

# E   DETAILS IN DATASET CONSTRUCTION

**Prompts.**   Figure 2 presents our dataset construction process. In Table 6, we provide prompts and corresponding models used in this process.

Table 6: Prompts and corresponding models used in the dataset construction process.

| Function (Model) | Prompt |
|---|---|
| #1 Pre-label visual clue (VLLM) | As an expert in the field of emotions, please focus on facial expressions, body language, environmental cues, and events in the video and predict the emotional state of the character. Please ignore the character's identity. We uniformly sample 3 frames from this video. Please consider the temporal relationship between these frames and provide a complete description of this video. Avoid using descriptions like "the first image" and "the second image", and instead use terms like "beginning", "middle", and "end" to denote the progression of time. |
| #2 Pre-label acoustic clue (ALLM) | As an expert in the field of emotions, please focus on the acoustic information in the audio to discern clues related to the emotions of the individual. Please provide a detailed description and ultimately predict the emotional state of the individual. |
| #3 Merge (LLM) | Please act as an expert in the field of emotions. We provide acoustic and visual clues that may be related to the character's emotional state, along with the original subtitle of the video. Please analyze which parts can infer the emotional state and explain the reasons. During the analysis, please integrate the textual, audio, and visual clues. |
| #4 Translation (LLM) | *Chinese→English:* Please translate the following sentence from Chinese into English. *English→Chinese:* Please translate the following sentence from English into Chinese. |
| #5 OV label extraction (LLM) | Please assume the role of an expert in the field of emotions. We provide clues that may be related to the emotions of the characters. Based on the provided clues, please identify the emotional states of the main characters. Please separate different emotional categories with commas and output only the clearly identifiable emotional categories in a list format. If none are identified, please output an empty list. |

**Merging Process.**   In this paper, we rely on the powerful reasoning capabilities of LLM for multimodal fusion. Specifically, as shown in Table 6, we ask LLM to integrate textual, acoustic, and visual clues to infer the emotional state. From the output, we observe that LLM can produce reasonable analytical results. However, ambiguities and contradictions in multimodal fusion are inevitable, which is a challenging and open problem. In this paper, we simply use LLM to address this problem, providing a practical solution for the OV-MER task. However, more effective strategies may exist, such as using more complex prompts or incorporating modality control measures. Therefore, we plan to explore this area in the future.

**Visualization.** In Figure 12, we visualize the output of the main steps.

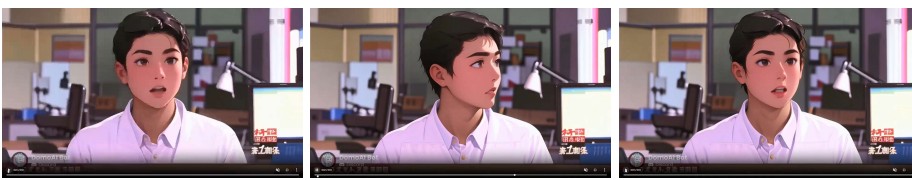

**Subtitle:** Why are you all looking at me like that? So, as long as it's a woman, does she have to have a relationship with me?

**Visual Clue (Prelabeled):** In the opening scene, we see a man wearing a white lab coat. He looks confident, with a slight smile on his lips and eyes directly facing the camera, as if he is explaining or narrating something. The background features computers and office equipment, suggesting that this might be an office or laboratory setting. As the scene progresses to the middle part, the man's head turns slightly to one side, and his gaze shifts sideways, seemingly interacting with someone next to him. His smile broadens, and his eyebrows lift slightly, indicating that he might be having a friendly conversation or sharing a light-hearted moment. In the next scene, the man's expression becomes more exaggerated; his mouth opens wide as if expressing surprise or emphasizing a point. This kind of facial expression and body language is typically associated with sharing surprising news or reacting strongly to something. In the final scene, he returns to facing the camera directly, his expression calm once again with a slight smile, giving the impression that he might have just finished explaining or narrating something and feels satisfied or confident. Overall, based on these scenes, we can infer that the man in the video is likely conducting some sort of demonstration or explanation. He interacts with the audience, shares information, and possibly expresses surprise at some point. His emotional state throughout the video appears to be positive, exhibiting confidence and a friendly demeanor.

**Visual Clue (Checked):** In the video, the screen shows a male character in an indoor setting. At the beginning of the video, his eyes are wide open and his mouth is also open, indicating a surprised facial expression. In the following scenes, he looks around, seemingly explaining or narrating something to the people around him. Overall, his emotions are not positive or optimistic.

**Acoustic Clue (Prelabeled):** The character in the voice is expressing frustration and annoyance. The tone is sarcastic and mocking, and the intonation is rising at the end of each sentence. There are pauses between each sentence, and the stress is on certain words to emphasize the frustration and annoyance.

**Acoustic Clue (Checked):** In the audio, the character speaks with a stutter, which usually expresses feelings of nervousness, anxiety, or unease. Combined with the text content, the character seems to be unhappy and angry due to the prejudice of the people around him.

**CLUE-Multi:** In the video, the screen shows a male character in an indoor setting. At the beginning of the video, his eyes are wide open and his mouth is also open, indicating a surprised facial expression. In the following scenes, he looks around, seemingly explaining or narrating something to the people around him. Overall, his emotions are not positive or optimistic. In the audio, the character speaks with a stutter, which usually expresses feelings of nervousness, anxiety, or unease. Combined with the text content, the character seems to be unhappy and angry due to the prejudice of the people around him. The subtitle in the text says, "Why are you all looking at me like that? So, as long as it's a woman, does she have to have a relationship with me?" This sentence expresses the male character's dissatisfaction and anger towards the people around him. Based on the surprised and negative facial expression of the male character in the video clues, as well as the stuttering speech in the audio clues, we can infer that the male character is expressing a feeling of dissatisfaction and anger in this sentence. He may feel troubled by the prejudice of the people around him and is unhappy with this unfair treatment.

Figure 12: An example to visualize the output of the main steps.

# F   ONE-HOT VS. OV LABELS

This section provides a deeper comparison between the one-hot labels in the MER2023 dataset and the OV labels in the OV-MERD dataset. Figure 13 shows the word cloud and label number distribution of OV labels. In Figure 13(a), we observe that OV labels cover a wider variety of emotions, some of which (such as *shy*, *nervous*, and *grateful*) are rarely discussed in previous datasets. In Figure 13(b), we notice that most samples have about 2 to 4 labels, much more than the traditional task where each sample is assigned only one emotion. Therefore, OV-MER provides richer labels.

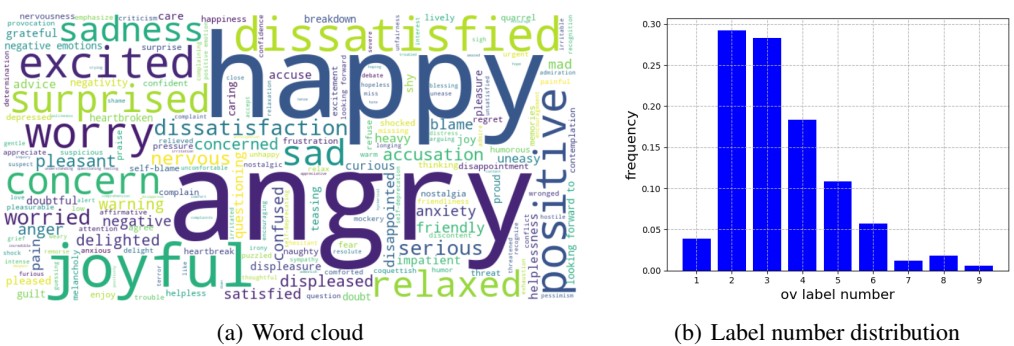

(a) Word cloud                              (b) Label number distribution

Figure 13: Word cloud and label number distribution of OV labels.

In Table 7, we report the performance of one-hot labels in OV-MER. We observe that one-hot labels have high *precision_S* but low *recall_S*, indicating that one-hot labels are correct but not comprehensive. Due to the limited label space and the constrained number of labels, one-hot labels cannot cover all emotions, highlighting the limitations of traditional MER and the importance of OV-MER. Additionally, these results reflect the necessity to use $F_S$ for the final ranking, which can balance accuracy and completeness.

Table 7: Performance of one-hot labels in OV-MER.

| Language | $F_S$ ↑ | Precision$_S$ ↑ | Recall$_S$ ↑ |
|---|---|---|---|
| English | $65.71_{\pm0.06}$ | $92.17_{\pm0.00}$ | $51.05_{\pm0.08}$ |
| Chinese | $66.16_{\pm0.02}$ | $93.07_{\pm0.00}$ | $51.32_{\pm0.03}$ |

Figure 14 shows the emotion distribution of OV labels. We observe that the number of samples for different emotions follows a long-tail distribution. These results indicate that OV labels not only cover some major labels but also capture subtle emotions that occur infrequently.

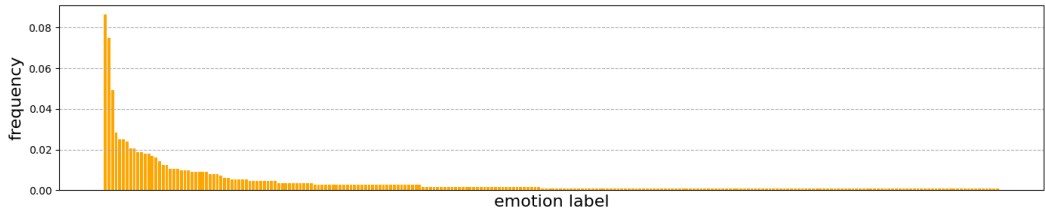

Figure 14: Emotion distribution of OV labels.

# G  DATASET COMPARISON

This paper introduces a new task, OV-MER, and constructs a dataset for this task called OV-MERD. Table 8 compares OV-MERD with existing datasets. The annotation types of these datasets can be broadly categorized into two types: dimensional emotions and discrete emotions. We classify sentiment analysis datasets (e.g., CMU-MOSI) as dimensional datasets because the definition of sentiment intensity overlaps with the valence in dimensional emotions. We observe that OV-MERD contains 248 emotion categories, with most samples having 2 to 4 labels, significantly exceeding the number of labels in existing datasets. In the future, as the scale of the dataset increases, the number of candidate labels can be further expanded. Meanwhile, we would like to emphasize that our OV-MERD is the first dataset that uses the human-LLM collaborative annotation strategy, aiming to provide richer labels to capture more nuanced emotions. We believe this work is an important extension of traditional MER and will contribute to the development of the field.

Table 8: Dataset comparison. In this table, "I", "A", "V", and "T" are abbreviations for image, audio, video, and text, respectively. Some datasets (such as CMU-MOSEI and MSP-Podcast) contain both discrete and dimensional emotions.

| Dataset | Modality | Annotation Type | # Categories | # Labels per Sample |
|---|---|---|---|---|
| MSP-Podcast (Lotfian & Busso, 2017) | A | Dimensional | 3 | 1 |
| SST (Socher et al., 2013) | T | Dimensional | 1 | 1 |
| Cornell (Pang et al., 2002) | T | Dimensional | 1 | 1 |
| Large Movie (Maas et al., 2011) | T | Dimensional | 1 | 1 |
| ICT-MMMO (Wöllmer et al., 2013) | A,V,T | Dimensional | 1 | 1 |
| YouTube (Morency et al., 2011) | A,V,T | Dimensional | 1 | 1 |
| MOUD (Pérez-Rosas et al., 2013) | A,V,T | Dimensional | 1 | 1 |
| CMU-MOSI (Zadeh et al., 2017) | A,V,T | Dimensional | 1 | 1 |
| CMU-MOSEI (Zadeh et al., 2018) | A,V,T | Dimensional | 1 | 1 |
| CH-SIMS (Yu et al., 2020) | A,V,T | Dimensional | 1 | 1 |
| CH-SIMS v2 (Liu et al., 2022a) | A,V,T | Dimensional | 1 | 1 |
| VAM (Grimm et al., 2008) | A,V,T | Dimensional | 3 | 1 |
| SEMAINE (McKeown et al., 2011) | A,V,T | Dimensional | 5 | 1 |
| AFEW-VA (Kossaifi et al., 2017) | A,V,T | Dimensional | 2 | 1 |
| SEWA(Kossaifi et al., 2019) | A,V,T | Dimensional | 3 | 1 |
| MSP-Podcast (Lotfian & Busso, 2017) | A | Discrete | 8 | 1 |
| JL-Corpus (James et al., 2018) | A | Discrete | 10 | 1 |
| EmoDB (Burkhardt et al., 2005) | A | Discrete | 7 | 1 |
| EMOVO (Costantini et al., 2014) | A | Discrete | 7 | 1 |
| MESD (Duville et al., 2021) | A | Discrete | 6 | 1 |
| SFEW 2.0 (Dhall et al., 2015) | I | Discrete | 7 | 1 |
| FER-2013 (Goodfellow et al., 2013) | I | Discrete | 7 | 1 |
| EmotioNet (Fabian Benitez-Quiroz et al., 2016) | I | Discrete | 23 | 1 |
| AffectNet (Mollahosseini et al., 2017) | I | Discrete | 7 | 1 |
| ExpW (Zhang et al., 2018) | I | Discrete | 7 | 1 |
| RAF-DB (Li et al., 2017) | I | Discrete | 19 | 1~2 |
| CMU-MOSEI (Zadeh et al., 2018) | A,V,T | Discrete | 6 | 1 |
| eNTERFACE (Martin et al., 2006) | A,V,T | Discrete | 6 | 1 |
| SAVEE (Jackson & Haq, 2014) | A,V,T | Discrete | 7 | 1 |
| AFEW 7.0 (Dhall et al., 2017) | A,V,T | Discrete | 7 | 1 |
| MAFW (Liu et al., 2022b) | A,V,T | Discrete | 11 | 1 |
| DFEW (Jiang et al., 2020) | A,V,T | Discrete | 7 | 1 |
| CREMA-D (Cao et al., 2014) | A,V,T | Discrete | 6 | 1 |
| MSP-IMPROV (Busso et al., 2016) | A,V,T | Discrete | 4 | 1 |
| RAVDESS Livingstone & Russo (2018) | A,V,T | Discrete | 8 | 1 |
| IEMOCAP (Busso et al., 2008) | A,V,T | Discrete | 10 | 1 |
| MELD (Poria et al., 2019) | A,V,T | Discrete | 7 | 1 |
| MC-EIU (Liu et al., 2024) | A,V,T | Discrete | 7 | 1 |
| MER2023 (Lian et al., 2023) | A,V,T | Discrete | 6 | 1 |
| MER2024 (Lian et al., 2024a) | A,V,T | Discrete | 6 | 1 |
| **OV-MERD (Ours)** | **A,V,T** | **Discrete** | **248 (arbitrary label)** | **1~9, most 2~4 (arbitrary number)** |

## H    DURATION DISTRIBUTION OF OV-MERD

In Figure 15, we analyze the duration distribution of the OV-MERD dataset. We observe that the majority of the samples have durations ranging from 1 to 4 seconds. This distribution is consistent with that of the MER2023 dataset, which was used as the original dataset for constructing OV-MERD (see Section 2.3 for details).

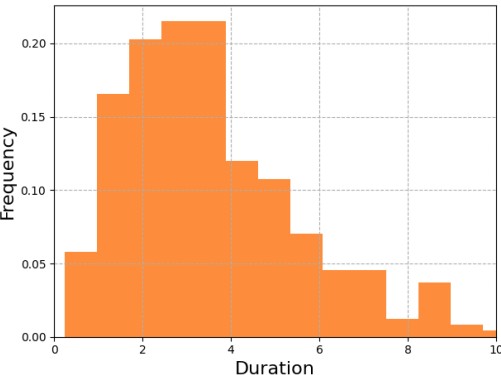

Figure 15: Duration distribution of the OV-MERD dataset.

## I    NUMBER OF SAMPLED FRAMES IN PRE-ANNOTATION

To generate pre-annotated visual clues, we sample three frames from each video and input them into GPT-4V. In this section, we discuss the rationale behind the choice of the number of sampled frames. Specifically, we categorize visual clues into two types: (1) visual clues with relatively long durations; and (2) visual clues with fast movements, such as eye movements, head movements, and micro-expressions.

**For the first type**, since the duration of most videos is between 1 and 4 seconds (see Appendix H) and the video content is usually continuous with only minor differences between adjacent frames (see Appendix C), uniformly sampling three frames are sufficient to capture this information; **For the second type**, we observe that current MLLMs (including the GPT-4V used in this paper) struggle to capture these fast movements. Increasing the number of sampled frames does not address this issue. Previous research has also shown that GPT-4V cannot recognize micro-expressions (Lian et al., 2024c). To capture these fast movements, we employ multiple professional annotators to manually add this information.

## J    ANNOTATION DETAILS

This section presents our annotation guidelines and the layout of the annotation platform. Our annotation process relies on the Label Studio (Tkachenko et al., 2020) toolkit. As shown in Figure 2, there are two parts that require manual checking: 1) the pre-annotated acoustic and visual clues; 2) the merged open-vocabulary labels. To reduce subjective bias, we hire eight annotators who are experts in affective computing and familiar with the definitions of emotions. Additionally, we conduct two rounds of checks with no overlap among annotators in each round. Specifically, in the first round, we randomly select four annotators to check the clues and labels; in the second round, we merge the clues and labels reviewed by the first four annotators and ask another four annotators to perform a second round of checks. Ultimately, we find that these checked clues and labels are well-aligned with the video content.

Figure 16 shows the layout of the annotation platform used for manually checking acoustic and visual clues. During the annotation process, we use the following instructions: We provide pre-labeled acoustic and visual clues. Please manually check these clues, remove errors, and add missing information. On the annotation platform, we design an interface with a time slider, allowing annotators

to start playing the video from any frame. This enables annotators to view the entire video during the manual check, helping them better annotate the details that may be missed in pre-annotation.

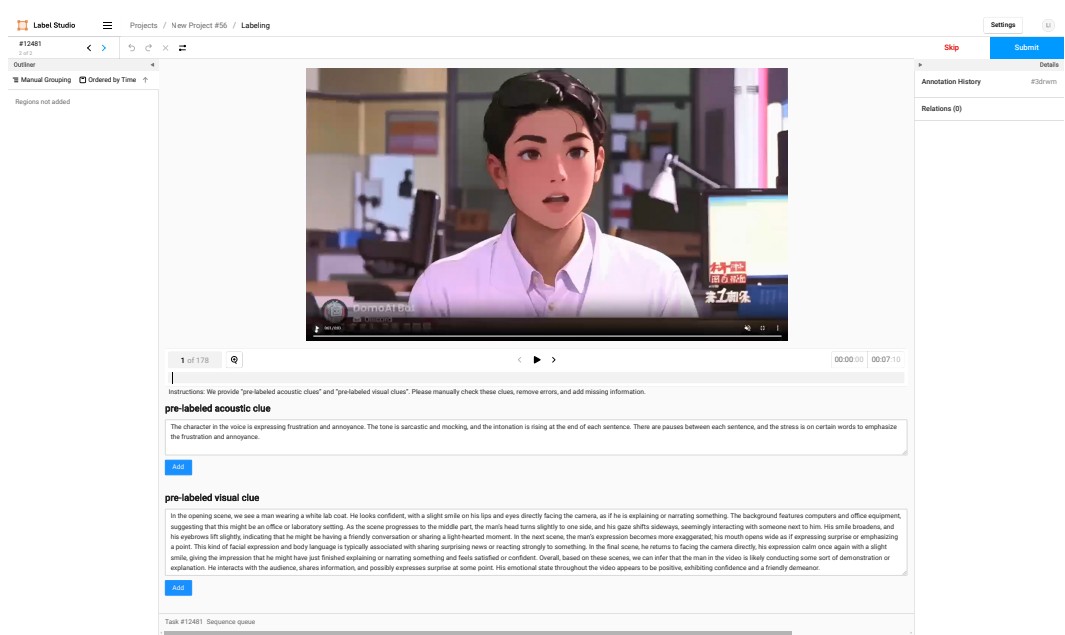

Figure 16: Layout of the annotation platform used for manually checking acoustic and visual clues.

Figure 17 displays the layout of the annotation platform used for manually checking emotional labels. During annotation, we use the following instructions: Please select all labels that match the character's emotional state in the "candidate emotions". If the provided candidate labels cannot perfectly describe the character's emotional state, you can also manually add new labels to the "other emotions" part. Specifically, annotators need to label two parts. First, we list all candidate labels from which annotators can choose what they believe to be the correct labels; second, when the candidate labels cannot perfectly describe the emotions, annotators can manually add additional labels in the "other emotions" part.

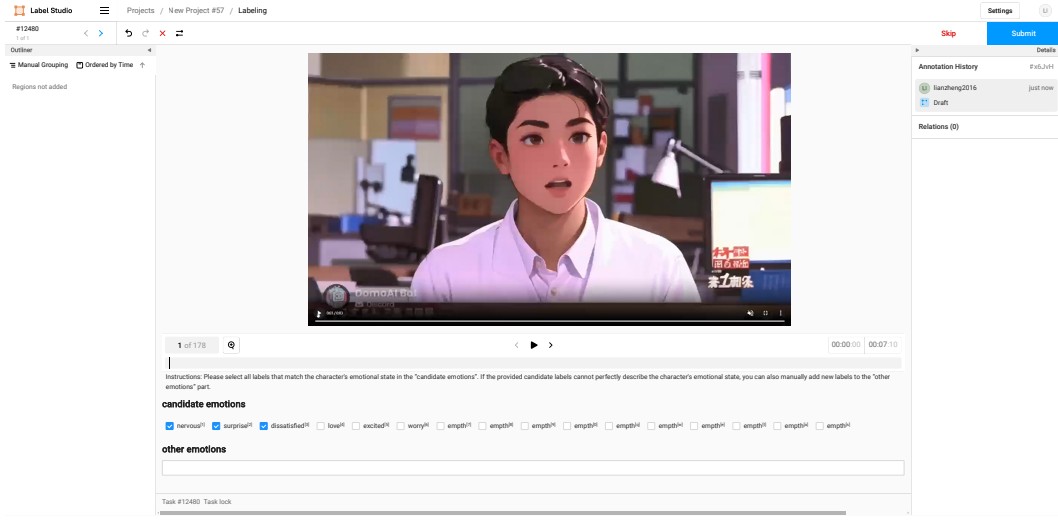

Figure 17: Layout of the annotation platform used for manually checking emotional labels.

To illustrate which labels are removed or kept, we provide two examples, each with labels from multiple annotators. To ensure annotation quality, we hire professional annotators who are experts in affective computing and familiar with the definition of emotions. Some of these annotators are members of our team who specialize in affective computing. In Figure 18, as the character doesn't know which doctor to see, most annotators provide labels such as *confused* or *puzzled*. Based on his tone and expression, some annotators further provide labels like *anxious* and *serious*. In Figure 19, most annotators notice his *disapprove* based on the textual content. Combining other modalities, some annotators further note his *blame* and *accuse* of what others are planning to do. From these examples, we can observe that these annotators provided relatively reliable labels. However, some annotators may focus only on the most relevant labels and overlook some details. To ensure the comprehensiveness of the annotation results, we merge the labels checked by four annotators. For example, in Figure 18, the final merged labels are *troubled*, *focused*, *puzzled*, *anxious*, *worried*, *confused*, and *serious*. In the next round of checks, we invite another four annotators for a second round of checks. Through this process, we can ensure that each retained label is confirmed by at least one annotator in each round, thereby ensuring the comprehensiveness and accuracy of the annotation results.

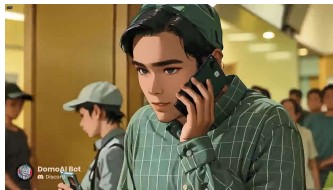 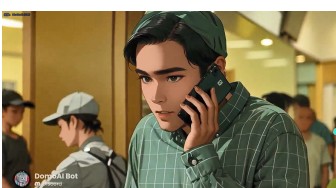 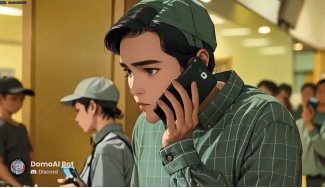

**Subtitle:** Huh? Which director?

**A1:** *troubled, focused, puzzled*

**A2:** *anxious, worried, confused*

**A3:** *confused, puzzled*

**A4:** *puzzled, anxious, confused, troubled, serious*

Figure 18: Example1 with labels from multiple annotators.

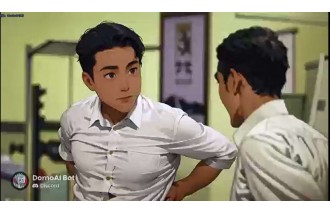 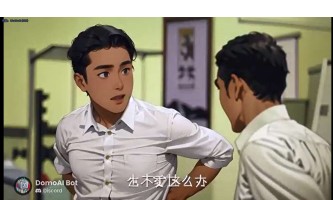 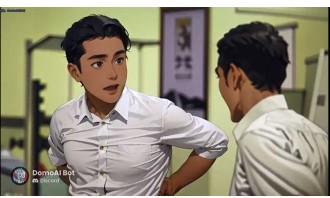

**Subtitle:** It's not right to do this, even if it's for the child's good!

**A1:** *surprised, dislike, disapprove, unexpected, worry*

**A2:** *disapprove, serious, worry*

**A3:** *surprised, serious, amazed, shocked*

**A4:** *disapprove, serious, accuse, blame*

Figure 19: Example2 with labels from multiple annotators.

## K  DETAILS OF LANGUAGE IMPACT EXPERIMENTS

**Experimental Design.**   In Figure 2, we analyze from two perspectives: 1) the impact of descriptive language (Clue-Multi), and 2) the impact of abstract language (OV labels). The $Y_{EE}$ to $Y_{EC}$ (or $Y_{CC}$ to $Y_{CE}$) experiment aims to keep the descriptive language consistent to analyze the effect of abstract language, while the $Y_{CE}$ to $Y_{EE}$ (or $Y_{EC}$ to $Y_{CC}$) experiment aims to keep the abstract language consistent to analyze the effect of descriptive language.

**Jaccard Similarity Coefficient.**   Figure 2 uses the Jaccard similarity coefficient to measure the similarity between two sets, which is slightly different from the evaluation metrics defined in Section 3. Specifically, in Section 3, we use the following metrics:

$$\text{Precision}_{\text{S}} = \frac{|\mathcal{Y} \cap \hat{\mathcal{Y}}|}{|\hat{\mathcal{Y}}|}, \ \text{Recall}_{\text{S}} = \frac{|\mathcal{Y} \cap \hat{\mathcal{Y}}|}{|\mathcal{Y}|}, \ \text{F}_{\text{S}} = 2 \times \frac{\text{Precision}_{\text{S}} \times \text{Recall}_{\text{S}}}{\text{Precision}_{\text{S}} + \text{Recall}_{\text{S}}}. \tag{5}$$

The motivation for the above metrics is that $\mathcal{Y}$ represents the ground truth, while $\hat{\mathcal{Y}}$ represents the prediction. However, in Figure 2, the two sets of emotions are considered equally important. Therefore, we use the Jaccard similarity coefficient to measure the similarity. This metric evaluates the similarity between two sets by comparing the size of their intersection to the size of their union:

$$\text{Similarity}_{\text{S}} = \frac{|\mathcal{Y} \cap \hat{\mathcal{Y}}|}{|\mathcal{Y} \cup \hat{\mathcal{Y}}|}. \tag{6}$$

## L  CLUE-MULTI ANALYSIS

In this section, we further analyze the reliability and comprehensiveness of CLUE-Multi from three aspects: discrete emotion recognition, dimensional emotion recognition, and visual clue statistics. Table 9 provides prompts and models for each part of the analysis.

Table 9: Prompts and corresponding models used in CLUE-Multi analysis.

| Function (Model) | Prompt |
|---|---|
| #1 Discrete Emotion Recognition (GPT-3.5) | Please assume the role of an expert in the emotional domain. We provide clues that may be related to the emotions of the character. Based on the provided clues, identify the emotional states of the main characters. We provide a set of emotional candidates, please rank them in order of likelihood from high to low. The candidate set is {happy, angry, worried, sad, surprise, neutral}. |
| #2 Valence Estimation (GPT-3.5) | As an expert in the emotional domain, we provide clues that may be related to the emotions of characters. Based on the provided clues, please identify the overall positive or negative emotional polarity of the main characters. The output should be a floating-point number ranging from -5 to +5. Here, -5 indicates extremely negative emotions, 0 indicates neutral emotions, and +5 indicates extremely positive emotions. Larger numbers indicate more positive emotions, while smaller numbers indicate more negative emotions. Please provide your judgment as a floating-point number with two decimal places, directly outputting the numerical result without including the analysis process. |
| #3 Visual Clue Analysis (GPT-3.5) | Please assume the role of an expert in the field of emotions. We provide clues related to the emotions of the characters in the video. Please output the facial movements and body gestures involved in the description, separated by commas. The output format should be in list form. |

**Discrete Emotion Recognition.**   Our dataset is based on MER2023, which provides relatively reliable one-hot labels. Therefore, we attempt to determine whether these one-hot labels can be identified from CLUE-Multi. This part of the analysis aims to verify whether CLUE-Multi can cover the traditional one-hot emotion recognition task. Experimental results indicate that the top-1 and top-2 scores can reach 93.48 and 96.89, respectively. Further analysis shows that the prediction errors are primarily due to the limitations of one-hot labels. For example, in Figure 1, the character shows a compound emotional state, including *surprised*, *nervous*, and *unsatisfied*. However, when we rank the candidate emotions, the output is: *angry, surprised, worried, neutral, sad, happy*. The top-1 label is *angry*, which differs from *surprise* in MER2023, leading to a prediction error. These results reveal the limitations of traditional one-hot labels in describing emotions.

**Valence Estimation.** Besides discrete labels, MER2023 also provides relatively reliable valence scores. Therefore, we attempt to verify whether CLUE-Multi can used for valence estimation. Through experimental analysis, we observe that the PCC score between predictions and annotations can reach 0.88, indicating that CLUE-Multi also contains clues for dimensional emotion recognition.

**Visual Clue Analysis.** Following that, we attempt to analyze the diversity of visual clues in CLUE-Multi. Through experimental analysis, we observe that each sample has an average of 4.95 visual clues. Therefore, we conclude that CLUE-Multi contains a wealth of clues that can help address discrete emotion recognition and valence estimation. Additionally, these results validate the completeness and reliability of CLUE-Multi.

## M    VISUALIZATION OF CLUE-M/A/T/V

Figure 20 provides an example and visualizes CLUE-M/A/T/V.

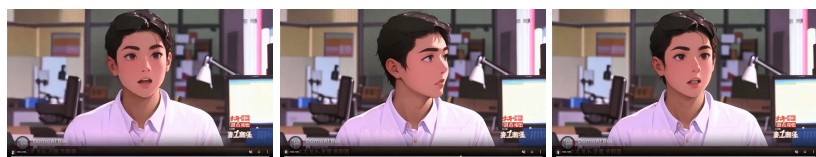

**Subtitle:** Why are you all looking at me like that? So, as long as it's a woman, does she have to have a relationship with me?

**CLUE-Multi:** In the video, the screen shows a male character in an indoor setting. At the beginning of the video, his eyes are wide open and his mouth is also open, indicating a surprised facial expression. In the following scenes, he looks around, seemingly explaining or narrating something to the people around him. Overall, his emotions are not positive or optimistic. In the audio, the character speaks with a stutter, which usually expresses feelings of nervousness, anxiety, or unease. Combined with the text content, the character seems to be unhappy and angry due to the prejudice of the people around him. The subtitle in the text says, "Why are you all looking at me like that? So, as long as it's a woman, does she have to have a relationship with me?" This sentence expresses the male character's dissatisfaction and anger towards the people around him. Based on the surprised and negative facial expression of the male character in the video clues, as well as the stuttering speech in the audio clues, we can infer that the male character is expressing a feeling of dissatisfaction and anger in this sentence. He may feel troubled by the prejudice of the people around him and is unhappy with this unfair treatment.

**CLUE-Audio:** In the audio, the character speaks with a stutter, which usually expresses feelings of nervousness, anxiety, or unease. Combined with the text content, the character seems to be unhappy and angry due to the prejudice of the people around him. In the text, the subtitle content: "Why are you all looking at me like that? So, as long as it's a woman, does she have to have a relationship with me?" This sentence expresses the character's dissatisfaction and anger. The stuttering described in the audio clues, which is usually associated with emotions like nervousness, anxiety, or unease, combined with the subtitle content suggests that the character feels dissatisfied and angry. The character feels dissatisfied with the prejudices of those around them and believes that they have certain expectations or demands of them. Therefore, combining the audio clues and the subtitle content, we can infer that the character's emotional state is one of nervousness, anxiety, unease, dissatisfaction, and anger.

**CLUE-Text:** Subtitle: "Why are you all looking at me like that? So, as long as it's a woman, does she have to have a relationship with me?" From this piece of text, we can infer that the character's emotional state is one of dissatisfaction or anger. The reasoning is as follows: 1. "Why are you all looking at me like that?" This question expresses the character's dissatisfaction with being stared at. It implies that the character feels uncomfortable or displeased with others watching him. 2. "So, as long as it's a woman, does she have to have a relationship with me?" This statement conveys the character's frustration with others' expectations or prejudices. The character feels annoyed or angry about the assumption that there should be a special relationship between him and a woman simply because she is female. In summary, the text suggests that the character is experiencing emotions of dissatisfaction or anger.

**CLUE-Video:** In the video, the screen shows a male character in an indoor setting. At the beginning of the video, his eyes are wide open and his mouth is also open, indicating a surprised facial expression. In the following scenes, he looks around, seemingly explaining or narrating something to the people around him. Overall, his emotions are not positive or optimistic.

Figure 20: Visualization of CLUE-M/A/T/V.

# N  DETAILS OF CLUE-MLLM

CLUE-MLLM directly utilizes the output from MLLM without any manual checking process. Table 10 provides model cards for different MLLMs. For each MLLM, we provide two types of prompts (see Table 11): one that ignores text and another that considers text. To ensure a fair comparison, we use similar prompts for audio, video, and audio-video LLMs.

Table 10: Model cards for MLLMs.

| Model | Link |
|---|---|
| SECap (Xu et al., 2024) | https://github.com/thuhcsi/SECap |
| SALMONN (Tang et al., 2023) | https://github.com/bytedance/SALMONN |
| Qwen-Audio (Chu et al., 2023) | https://github.com/QwenLM/Qwen-Audio |
| Otter (Li et al., 2023a) | https://github.com/Luodian/Otter |
| OneLLM (Han et al., 2023) | https://github.com/csuhan/OneLLM |
| PandaGPT (Su et al., 2023) | https://github.com/yxuansu/PandaGPT |
| VideoChat (Li et al., 2023b) | https://github.com/OpenGVLab/Ask-Anything/tree/main/video_chat |
| VideoChat2 (Li et al., 2024) | https://github.com/OpenGVLab/Ask-Anything/tree/main/video_chat2 |
| Video-LLaMA (Zhang et al., 2023) | https://github.com/DAMO-NLP-SG/Video-LLaMA |
| Video-LLaVA (Lin et al., 2023) | https://github.com/PKU-YuanGroup/Video-LLaVA |
| Video-ChatGPT (Maaz et al., 2024) | https://github.com/mbzuai-oryx/Video-ChatGPT |
| LLaMA-VID (Li et al., 2023c) | https://github.com/dvlab-research/LLaMA-VID |
| mPLUG-Owl (Ye et al., 2023) | https://github.com/X-PLUG/mPLUG-Owl |
| Chat-UniVi (Jin et al., 2023) | https://github.com/PKU-YuanGroup/Chat-UniVi |
| GPT-4V (OpenAI, 2023) | https://openai.com/ |

Table 11: Prompts for extracting emotion-related descriptions using MLLMs.

| Model | Text | Prompt |
|---|---|---|
| Audio LLM | w/o | As an expert in the field of emotions, please focus on the acoustic information in the audio to discern clues related to the emotions of the individual. Please provide a detailed description and ultimately predict the emotional state of the individual. |
| | w/ | Subtitle content of the audio: {subtitle}; As an expert in the field of emotions, please focus on the acoustic information and subtitle content in the audio to discern clues related to the emotions of the individual. Please provide a detailed description and ultimately predict the emotional state of the individual in the audio. |
| Video LLM | w/o | As an expert in the field of emotions, please focus on the facial expressions, body movements, environment, etc., in the video to discern clues related to the emotions of the individual. Please provide a detailed description and ultimately predict the emotional state of the individual in the video. |
| | w/ | Subtitle content of the video: {subtitle}; As an expert in the field of emotions, please focus on the facial expressions, body movements, environment, subtitle content, etc., in the video to discern clues related to the emotions of the individual. Please provide a detailed description and ultimately predict the emotional state of the individual. |
| Audio-Video LLM | w/o | As an expert in the field of emotions, please focus on the facial expressions, body movements, environment, acoustic information, etc., in the video to discern clues related to the emotions of the individual. Please provide a detailed description and ultimately predict the emotional state of the individual in the video. |
| | w/ | Subtitle content of the video: {subtitle}; As an expert in the field of emotions, please focus on the facial expressions, body movements, environment, acoustic information, subtitle content, etc., in the video to discern clues related to the emotions of the individual. Please provide a detailed description and ultimately predict the emotional state of the individual in the video. |

This paper tests different CLUE-MLLM generation strategies: S0, S1, and S2. Experimental results are shown in Table 12. We observe that S2 generally outperforms both S0 and S1. Therefore, we adopt S2 as the default strategy.

Table 12: Performance comparison of different strategies for generating CLUE-MLLM.

| Model | Strategy | English | | | Chinese | | |
|---|---|---|---|---|---|---|---|
| | | $F_S$ | Precision$_S$ | Recall$_S$ | $F_S$ | Precision$_S$ | Recall$_S$ |
| Otter | S0 | $34.75_{\pm0.02}$ | $40.41_{\pm0.03}$ | $30.48_{\pm0.01}$ | $31.08_{\pm0.10}$ | $35.71_{\pm0.15}$ | $27.51_{\pm0.07}$ |
| | S1 | $22.54_{\pm0.05}$ | $26.05_{\pm0.08}$ | $19.86_{\pm0.04}$ | $25.06_{\pm0.04}$ | $29.14_{\pm0.03}$ | $21.99_{\pm0.05}$ |
| | S2 | $43.51_{\pm0.09}$ | $50.71_{\pm0.10}$ | $38.09_{\pm0.09}$ | $46.22_{\pm0.01}$ | $52.65_{\pm0.16}$ | $41.18_{\pm0.08}$ |
| PandaGPT | S0 | $26.99_{\pm0.01}$ | $29.18_{\pm0.08}$ | $25.10_{\pm0.04}$ | $28.70_{\pm0.01}$ | $30.95_{\pm0.00}$ | $26.76_{\pm0.03}$ |
| | S1 | $34.75_{\pm0.21}$ | $36.77_{\pm0.30}$ | $32.94_{\pm0.14}$ | $34.74_{\pm0.17}$ | $37.27_{\pm0.15}$ | $32.53_{\pm0.18}$ |
| | S2 | $45.89_{\pm0.20}$ | $50.03_{\pm0.01}$ | $42.38_{\pm0.33}$ | $47.33_{\pm0.04}$ | $53.01_{\pm0.08}$ | $42.75_{\pm0.11}$ |
| Video-ChatGPT | S0 | $34.77_{\pm0.04}$ | $37.66_{\pm0.13}$ | $32.30_{\pm0.03}$ | $37.62_{\pm0.16}$ | $40.33_{\pm0.05}$ | $35.25_{\pm0.25}$ |
| | S1 | $41.74_{\pm0.24}$ | $45.59_{\pm0.24}$ | $38.49_{\pm0.23}$ | $40.81_{\pm0.03}$ | $45.07_{\pm0.00}$ | $37.28_{\pm0.05}$ |
| | S2 | $50.52_{\pm0.06}$ | $54.03_{\pm0.04}$ | $47.44_{\pm0.07}$ | $54.73_{\pm0.00}$ | $61.15_{\pm0.10}$ | $49.52_{\pm0.06}$ |
| Video-LLaMA | S0 | $28.17_{\pm0.26}$ | $28.64_{\pm0.36}$ | $27.72_{\pm0.18}$ | $30.70_{\pm0.11}$ | $30.09_{\pm0.14}$ | $31.34_{\pm0.08}$ |
| | S1 | $34.43_{\pm0.16}$ | $35.82_{\pm0.20}$ | $33.15_{\pm0.11}$ | $34.01_{\pm0.25}$ | $35.16_{\pm0.22}$ | $32.94_{\pm0.26}$ |
| | S2 | $44.73_{\pm0.14}$ | $44.14_{\pm0.13}$ | $45.34_{\pm0.15}$ | $47.26_{\pm0.03}$ | $47.98_{\pm0.07}$ | $46.56_{\pm0.01}$ |
| VideoChat | S0 | $31.95_{\pm0.02}$ | $31.73_{\pm0.13}$ | $32.17_{\pm0.10}$ | $34.53_{\pm0.02}$ | $33.53_{\pm0.01}$ | $35.60_{\pm0.05}$ |
| | S1 | $45.10_{\pm0.07}$ | $46.24_{\pm0.05}$ | $44.01_{\pm0.10}$ | $44.25_{\pm0.09}$ | $44.76_{\pm0.02}$ | $43.75_{\pm0.16}$ |
| | S2 | $45.53_{\pm0.11}$ | $42.90_{\pm0.27}$ | $48.49_{\pm0.10}$ | $45.57_{\pm0.03}$ | $47.20_{\pm0.12}$ | $44.05_{\pm0.05}$ |
| VideoChat2 | S0 | $35.70_{\pm0.06}$ | $43.08_{\pm0.00}$ | $30.47_{\pm0.09}$ | $35.27_{\pm0.01}$ | $41.16_{\pm0.00}$ | $30.86_{\pm0.01}$ |
| | S1 | $37.56_{\pm0.07}$ | $44.62_{\pm0.00}$ | $32.43_{\pm0.10}$ | $38.71_{\pm0.10}$ | $45.14_{\pm0.13}$ | $33.88_{\pm0.08}$ |
| | S2 | $49.07_{\pm0.26}$ | $54.72_{\pm0.41}$ | $44.47_{\pm0.15}$ | $48.86_{\pm0.05}$ | $57.12_{\pm0.08}$ | $42.68_{\pm0.04}$ |
| mPLUG-Owl | S0 | $39.21_{\pm0.14}$ | $40.56_{\pm0.15}$ | $37.94_{\pm0.12}$ | $40.53_{\pm0.33}$ | $40.44_{\pm0.24}$ | $40.62_{\pm0.43}$ |
| | S1 | $45.80_{\pm0.06}$ | $47.49_{\pm0.04}$ | $44.22_{\pm0.07}$ | $47.97_{\pm0.04}$ | $49.33_{\pm0.03}$ | $46.69_{\pm0.05}$ |
| | S2 | $52.73_{\pm0.13}$ | $54.54_{\pm0.13}$ | $51.04_{\pm0.13}$ | $50.95_{\pm0.06}$ | $56.40_{\pm0.11}$ | $46.47_{\pm0.18}$ |
| SALMONN | S0 | $40.71_{\pm0.10}$ | $41.38_{\pm0.25}$ | $40.07_{\pm0.04}$ | $43.45_{\pm0.23}$ | $43.24_{\pm0.30}$ | $43.66_{\pm0.16}$ |
| | S1 | $39.79_{\pm0.03}$ | $39.54_{\pm0.01}$ | $40.05_{\pm0.06}$ | $41.43_{\pm0.13}$ | $41.11_{\pm0.03}$ | $41.76_{\pm0.22}$ |
| | S2 | $47.96_{\pm0.04}$ | $50.20_{\pm0.04}$ | $45.92_{\pm0.04}$ | $48.24_{\pm0.03}$ | $52.24_{\pm0.00}$ | $44.82_{\pm0.05}$ |
| Qwen-Audio | S0 | $30.64_{\pm0.06}$ | $41.92_{\pm0.00}$ | $24.14_{\pm0.08}$ | $30.50_{\pm0.05}$ | $40.84_{\pm0.13}$ | $24.33_{\pm0.03}$ |
| | S1 | $35.23_{\pm0.10}$ | $46.69_{\pm0.15}$ | $28.29_{\pm0.08}$ | $44.09_{\pm0.00}$ | $58.08_{\pm0.00}$ | $35.53_{\pm0.00}$ |
| | S2 | $38.13_{\pm0.05}$ | $49.42_{\pm0.18}$ | $31.04_{\pm0.00}$ | $41.14_{\pm0.07}$ | $53.71_{\pm0.00}$ | $33.34_{\pm0.09}$ |
| Video-LLaVA | S0 | $32.64_{\pm0.03}$ | $33.31_{\pm0.01}$ | $32.00_{\pm0.05}$ | $32.76_{\pm0.03}$ | $33.19_{\pm0.06}$ | $32.33_{\pm0.00}$ |
| | S1 | $30.19_{\pm0.02}$ | $34.10_{\pm0.03}$ | $27.08_{\pm0.05}$ | $31.93_{\pm0.11}$ | $33.40_{\pm0.19}$ | $30.58_{\pm0.04}$ |
| | S2 | $47.07_{\pm0.16}$ | $48.58_{\pm0.02}$ | $45.66_{\pm0.29}$ | $49.21_{\pm0.06}$ | $53.95_{\pm0.03}$ | $45.23_{\pm0.13}$ |
| LLaMA-VID | S0 | $35.14_{\pm0.14}$ | $36.71_{\pm0.15}$ | $33.69_{\pm0.14}$ | $33.30_{\pm0.04}$ | $33.12_{\pm0.06}$ | $33.48_{\pm0.03}$ |
| | S1 | $42.37_{\pm0.03}$ | $43.97_{\pm0.04}$ | $40.89_{\pm0.03}$ | $42.56_{\pm0.08}$ | $43.28_{\pm0.11}$ | $41.86_{\pm0.04}$ |
| | S2 | $51.25_{\pm0.09}$ | $52.71_{\pm0.18}$ | $49.87_{\pm0.00}$ | $52.01_{\pm0.02}$ | $57.30_{\pm0.00}$ | $47.61_{\pm0.03}$ |
| Chat-UniVi | S0 | $39.89_{\pm0.18}$ | $42.32_{\pm0.21}$ | $37.72_{\pm0.15}$ | $36.83_{\pm0.30}$ | $37.74_{\pm0.27}$ | $35.96_{\pm0.33}$ |
| | S1 | $47.94_{\pm0.19}$ | $50.96_{\pm0.20}$ | $45.26_{\pm0.18}$ | $47.02_{\pm0.00}$ | $48.07_{\pm0.00}$ | $46.01_{\pm0.00}$ |
| | S2 | $53.08_{\pm0.01}$ | $53.68_{\pm0.00}$ | $52.50_{\pm0.02}$ | $53.86_{\pm0.02}$ | $58.54_{\pm0.01}$ | $49.86_{\pm0.03}$ |

## O    RELATIONSHIP BETWEEN DESCRIPTION LENGTH AND LABEL NUMBERS

This section further discusses the relationship between description length and the number of labels per sample, i.e., whether longer descriptions correlate with more labels. To this end, we compute their PCC scores. We observe that, for the human-only strategy, the PCC score is 0.3416, and for the human-LLM collaboration strategy, the PCC score is 0.2939. Therefore, although from the dataset level, the length of descriptions is related to the richness of labels (see Figure 5), these two metrics do not show a strong correlation at the sample level.

## P    COST OF GPT-BASED METRICS

This paper reports zero-shot performance, only focusing on the inference process. The cost of evaluating our OV-MERD dataset is about $1 per evaluation, which may not seem high. However, for future work aimed at training frameworks to better address the OV-MER task, this cost will become prohibitive. For example, if we plan to train a model for 100 epochs, the evaluation cost will rise to $1 \times 100$ epochs = \$100. If we intend to test $N$ different parameter combinations and $M$ different frameworks, the evaluation cost will further increase to $\$100 \times M \times N$. Moreover, we

plan to expand the OV-MERD dataset in the future. This cost will further increase. Therefore, this paper explores alternatives to GPT-based metrics.

## Q  GPT-BASED VS. MATCHING-BASED METRICS

Table 13 provides raw scores for GPT- and matching-based metrics. See Section 5 for more analysis.

Table 13: GPT-based vs. matching-based metrics. "$P_S$", "$R_S$", "$B_1$", "$B_4$", "'M', and "$R_l$" are abbreviations for Precision$_S$, Recall$_S$, BLEU$_1$, BLEU$_4$, METEOR, and ROUGE$_l$, respectively.

| MLLM | L | V | A | English | | | | | | | Chinese | | | | | | |
| | | | | GPT-based | | | Matching-based | | | | GPT-based | | | Matching-based | | | |
| | | | | $F_S$ | $P_S$ | $R_S$ | $B_1$ | $B_4$ | M | $R_l$ | $F_S$ | $P_S$ | $R_S$ | $B_1$ | $B_4$ | M | $R_l$ |
|---|---|---|---|---|---|---|---|---|---|---|---|---|---|---|---|---|---|
| Qwen-Audio | √ | × | √ | 38.13 | 49.42 | 31.04 | 21.87 | 06.55 | 21.65 | 20.81 | 41.14 | 53.71 | 33.34 | 27.64 | 12.07 | 26.09 | 25.24 |
| OneLLM | √ | × | √ | 42.84 | 45.92 | 40.15 | 33.81 | 08.54 | 28.00 | 22.46 | 46.17 | 52.07 | 41.47 | 42.75 | 16.60 | 34.42 | 26.81 |
| Otter | √ | √ | × | 43.51 | 50.71 | 38.09 | 27.26 | 07.55 | 23.42 | 21.05 | 46.22 | 52.65 | 41.18 | 35.35 | 14.41 | 29.34 | 25.91 |
| Video-LLaMA | √ | √ | × | 44.73 | 44.14 | 45.34 | 28.76 | 06.41 | 31.22 | 20.41 | 47.26 | 47.98 | 46.56 | 34.88 | 12.13 | 37.61 | 24.25 |
| VideoChat | √ | √ | × | 45.53 | 42.90 | 48.49 | 26.44 | 05.41 | 30.58 | 19.11 | 45.57 | 47.20 | 44.05 | 31.36 | 10.86 | 37.48 | 22.57 |
| PandaGPT | √ | √ | √ | 45.89 | 50.03 | 42.38 | 33.69 | 07.64 | 30.29 | 22.07 | 47.33 | 53.01 | 42.75 | 43.02 | 15.83 | 37.94 | 26.87 |
| Video-LLaVA | √ | √ | × | 47.07 | 48.58 | 45.66 | 33.48 | 08.25 | 29.68 | 22.34 | 49.21 | 53.95 | 45.23 | 42.72 | 15.97 | 36.87 | 26.90 |
| SALMONN | √ | × | √ | 47.96 | 50.20 | 45.92 | 31.89 | 07.19 | 28.42 | 20.99 | 48.24 | 52.24 | 44.82 | 39.00 | 14.00 | 35.12 | 25.35 |
| VideoChat2 | √ | √ | × | 49.07 | 54.72 | 44.47 | 31.60 | 08.10 | 26.61 | 21.65 | 48.86 | 57.12 | 42.68 | 41.18 | 16.15 | 33.54 | 26.80 |
| Video-ChatGPT | √ | √ | × | 50.52 | 54.03 | 47.44 | 32.64 | 07.65 | 30.25 | 22.01 | 54.73 | 61.15 | 49.52 | 41.96 | 15.50 | 38.18 | 26.35 |
| OneLLM | √ | √ | × | 50.52 | 55.93 | 46.06 | 32.19 | 08.10 | 28.44 | 22.25 | 51.44 | 56.43 | 47.26 | 41.31 | 15.15 | 35.15 | 25.98 |
| LLaMA-VID | √ | √ | × | 51.25 | 52.71 | 49.87 | 33.81 | 08.26 | 30.31 | 22.36 | 52.01 | 57.30 | 47.61 | 43.01 | 16.23 | 37.92 | 27.20 |
| mPLUG-Owl | √ | √ | × | 52.73 | 54.54 | 51.04 | 33.04 | 07.75 | 30.24 | 21.75 | 50.95 | 56.40 | 46.47 | 41.69 | 15.16 | 37.81 | 26.39 |
| Chat-UniVi | √ | √ | × | 53.08 | 53.68 | 52.50 | 32.80 | 07.83 | 31.12 | 22.15 | 53.86 | 58.54 | 49.86 | 40.76 | 15.05 | 38.75 | 26.43 |
| GPT-4V | √ | √ | × | 55.51 | 48.52 | 64.86 | 39.40 | 18.41 | 43.67 | 32.60 | 57.21 | 54.61 | 60.07 | 45.45 | 29.08 | 53.76 | 40.37 |

In Table 13, we observe that there is no strong correlation between the GPT-based metrics and the matching-based metrics. To clarify this point, we use the following three sentences as examples:

#1. The clue is "the weather is great". His emotion is "happy".

#2. The clue is "the weather is bad". His emotion is "sad".

#3. His emotion is "happy".

For matching-based metrics, we use BLEU$_1$ as an example. The BLEU$_1$ score between #1 and #2 is 0.8181, while the BLEU$_1$ score between #1 and #3 is 0.1738. Therefore, based on the BLEU$_1$ score, #1 is closer to #2. For LLM-based metrics, we first extract the emotion labels and compare their similarity, so #1 is closer to #3. This demonstrates that matching-based metrics are not suitable for evaluating emotion recognition performance.

## R    EMOTION WHEEL

The emotion wheel provides psychologically based emotion grouping information. In this paper, we select five representative emotion wheels (W1∼W5) and use their grouping information for metric calculation. See Figure 21 for more details.

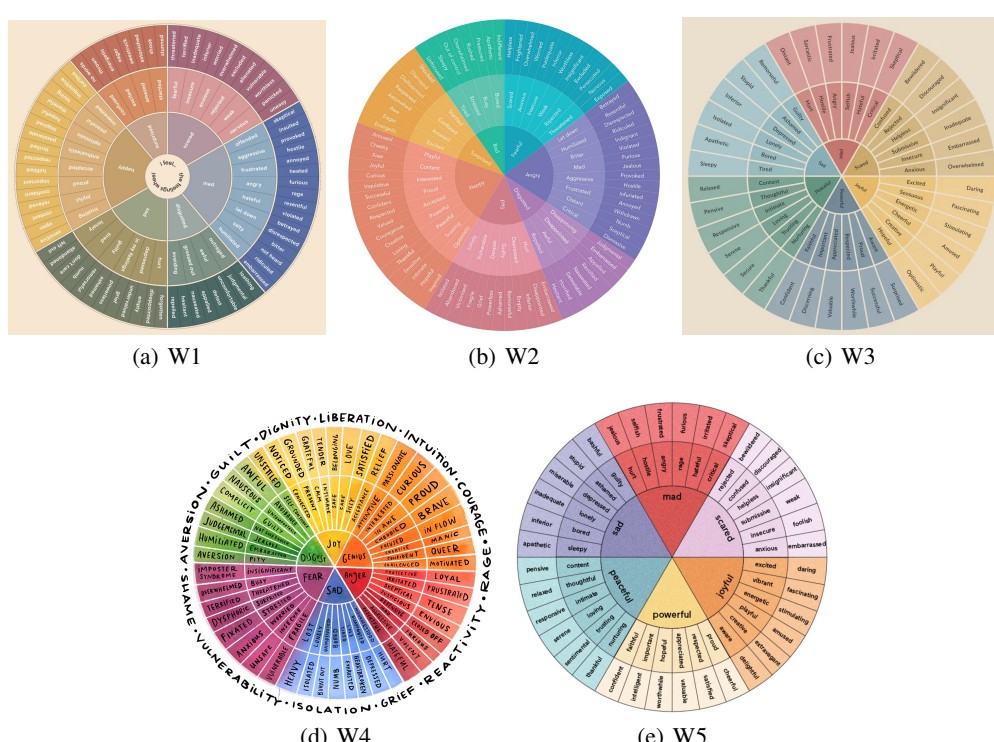

(a) W1                    (b) W2                    (c) W3

(d) W4                    (e) W5

Figure 21: Emotion wheels. This paper selects five representative emotion wheels (please zoom in to clearly view the emotional hierarchy): (a) W1 (b) W2 (c) W3 (d) W4 (e) W5

