# OpenReview forum: "Open-vocabulary Multimodal Emotion Recognition: Dataset, Metric, and Benchmark"
_ICLR.cc/2025/Conference — Submitted to ICLR 2025_

### Official Review · Reviewer_GTfB · 2024-10-17

**Soundness:** 3
**Presentation:** 3
**Contribution:** 3
**Rating:** 6
**Confidence:** 3

**Summary:**

The paper introduces a new task in the realm of MER, termed Open-vocabulary MER (OV-MER), which contains a broader and more diverse range of emotion labels. The authors propose an LLM-based framework to generate labels for OV-MER, along with corresponding metrics and benchmarks for evaluation.

**Strengths:**

1. Extensive experiments with plenty of SOTA LLM methods are conducted.
2. The visualizations are clear and straightforward, making the content easy to follow.
3. The idea of extending MER to OV-MER is novel and adds a fresh perspective to the field.

**Weaknesses:**

1. The dataset scale is not mentioned or compared in Sec 2 or Tab 1.
2. There is no quality evaluation for the generated labels (e.g., comparison with manual annotations), and their validity has not been assessed (e.g., by using the dataset for pre-training or other downstream tasks). Although there are comparisons with humans regarding clue length, label count, and word cloud, the label accuracy is never evaluated or discussed. This raises concerns about the reliability and overall usefulness of the dataset for the study.
3. The setup in Table 2 and 3 presents a data leakage issue. The ground truth label is generated using SALMONN and GPT-4V, but both of these methods are also being evaluated.

I am willing to raise my score if the authors could address my second concern.

**Questions:**

1. Sec 2.1, pre-annotation, what is the average duration of the videos? I'm concerned that using only three frames as input may miss a significant amount of visual info, especially since the paper later notes that there are errors, duplicates, and missing info related to visual clues.
2. Sec 2.1, manual check, could you please clarify why ALLM struggles to capture emotion-related acoustic features? From my understanding, paralinguistic information in the audio modality contains meaningful emotion-related features, which should be useful in this context.
3. Table 1, CMU-MOSEI has discrete emotion labels.
4. In Sec 3.1, the GPT-based grouping is described as costly, but if the entire process only costs $1, it doesn't seem to be a significant expense. Could you clarify the reason?
5. Fig 4 (a), typo in video branch: ALLM -> VLLM.
6. Sec 4.1, CLUE-A, could you explain why text is used to generate acoustic clues when the audio already contains the necessary content information? It seems redundant.
7. It is not clear what fusion method you use in Tab 3.

---

> ### Author Response · Authors · 2024-11-20
> **Response to Review GTfB (Part1)**
>
> Thanks for recognizing the novelty of our ideas and for your positive feedback on our extensive experiments and writing. While there are some minor issues with the expression, we have made every effort to clarify them. We believe this work is an important extension of the traditional MER task and will contribute to the development of this field.
>
> **Q1:**
> The dataset scale is not mentioned or compared in Sec 2 or Tab 1.
>
> **A1:**
> Our benchmark dataset contains 1,121 emotional labels and 332 samples, all of which are used to establish a zero-shot learning benchmark for the OV-MER task. The main contribution of this work is the definition of a new task and the development of foundational research, including the datasets, metrics, and solutions. In addition to this dataset, we are also working on a series of follow-up efforts to expand the dataset.
>
>
>
> **Q2:**
> There is no quality evaluation for the generated labels (e.g., comparison with manual annotations), and their validity has not been assessed (e.g., by using the dataset for pre-training or other downstream tasks). Although there are comparisons with humans regarding clue length, label count, and word cloud, the label accuracy is never evaluated or discussed. This raises concerns about the reliability and overall usefulness of the dataset for the study.
>
> **A2:** As shown in Figure 2, our ground truth generation process has gone through multiple rounds of strict manual checks by professional annotators. **Therefore, the ground-truth labels are accurate and reliable.**
>
> In Appendix F, we provide a deeper comparison between the one-hot labels in the MER2023 dataset and the OV labels in the OV-MERD dataset. We observe that OV labels cover a wider variety of emotions, some of which (such as *shy*, *nervous*, and *grateful*) are rarely discussed in previous datasets. Meanwhile, we notice that most samples have about 2 to 4 labels, much more than the traditional task where each sample is assigned only one emotion. Therefore, our dataset provides richer and more comprehensive labels for each sample.
>
>
>
> **Q3:**
> The setup in Table 2 and 3 presents a data leakage issue. The ground truth label is generated using SALMONN and GPT-4V, but both of these methods are also being evaluated.
>
> **A3:**
> SALMONN and GPT-4V were used only to generate initial descriptions. Subsequently, we conducted manual checks independent of SALMONN and GPT-4V, and the final descriptions differed significantly from the initial ones. Therefore, there is no issue of data leakage. Tables 2 and 3 list various MLLMs, and the primary reason for including SALMONN and GPT-4V in these tables is to provide a comprehensive performance comparison.
>
>
> **Q4:**
> Sec 2.1, pre-annotation, what is the average duration of the videos? I'm concerned that using only three frames as input may miss a significant amount of visual info, especially since the paper later notes that there are errors, duplicates, and missing info related to visual clues.
>
> **A4:**
> Thanks for your comments. In Appendix H, we analyze the duration distribution of the OV-MERD dataset. We observe that the majority of the samples have a duration between 1 and 4 seconds.
>
> Regarding the sampling strategy, we did not provide a detailed explanation due to space limitations. Specifically, we categorize visual clues into two types: **(1) visual clues with relatively long durations**; and **(2) visual clues with fast movements, such as eye movements, head movements, and micro-expressions.**
>
> **For the first type**, since the duration of most videos is between 1 and 4 seconds (see Appendix H) and the video content is usually continuous with only minor differences between adjacent frames (see Appendix C), uniformly sampling three frames are sufficient to capture this information; **For the second type**, we observe that current MLLMs (including the GPT-4V used in this paper) struggle to capture these fast movements. Increasing the number of sampled frames does not address this issue. Previous research has also shown that GPT-4V cannot recognize micro-expressions [1]. To capture these fast movements, we employ multiple professional annotators to manually add this information.
>
>
> [1] Lu, Hao, Xuesong Niu, Jiyao Wang, Yin Wang, Qingyong Hu, Jiaqi Tang, Yuting Zhang et al. "Gpt as psychologist? preliminary evaluations for gpt-4v on visual affective computing." In Proceedings of the IEEE/CVF Conference on Computer Vision and Pattern Recognition, pp. 322-331. 2024.

---

> ### Author Response · Authors · 2024-11-20
> **Response to Review GTfB (Part2)**
>
> **Q5:**
> Sec 2.1, manual check, could you please clarify why ALLM struggles to capture emotion-related acoustic features? From my understanding, paralinguistic information in the audio modality contains meaningful emotion-related features, which should be useful in this context.
>
> **A5:**
> The paralinguistic information in the audio contains meaningful emotion-related features. However, current ALLMs struggle to capture these emotion-related paralinguistic features. We believe the main reason is that existing ALLMs primarily focus on datasets related to tasks like ASR or audio event detection, with less emphasis on paralinguistic information, which results in relatively poor performance in paralinguistic understanding.
>
> **Q6:**
> Table 1, CMU-MOSEI has discrete emotion labels.
>
> **A6:**
> Thanks for your comments. We correct this issue in the revised PDF. More dataset comparison is provided in Appendix G.
>
> **Q7:**
> In Sec 3.1, the GPT-based grouping is described as costly, but if the entire process only costs \$1, it doesn't seem to be a significant expense. Could you clarify the reason?
>
> **A7:**
> Thanks for pointing it out. This paper reports zero-shot performance, only focusing on the inference process. The cost of evaluating our OV-MERD dataset is about \$1 per evaluation, which may not seem high.
>
> However, for future work aimed at training frameworks to better address the OV-MER task, this cost will become prohibitive. For example, if we plan to train a model for 100 epochs, the evaluation cost will rise to  \$100.
>
> If we intend to test N different parameter combinations and M different frameworks, the evaluation cost will further increase to \$100 * M *  N. Moreover, we plan to expand the OV-MERD dataset in the future. This cost will further increase. Therefore, this paper explores alternatives to GPT-based metrics. We place this discussion in the appendix.
>
>
> **Q8:**
> Fig 4 (a), typo in video branch: ALLM $\rightarrow$ VLLM.
>
> **A8:**
> Thanks for your careful reading. We correct it in our revised PDF.
>
> **Q9:**
> Sec 4.1, CLUE-A, could you explain why text is used to generate acoustic clues when the audio already contains the necessary content information? It seems redundant.
>
> **A9:**
> Thanks for your comment. In Section 5, we discussed this issue and highlighted the impact of different CLUE-MLLM generation strategies: 1) *S0* does not use text and inputs the audio into MLLM; 2) *S1* inputs both text and audio into MLLM simultaneously; 3) *S2* first uses MLLM to extract descriptions and then combines with text using another LLM. In general, \textbf{S2} achieves the best performance. **These results suggest that the ALLM cannot fully leverage the text, and using an additional LLM to emphasize the text can further improve performance. Based on these observations, we adopt this strategy for generating CLUE-Audio.**
>
> **Q10:**
> It is not clear what fusion method you use in Tab 3.
>
> **A10:**
> Thanks for your careful reading. To fuse different modalities, we input prelabeled acoustic and visual clues into LLM and leverage its reasoning capabilities for multimodal integration (see Figure 4). This approach is consistent with the fusion method used during our dataset construction process (see Section 2). We add this to our revised paper.

---

> > ### Comment · Reviewer_GTfB · 2024-11-24
> >
> > Thanks authors for the detailed response and I appreciate your hard work. Most of my concerns are well-addressed in the rebuttal. I raise my score from 5 to 6.

---

> > > ### Author Response · Authors · 2024-11-25
> > > **Appreciation to Reviewer GTfB**
> > >
> > > We sincerely appreciate your support.

---

### Official Review · Reviewer_sjuE · 2024-10-21

**Soundness:** 2
**Presentation:** 3
**Contribution:** 2
**Rating:** 5
**Confidence:** 5

**Summary:**

This paper introduces a novel paradigm in multimodal emotion recognition called Open-vocabulary Multimodal Emotion Recognition (OV-MER), which aims to address the limitations of traditional methods that rely on a fixed set of emotion labels. The OV-MER aims to better capture the subtle variations in human emotions by allowing predictions across a broader range of emotion categories. To support this method, the authors developed a new dataset named OV-MERD, using a collaborative annotation strategy between humans and large language models (LLMs) to generate detailed emotion-related descriptions (CLUE), which enrich the variety and depth of emotion labels. The paper defines new evaluation metrics based on emotional similarity grouping and conducts extensive experiments with various baseline models to highlight the challenges and limitations of this task. The study also emphasizes the advantages of human-LLM collaboration in generating more comprehensive emotion descriptions, as opposed to human-only annotations.

**Strengths:**

1. The paper introduces a novel paradigm, which extends beyond traditional methods that rely on a fixed set of emotion labels.
2. The development of the OV-MERD dataset, which includes a wide range of emotion categories, is a significant contribution. Its rich diversity allows models to learn from a broader spectrum of emotional expressions, which may be used to improve the generalizability of emotion recognition systems.
3. The paper conducts comprehensive experiments and benchmarks, demonstrating the effectiveness of human-LLM collaboration-based  CULE-Multi.

**Weaknesses:**

1. The baseline results rely on CLUE-Multi/A/V/T descriptions that include human checks. In real-world scenarios, the human intervention is impractical and the actual performance may be lower.
2. The dataset’s cartoonish style could hinder the model’s ability to generalize to real-world data, potentially limiting its effectiveness in practical applications.
3. There is a lack of information on the total number of samples and the train/validation/test split.

**Questions:**

1. The models in the paper are trained using CLUE-Multi/A/V/T data that includes manual checks, which ensures high-quality and detailed emotion descriptions. However, in real-world applications, it is likely that only LLMs-generated CLUEs without any human oversight will be available. Would this discrepancy lead to significant performance degradation in models trained using CLUE-Multi/A/V/T data?
2. Would the cartoonish style of the dataset cause a significant domain shift problem? Specifically, how would a model pre-trained on this dataset perform when tested or fine-tuned on other datasets like DFEW and IEMOCAP? Do you consider any adaptation strategies to mitigate potential domain shifts?
3. What is the sample size of the dataset, and how are the train/validation/test sets divided?
4. The paper mentioned, "This is because, in MER, text is emotionally ambiguous; without integrating other modalities, we cannot accurately understand the emotions conveyed through text." However, as far as I know, there are datasets like MOSI and MOSEI where the text/language modality contributes more than other modalities, as evidenced by relevant experimental results [1-4]. Is this explanation of yours appropriate, or is it specific to this particular dataset?

Reference

[1] Hazarika, Devamanyu, Roger Zimmermann, and Soujanya Poria. "Misa: Modality-invariant and-specific representations for multimodal sentiment analysis." in ACM MM 20.

[2] Yu, Wenmeng, et al. "Learning modality-specific representations with self-supervised multi-task learning for multimodal sentiment analysis." in AAAI 2021.

[3] Zhang, Haoyu, et al. "Learning Language-guided Adaptive Hyper-modality Representation for Multimodal Sentiment Analysis." in EMNLP 2023.

[4] Feng, Xinyu, et al. "Knowledge-Guided Dynamic Modality Attention Fusion Framework for Multimodal Sentiment Analysis." in EMNLP 2024.

---

> ### Author Response · Authors · 2024-11-20
> **Response to Reviewer sjuE**
>
> We greatly appreciate your positive comment on the task and dataset we proposed, as well as your encouraging evaluation of our experiment. While there are some minor issues with the expression, we have made every effort to clarify them. We believe this work is an important extension of the traditional MER task and will contribute to the development of this field.
>
> **Q1:**
> The baseline results rely on CLUE-Multi/A/V/T descriptions that include human checks. In real-world scenarios, the human intervention is impractical and the actual performance may be lower.
>
> **A1:**
> You may misunderstand the purpose of these baselines. **The results on CLUE-Multi/A/V/T only reflect the performance upper bound, which is used to evaluate whether current MLLMs can address the OV-MER task.**
>
>
> **Q2:**
> The models in the paper are trained using CLUE-Multi/A/V/T data that includes manual checks, which ensures high-quality and detailed emotion descriptions. However, in real-world applications, it is likely that only LLMs-generated CLUEs without any human oversight will be available. Would this discrepancy lead to significant performance degradation in models trained using CLUE-Multi/A/V/T data?
>
> **A2:**
> It seems there have been some misunderstandings. **The main goal of this paper is to provide a valuable dataset with rich emotional labels to further advance the development of MER.**
>
>
> **Q3:**
> The dataset’s cartoonish style could hinder the model’s ability to generalize to real-world data, potentially limiting its effectiveness in practical applications. Would the cartoonish style of the dataset cause a significant domain shift problem? Specifically, how would a model pre-trained on this dataset perform when tested or fine-tuned on other datasets like DFEW and IEMOCAP? Do you consider any adaptation strategies to mitigate potential domain shifts?
>
> **A3:**
> **Our dataset is not in a cartoon style.** The original data comes from MER2023, which is a dataset containing real-person videos collected from movies and TV series. The cartoon-style videos mentioned in the paper were generated by using DemoAI to remove personal information, addressing copyright concerns. For the original data, please download the MER2023 dataset.
>
>
> **Q4:**
> There is a lack of information on the total number of samples and the train/validation/test split. What is the sample size of the dataset, and how are the train/validation/test sets divided?
>
> **A4:**
> This paper provides a zero-shot learning benchmark that does not require the training process. Our benchmark dataset contains 1,121 emotional labels and 332 samples, all of which are used to test the performance of various MLLMs. The main contribution of this work is the definition of a new task and the development of foundational research, including the datasets, metrics, and solutions. In addition to this dataset, we are also working on a series of follow-up efforts to expand the dataset.
>
> **Q5:**
> The paper mentioned, "This is because, in MER, text is emotionally ambiguous; without integrating other modalities, we cannot accurately understand the emotions conveyed through text." However, as far as I know, there are datasets like MOSI and MOSEI where the text/language modality contributes more than other modalities, as evidenced by relevant experimental results [1-4]. Is this explanation of yours appropriate, or is it specific to this particular dataset?
>
> **A5:**
> Thanks for your valuable comment. The contribution of different modalities varies across different datasets. As proved in previous research [1], the contribution of the text/language modality in MER2023 is smaller than that of other modalities (our OV-MERD is derived from MER2023). However, in the CMU-MOSI and CMU-MOSEI datasets, the contribution of the text/language modality is greater than that of the other modalities. The original text was not accurate, and we will correct our expression in the revised PDF.
>
> [1] Lian, Zheng, Licai Sun, Yong Ren, Hao Gu, Haiyang Sun, Lan Chen, Bin Liu, and Jianhua Tao. "Merbench: A unified evaluation benchmark for multimodal emotion recognition." arXiv preprint arXiv:2401.03429 (2024).

---

> ### Comment · Reviewer_sjuE · 2024-11-20
> **Response to the Authors**
>
> Thanks for your reply. I can see that you have made some revisions in the rebuttal revision based on the replies. For example, in the section Main Results on CLUE-M/A/T/V. You mentioned that the results on CLUE-Multi/A/V/T reflect the performance upper bound. Could you tell me briefly what changes have been made?

---

> > ### Author Response · Authors · 2024-11-21
> > **Response to Reviewer sjuE**
> >
> > Thanks very much for your response. Based on your and other reviewers' valuable comments, we made some revisions:
> >
> >    **Section 1:** In Figure 1, we clarified that the original data is not in a cartoon style. Since the original videos contain real people, we used DemoAI to remove personal information to address copyright concerns. Additionally, we added a more detailed explanation of the motivation in Appendix A.
> >
> >    **Section 2:** We added more dataset statistics and compared our dataset with more emotional datasets. We further explained the challenges of ALLM in capturing paralinguistic information. Moreover, we discussed in more detail the LLM-based merging process and the reasons behind it. We also elaborated on the experimental design and similarity measures used in the language influence experiment.
> >
> >    **Section 4:** We provided a more detailed explanation of the baseline generation process and corrected a typo in Figure 4.
> >
> >    **Section 5:** We explained the reasons for selecting CLUE-M/A/T/V as baselines and redrew Figure 5 to meet aesthetic requirements. We conducted a deeper analysis of multimodal fusion, human-LLM collaboration, and metric comparison. Additionally, we improved the analysis of modality preference in our experiment part.
> >
> >    **Section 6:** We further explained future work and added some suggestions on how to design a powerful framework to address our OV-MER task.
> >
> >   **Appendix:** Due to page limitations, we included additional experimental analysis in the appendix.
> >
> >
> > While there are some minor issues with the expression, we make every effort to clarify them. We believe this work is an important extension of the traditional MER task and will contribute to the development of this field. We look forward to further discussion with you and hope that you will consider increasing the score.

---

> > > ### Comment · Reviewer_sjuE · 2024-11-22
> > > **Response to the Authors**
> > >
> > > Thank you for your reply. I find the revised version clearer than the original. I currently have no further questions and I will consider adjusting the score.

---

> > > > ### Author Response · Authors · 2024-11-22
> > > > **Appreciation to Reviewer sjuE**
> > > >
> > > > We sincerely appreciate your support. A gentle reminder: in the current system, you can edit your original comment to adjust the score. Furthermore, in your comment, we notice your recognition of the contributions of our paper. Therefore, we hope you can give us a relatively high score to support our work.

---

> > > > > ### Author Response · Authors · 2024-11-26
> > > > > **Appreciation to Reviewer sjuE**
> > > > >
> > > > > Thanks for increasing the score. We sincerely appreciate your support.

---

> ### Comment · Reviewer_sjuE · 2024-11-27
>
> Hi, sorry for the late reply. I found this paper interesting, and you have addressed most of my concerns. Therefore, I have increased my score to 6. However, I have also noticed some questions raised by other reviewers and the AC ikpq, which require further discussion and consideration. I reserve the possibility of further adjustment of scores in the follow-up.
>
> In addition, I would like to thank the AC for his/her active participation in the discussion, which I believe is beneficial to the ICLR.

---

### Official Review · Reviewer_tZuA · 2024-10-28

**Soundness:** 2
**Presentation:** 1
**Contribution:** 2
**Rating:** 3
**Confidence:** 5

**Summary:**

Traditional emotion recognition tasks typically focus on predicting a single label for a given data point, such as "angry," "happy," or "surprised." This work argues that more nuanced emotions cannot be captured by this limited set of labels. Therefore, it proposes an open-vocabulary approach (OV-MER), where the number of categories is greatly expanded, thus facilitating the development of emotion AI. Overall, the main contributions include the development of datasets, new metrics, and a new task.

**Strengths:**

1. The authors have effectively expanded upon a previous dataset by implementing thoughtfully designed metrics grounded in fundamental emotional theories.

2. The problem formulation is clear, and the motivation is intuitive.

3. The experiments are well-conducted and organized.

4. Significant effort and comprehensive content have been included to ensure reproducibility.

**Weaknesses:**

1. While it's true that nuanced emotional labels can't always be conveyed by a single adjective, like "surprised," as it may stem from either positive or negative experiences, the VAD (valence, arousal, and dominance) framework can help mitigate this issue. This is why emotion recognition tasks are generally categorized as either discrete label-based or dimensional-based. By using VAD scores, we can better distinguish whether "surprise" is elicited by positive or negative emotions. Discrepancies do exist within each subject's understanding and annotators's agreement. However, it is not clear whether the issue was addressed by the proposed approach. In Appendix Section G, the authors state, 'Specifically, in the first round, we randomly select four annotators to check the clues and labels; in the second round, we merge the clues and labels reviewed by the first four annotators and ask another four annotators to perform a second round of checks.' How exactly are they aligned, and what has been removed/kept?

2. Section 2, which focuses on the construction of the 'OV-MERD dataset', is one of the most significant contributions of this work. However, the authors first discuss feature extractions and annotations, leaving readers unclear about the original data source until Section 2.3. This structure is not well-designed; as a reader, it became frustrating to navigate from the first page to Figure 2 and then through Sections 2.1 and 2.2 without understanding the origins (where and what) of the original data until Section 2.3 on page 4. This makes the paper difficult to follow.

3. In Section 2.3, the phrase "evenly select samples with further annotation" is vague. Does this refer to the original dataset (MER2023), which (I can only assume) has an even number of samples under each category? Or is it discussing the newly constructed dataset? I interpret it as referring to the original dataset based on Section E in the Appendix, but this should be stated more clearly.

4. Another concern is the grouping strategy. The authors utilized LLM and Putchik's emotional wheels to group several emotions into single labels. However, the order of discrete emotions in each group—such as "surprise," "nervous," and "dissatisfied"—is crucial, as different arrangements can lead to varied interpretations of the data points. This perspective is not addressed or mentioned anywhere in the paper.

5. Additionally, what are the visual and acoustic cues mentioned? Are there any low-level details? Or are they just text output from LLM?

6. A key issue in emotion recognition and affective computing is the cultural differences that influence emotional interpretation. People from different linguistic and cultural backgrounds can interpret and express emotions, like anger, in distinct ways. Although the authors primarily used Chinese and English in this work, they have not discussed these potential differences, especially considering they propose new metrics and benchmarks. Further background on the 'expert annotators' is not detailed enough.

7. The discussion regarding "Limitations" is quite vague. What does "more effective" model mean? How do the authors intend to "broaden the scope of your evaluations"?

Minor Issues:
Does 'MLLM' first mentioned in the third contribution stand for Multimodal LLM? It needs to be specified.

The dataset comparison can be more comprehensive by including more recently released datasets in 2023 and 2024.

Overall, I appreciate the authors' efforts in constructing a new dataset and introducing new perspectives and metrics to this task. However, substantial changes are needed to improve the quality of the work.

**Questions:**

Please see the 'weaknesses.'

---

> ### Author Response · Authors · 2024-11-20
> **Response to Reviewer tZuA (Part1)**
>
> We sincerely thank you for your positive feedback on our motivations, problem formulation, experiments, and reproducibility. Meanwhile, your valuable comments help us improve this paper. We believe this work is an important extension of the traditional MER task and will contribute to the development of this field.
>
> **Q1:**
> While it's true that nuanced emotional labels can't always be conveyed by a single adjective, like "surprised," as it may stem from either positive or negative experiences, the VAD (valence, arousal, and dominance) framework can help mitigate this issue. This is why emotion recognition tasks are generally categorized as either discrete label-based or dimensional-based. By using VAD scores, we can better distinguish whether "surprise" is elicited by positive or negative emotions. Discrepancies do exist within each subject's understanding and annotators's agreement. However, it is not clear whether the issue was addressed by the proposed approach. In Appendix Section G, the authors state, 'Specifically, in the first round, we randomly select four annotators to check the clues and labels; in the second round, we merge the clues and labels reviewed by the first four annotators and ask another four annotators to perform a second round of checks.' How exactly are they aligned, and what has been removed/kept?
>
> **A1:**
> In Section 1, we discuss the motivation behind OV-MER. Although dimensional models can be used to distinguish different emotions, they are abstract and difficult for the general public to understand. For instance, the valence-arousal-dominance (VAD) model is one of the most widely used dimensional models. Given a VAD score (V=3.3, A=2.9, D=4.7), it is challenging for people to accurately interpret the corresponding emotion. According to previous research [1], this VAD score represents *sad*. Furthermore, the VAD model attempts to simplify emotions into three linear dimensions. However, emotions are inherently complex and often involve combinations of multiple emotional states, which cannot be fully captured by these three dimensions. **Therefore, this paper uses discrete models to describe emotions. To address the limitations of traditional one-hot MER, we propose the OV-MER task, aiming to better describe subtle emotional nuances in an open-vocabulary manner.**
>
> Additionally, in contrast to dimensional models, discrete models align more closely with how humans intuitively understand emotions, making it easier to achieve consensus across different annotators.
>
> [1] Verma, Gyanendra K., and Uma Shanker Tiwary. "Affect representation and recognition in 3D continuous valence–arousal–dominance space." Multimedia Tools and Applications 76 (2017): 2159-2183.
>
> **Q2:**
> Section 2, which focuses on the construction of the 'OV-MERD dataset', is one of the most significant contributions of this work. However, the authors first discuss feature extractions and annotations, leaving readers unclear about the original data source until Section 2.3. This structure is not well-designed; as a reader, it became frustrating to navigate from the first page to Figure 2 and then through Sections 2.1 and 2.2 without understanding the origins (where and what) of the original data until Section 2.3 on page 4. This makes the paper difficult to follow.
>
> **A2:**
> The reason we use this structure is that **in Section 2, we primarily emphasize the human-LLM collaborative annotation process, as this is one of the key contributions of our paper, while the data source is just the practical part and can be replaced with other datasets.** Therefore, we first discuss the annotation process and then use this process to create the OV-MERD dataset. To address your concerns, we will add an additional introduction about the dataset in Section 2.
>
>
> **Q3:**
> In Section 2.3, the phrase "evenly select samples with further annotation" is vague. Does this refer to the original dataset (MER2023), which (I can only assume) has an even number of samples under each category? Or is it discussing the newly constructed dataset? I interpret it as referring to the original dataset based on Section E in the Appendix, but this should be stated more clearly.
>
> **A3:**
> The original text is: "This dataset is an extension of MER2023, from which we evenly select samples with further annotations.". This sentence does not mean that MER2023 has an equal number of samples for each category but discusses the newly constructed dataset. We will clarify this sentence in the revised PDF.

---

> > ### Author Response · Authors · 2024-11-20
> > **Response to Reviewer tZuA (Part2)**
> >
> > **Q4:**
> > Another concern is the grouping strategy. The authors utilized LLM and Putchik's emotional wheels to group several emotions into single labels. However, the order of discrete emotions in each group—such as "surprise," "nervous," and "dissatisfied"—is crucial, as different arrangements can lead to varied interpretations of the data points. This perspective is not addressed or mentioned anywhere in the paper.
> >
> > **A4:**
> > **The order of the discrete emotions has no effect on the final results.** Specifically, if we change the order of predictions from ["surprise", "nervous", "dissatisfied"] to ["nervous", "dissatisfied", "surprise"], we will obtain the same evaluation results.
> >
> > Meanwhile, we would like to emphasize that in OV-MER, we do not assign different levels of importance to each label (see Section 6). Every emotion holds equal significance, and neglecting anyone can impact the performance of downstream tasks.
> >
> > **Q5:**
> > Additionally, what are the visual and acoustic cues mentioned? Are there any low-level details? Or are they just text output from LLM?
> >
> > **A5:**
> > **This is what we do during the annotation process (see Section 2).** To generate rich and diverse emotion-related descriptions for audio and video, we first used ALLM and VLLM to extract the initial cues from audio and video, followed by two rounds of manual checks to eliminate errors and duplicates, while also adding missing information. To help understanding, we provide examples in Appendix E that contain rich emotion-related low-level details, such as descriptions of eyes, mouth, head movements, tone of voice, etc.
> >
> >
> > **Q6:**
> > A key issue in emotion recognition and affective computing is the cultural differences that influence emotional interpretation. People from different linguistic and cultural backgrounds can interpret and express emotions, like anger, in distinct ways. Although the authors primarily used Chinese and English in this work, they have not discussed these potential differences, especially considering they propose new metrics and benchmarks. Further background on the 'expert annotators' is not detailed enough.
> >
> > **A6:**
> > The cultural difference in emotion interpretation is a big, challenging, and open problem, but **this paper does not involve cultural differences**. Specifically, our original data is in Chinese, and the annotators we hired are also native Chinese speakers. Cultural differences only exist when the original data and the annotators are from different cultures.
> >
> > You may misunderstand some parts of our paper. The reason why we use both Chinese and English OV labels is that some emotions are easy to describe in one language but difficult to describe in another. This "language influence" discussed in our paper is different from the "cultural differences in emotion interpretation" that you mentioned. **In the future, we will also try to extend our method to other cultures and conduct further analysis of cultural differences.**
> >
> >
> > **Q7:**
> > The discussion regarding "Limitations" is quite vague. What does "more effective" model mean? How do the authors intend to "broaden the scope of your evaluations"?
> >
> > **A7:**
> > For the first limitation, the original text is: "The main contribution of this paper is the definition of a new task and the conduct of foundational research, including the development of datasets, metrics, and solutions. In the future, we plan to design more effective frameworks to solve OV-MER." **“More effective models” refers to our plan to design more efficient frameworks to better solve the OV-MER task.** To achieve this, we can conduct research in the following aspects: 1) We plan to follow the current mainstream approaches in MLLMs, including incorporating more emotion-related instruction datasets and utilizing more powerful LLMs. 2) As mentioned in our paper, how to integrate subtitle information and fuse multimodal inputs also plays a crucial role in the final performance. We will continue to work on these aspects in our future research. 3) We plan to use unsupervised learning or semi-supervised learning to exploit the relevant knowledge in unlabeled data.
> >
> > For the second limitation, the original text is: "Meanwhile, we have evaluated some representative MLLMs, but not all models are covered. In the future, we will broaden the
> > scope of our evaluations." Here, **"broaden the scope of your evaluations" means that in future work, we plan to include more newly emerging MLLMs to enrich the benchmark.**
> >
> >
> > **Q8:**
> > Does 'MLLM' first mentioned in the third contribution stand for Multimodal LLM? It needs to be specified.
> >
> > **A8:**
> > Yes. We have specified it in the revised PDF. In Table 5 (see Appendix), we also summarize the main abbreviations and their meanings.

---

> > > ### Author Response · Authors · 2024-11-20
> > > **Response to Reviewer tZuA (Part3)**
> > >
> > > **Q9:**
> > > The dataset comparison can be more comprehensive by including more recently released datasets in 2023 and 2024.
> > >
> > > **A9:**
> > > In Appendix G, we have provided a comparison with some well-known emotion datasets. Based on your feedback, we have added more datasets, such as MSP-Podcast, JL-Corpus, EmoDB, EMOVO, MESD, eNTERFACE, etc. We welcome any suggestions for additional datasets that may be missing, and we will add them.
> > >
> > > We look forward to further discussion with you and hope that you will consider increasing the score.

---

> > > > ### Comment · Reviewer_tZuA · 2024-11-22
> > > > **Response to authors**
> > > >
> > > > Thank you to the authors for their responses. Please find my comments and follow-up questions below.
> > > >
> > > > **1.** Both dimensional and discrete approaches have their respective advantages and disadvantages in the emotion recognition task. For VAD, using a Likert scale to quantify the scores allows participants to rate the intensity of emotions on a defined scale, which can be helpful. On the other hand, discrete labels are more intuitive for the general public but often fail to quantify the exact degree of emotions experienced. It is unnecessary for the authors to argue the advantages of one approach over the other, as both serve different purposes and contexts. Additionally, my question regarding the annotators mentioned in Appendix Section G was not addressed.
> > > >
> > > > **2.** It is certain that the original data (MER2023) is not a contribution for this work, but the constructed new dataset is. Therefore, it is necessary to introduce it first with enough background information. If the annotation process is the primary contribution, and can be applied to other datasets as authors stated, then OV-MER should be applied to more datasets in this work, particularly those with a limited number of emotion labels, as listed in Table 1, instead of focusing solely on MER2023, even just for comparison purpose.
> > > >
> > > > **3.** Please clarify whether the current PDF available for review includes all the revisions made in response to the feedback.
> > > >
> > > > **4.** If changing the order of the annotations does not affect the evaluation results, this should be demonstrated through experiments or at least discussed in the paper.
> > > >
> > > > **5.** To confirm, it is the textual content generated by a LLM? That is ok; I just wanted to ensure this with a clear answer.
> > > >
> > > > **6.** Certainly, I understand that this is a large and challenging problem. I am not suggesting that the authors to address it in this paper, but it should at least be acknowledged or discussed within the scope of this work.
> > > >
> > > > **7.** Are these clarifications reflected in the revised PDF?
> > > >
> > > > **8.** Thank you for addressing my concern about the abbreviations; I noticed them.
> > > >
> > > > **9.** These datasets are among the most widely used in this field. Are there no other related datasets published between 2019 and the present?

---

> ### Author Response · Authors · 2024-11-25
> **Response to Reviewer tZuA (Part1)**
>
> Thanks for your reply. In the revised paper, we further clarify our work and highlight all changes in red.
>
> **Q1:**
> Both dimensional and discrete approaches have their respective advantages and disadvantages in the emotion recognition task. For VAD, using a Likert scale to quantify the scores allows participants to rate the intensity of emotions on a defined scale, which can be helpful. On the other hand, discrete labels are more intuitive for the general public but often fail to quantify the exact degree of emotions experienced. It is unnecessary for the authors to argue the advantages of one approach over the other, as both serve different purposes and contexts. Additionally, my question regarding the annotators mentioned in Appendix Section G was not addressed.
>
> **A1:**
> Yes, both discrete models and dimensional models have their advantages and drawbacks, and our OV-MER captures the strengths of both emotional models. Compared with the one-hot discrete models, OV-MER uses more dense discrete labels to capture more nuanced emotions, mimicking the continuous attributes of the dimensional model. Compared with the dimensional model, OV-MER uses discrete labels, making it closer to the way humans understand emotions. **Therefore, our OV-MER provides a bridge connecting discrete and dimensional models.**
>
> To illustrate which labels are removed or kept, we provide two examples in Appendix J, each with labels from multiple annotators. To ensure annotation quality, we hire professional annotators who are experts in affective computing and familiar with the definition of emotions. Some of these annotators are members of our team who specialize in affective computing. In Figure 18, as the character doesn't know which doctor to see, most annotators provide labels such as *confused* or *puzzled*. Based on his tone and expression, some annotators further provide labels like *anxious* and *serious*. In Figure 19, most annotators notice his *disapprove* based on the textual content. Combining other modalities, some annotators further note his *blame* and *accuse* of what others are planning to do. From these examples, we can observe that these annotators provided relatively reliable labels. However, some annotators may focus only on the most relevant labels and overlook some details. To ensure the comprehensiveness of the annotation results, we merge the labels checked by four annotators. For example, in Figure 18, the final merged labels are *troubled*, *focused*, *puzzled*, *anxious*, *worried*, *confused*, and *serious*. In the next round of checks, we invite another four annotators for a second round of checks. Through this process, we can ensure that each retained label is confirmed by at least one annotator in each round, thereby ensuring the comprehensiveness and accuracy of the annotation results.
>
>
> **Q2:**
> It is certain that the original data (MER2023) is not a contribution for this work, but the constructed new dataset is. Therefore, it is necessary to introduce it first with enough background information. If the annotation process is the primary contribution, and can be applied to other datasets as authors stated, then OV-MER should be applied to more datasets in this work, particularly those with a limited number of emotion labels, as listed in Table 1, instead of focusing solely on MER2023, even just for comparison purpose.
>
> **A2:**
> Thanks for your comment. In the revised paper, we add more background information about the dataset.
>
> **Q3:**
> Please clarify whether the current PDF available for review includes all the revisions made in response to the feedback.
>
> **A3:**
> Thanks for your comments. In the revised paper, we clarify this statement in Section 2.3.
>
>
> **Q4:**
> If changing the order of the annotations does not affect the evaluation results, this should be demonstrated through experiments or at least discussed in the paper.
>
> **A4:**
> In the revised paper, we add more discussion on the label order. This paper proposes three set-based metrics.
>
> $\text{Precision}_{\text{s}}$  indicates the number of correctly predicted labels;
>
> $\text{Recall}_{\text{s}}$ indicates whether the prediction covers all ground truth;
>
> and $\text{F}_{\text{s}}$ is the harmonic mean of two metrics, which is used for the final ranking. **It should be noted that if we change the label order in $\text{Y}$ and $\hat{\text{Y}}$, it will not result in any change in performance. Therefore, the order of the emotions has no effect on the final result.**
>
>
> **Q5:**
> To confirm, it is the textual content generated by a LLM? That is ok; I just wanted to ensure this with a clear answer.
>
> **A5:**
> Yes. We use ALLM and VLLM for pre-annotation and LLM for clue merging. All outputs are in text format.

---

> > ### Author Response · Authors · 2024-11-25
> > **Response to Reviewer tZuA (Part2)**
> >
> > **Q6:**
> > Certainly, I understand that this is a large and challenging problem. I am not suggesting that the authors to address it in this paper, but it should at least be acknowledged or discussed within the scope of this work.
> >
> > **A6:**
> > Thanks for your suggestion. We will discuss the cultural influences on emotional understanding in the revised paper.
> >
> > **Q7:**
> > Are these clarifications reflected in the revised PDF?
> >
> > **A7:**
> > Yes. All clarifications are reflected in the revised PDF. We mark all new content in red.
> >
> >
> > **Q8:**
> > Thank you for addressing my concern about the abbreviations; I noticed them.
> >
> > **A8:**
> > Yes. We add the missing abbreviation. Thanks for your suggestion.
> >
> > **Q9:**
> > These datasets are among the most widely used in this field. Are there no other related datasets published between 2019 and the present?
> >
> > **A9:**
> > Thanks for your suggestion. We try our best to search for emotion datasets released in 2023 and 2024, and we add the MC-EIU dataset to Appendix G. We welcome any suggestions for additional datasets that may be missing, and we will add them. **At the same time, we would like to emphasize that our OV-MERD is the first dataset that uses the human-LLM collaborative annotation strategy, aiming to provide richer labels to capture more nuanced emotions. We believe this work is an important extension of traditional MER and will contribute to the development of the field.**
> >
> > We look forward to further discussion with you and hope that you will consider increasing the score.

---

### Official Review · Reviewer_5yry · 2024-10-31

**Soundness:** 2
**Presentation:** 3
**Contribution:** 2
**Rating:** 5
**Confidence:** 4

**Summary:**

This work focuses on introducing a new perspective on emotion annotation by not restricting it to classic categories and taking advantage of large language models in collaboration with humans for a more nuanced translation into labels. The proposed method increases the label space of emotion labels with a more detailed multi-label perspective, which can promote efficiency and understanding of emotions by systems. Furthermore, using an LLM over the output from multimodal LLMs can promote a detailed, closer-to-machine understanding of these labels. By grouping labels and defining metrics for sound evaluation, the authors provide a deeper view of label similarity to assess the system’s true understanding of emotion.

**Strengths:**

The work presents an improvement over conventional emotion labeling by proposing a richer annotation which can enhance machine’s understanding of emotion. Additionally, the study focuses on the ambiguities and overlaps in emotional labels—a vital contribution to the field. Leveraging LLMs along human experts bridges the gap between human intuition and machine learning, making the model's emotional comprehension more transferable and adaptable.

**Weaknesses:**

dependency of this approach to a certain versions of large language model definitely has its own limitations. From known weaknesses of this models to potential future risks of such approach. Furthermore, attempts to control this is quite expensive and risky. The bigger concern that comes to my mind is the effect of mirrors reflection in learning. How can we be sure that from this circle of learning, a limited and biased understanding is not created for machine learning?
A broader concern is the issue of model feedback loops—how can we be confident that machine learning isn't developing a narrow, biased interpretation of emotions from these circular, LLM-driven annotations? This feedback cycle may compromise the depth of emotional understanding in machine learning.

**Questions:**

What is meant by the "short duration of video"? How can you be sure that a few frames (3) can thoroughly describe and capture facial motions indicating emotions? As you know, previous work in the field highlights the importance of facial movement features in video inputs, and as humans, we can imagine how fast movements can show excitement, which may not necessarily be captured by three frames. Furthermore, have you experimented to see if increasing the frame count could improve levels of hallucination in VLLM? Could the information gap be pushing the model?

As you mentioned, the role of VLLM control annotators includes adding missing content. How did the human annotators who evaluated the VLLM output capture visual data? Did they view the entire video or the same inputs as the VLLM?

What is the reason behind the difference in performance between matching-based and LLM-based metrics? I would prefer a more analytical study rather than just reporting results. I also suggest a deeper study on the effectiveness of multimodal fusion, as this title suggests an important question that has not been investigated as deeply as it could be.

Did you consider human evaluation of the combined output of ALLM and VLLM? How capable do you think this combination is of handling ambiguous and contradictory cases? Furthermore, how can we ensure that our fusion is unbiased?

I see that no annotation control has been used for the merging step. Do you have a control measure for this step? How can we be sure the merge is effective enough?

Regarding the language test, I appreciate that you included it, but one question that comes to mind is the purpose of comparing the similarity of the label sets from Yce to Yee. I believe these sets are not from the same data, as the former comes from data originally in Chinese (translated to English) and the latter from data in English. Given this, I understand the importance of comparing Yec to Yee, which can ensure that the model is not biased by language in the same data, but not Yce to Yee. In any case, I believe you could include more details on language.

In a comparison of human-only and human-LLM collaboration in clue generation, I couldn't understand why you relate the longer length of clues in human-LLM collaboration to informativeness regarding minor details. I don't question the assumption, as I see this fact more clearly in the comparison of label counts; perhaps elaborate more on that.

---

> ### Author Response · Authors · 2024-11-20
> **Response to Reviewer 5yry (Part1)**
>
> We appreciate your recognition of the "vital contribution to the field" made by this paper. We also thank you for highlighting that our approach can "make the model's emotional comprehension more transferable and adaptable." While there are some minor issues with the expression, we have made every effort to clarify them. We believe this work is an important extension of the traditional MER task and will contribute to the development of this field.
>
> **Q1:**
> Dependency of this approach to a certain versions of large language model definitely has its own limitations. From known weaknesses of this models to potential future risks of such approach. Furthermore, attempts to control this is quite expensive and risky. The bigger concern that comes to my mind is the effect of mirrors reflection in learning. How can we be sure that from this circle of learning, a limited and biased understanding is not created for machine learning? A broader concern is the issue of model feedback loops—how can we be confident that machine learning isn't developing a narrow, biased interpretation of emotions from these circular, LLM-driven annotations? This feedback cycle may compromise the depth of emotional understanding in machine learning.
>
> **A1:**
> You raise a very interesting point. In fact, we considered this issue when designing the annotation process and conducted some experimental validation.
>
> (1) **Annotation Process Design**: We employed multiple professional annotators and conducted several rounds of checking to address machine bias. Specifically, LLM-driven annotations are used as initial results, and all final decisions are made by multiple annotators (see Figure 2). To ensure annotation quality, we selected annotators with expertise in affective computing and familiarity with different emotion definitions. The involvement of multiple professional annotators ensures that the machine does not lead to narrow or biased interpretations. Additionally, as shown in Figure 17 (in Appendix J), our annotation platform includes an "Other Emotion" part. If the LLM-driven annotations fail to perfectly capture the character’s emotional state, annotators can manually add other labels. This process is also conducted by multiple professional annotators, ensuring that the emotional understanding is not constrained by machine learning biases.
>
> (2) **Experimental Validation**: We also considered this issue in our experiments by comparing the human-only annotation strategy with the human-LLM collaborative strategy (see Figure 5). From these results, the human-LLM collaborative strategy covers a broader range of emotions and provides more diverse labels for each sample. These results confirm that this feedback loop does not undermine the depth of emotional understanding. On the contrary, it helps uncover more complex and subtle emotional nuances.

---

> > ### Author Response · Authors · 2024-11-20
> > **Response to Reviewer 5yry (Part2)**
> >
> > **Q2:**
> > What is meant by the "short duration of video"? How can you be sure that a few frames (3) can thoroughly describe and capture facial motions indicating emotions? As you know, previous work in the field highlights the importance of facial movement features in video inputs, and as humans, we can imagine how fast movements can show excitement, which may not necessarily be captured by three frames. Furthermore, have you experimented to see if increasing the frame count could improve levels of hallucination in VLLM? Could the information gap be pushing the model?
> >
> > **A2:**
> > Thanks very much for your professional comment. In this paper, "short duration" does not refer to micro-expressions that last less than 0.04 seconds. It refers to our videos being relatively short compared to long videos (over 1 minute). In Appendix H, we analyze the duration distribution of the OV-MERD dataset and find that most samples have durations between 1 and 4 seconds. **In the revised PDF, we will modify the vague expression “short” to a more specific duration distribution.**
> >
> > Regarding the sampling strategy, we did not provide a detailed explanation due to space limitations. Specifically, we categorize visual clues into two types: **(1) visual clues with relatively long durations**; and **(2) visual clues with fast movements, such as eye movements, head movements, and micro-expressions.**
> >
> > **For the first type**, since the duration of most videos is between 1 and 4 seconds (see Appendix H) and the video content is usually continuous with only minor differences between adjacent frames (see Appendix C), uniformly sampling three frames are sufficient to capture this information; **For the second type**, we observe that current MLLMs (including the GPT-4V used in this paper) struggle to capture these fast movements. Increasing the number of sampled frames does not address this issue. Previous research has also shown that GPT-4V cannot recognize micro-expressions [1]. To capture these fast movements, we employ multiple professional annotators to manually add this information.
> >
> >
> > [1] Lu, Hao, Xuesong Niu, Jiyao Wang, Yin Wang, Qingyong Hu, Jiaqi Tang, Yuting Zhang et al. "Gpt as psychologist? preliminary evaluations for gpt-4v on visual affective computing." In Proceedings of the IEEE/CVF Conference on Computer Vision and Pattern Recognition, pp. 322-331. 2024.
> >
> >
> > **Q3:**
> > As you mentioned, the role of VLLM control annotators includes adding missing content. How did the human annotators who evaluated the VLLM output capture visual data? Did they view the entire video or the same inputs as the VLLM?
> >
> > **A3:**
> > Appendix J provides detailed information about our annotation process. In this process, we designed an interface with a time slider that allows annotators to play the video starting from any frame, enabling them to capture subtle changes in the video. Therefore, we allow annotators to view the entire video during the manual check. Compared to VLLM, annotators can observe more information, helping them better annotate details missed by VLLM, such as fast-moving visual cues, including eye movements, head motions, and micro-expressions. To reduce annotation bias and add missing content, we also employed multiple professional annotators and conducted multiple rounds of manual checks.
> >
> > **Q4:**
> > What is the reason behind the difference in performance between matching-based and LLM-based metrics? I would prefer a more analytical study rather than just reporting results.
> >
> > **A4:**
> > In Section 5, we provide some analysis of these two metrics. These results highlight the limitations of using matching-based metrics to evaluate the OV-MER task. The main reason is that matching-based metrics focus on low-level word-level matches, while emotion understanding is a relatively complex, high-level perceptual task. Therefore, matching-based metrics are not suitable. Consider the following three sentences:
> >
> > \#1. The clue is "the weather is great". His emotion is "happy".
> >
> > \#2. The clue is "the weather is bad". His emotion is "sad".
> >
> > \#3. His emotion is "happy".
> >
> > For matching-based metrics, we use $BLEU_{1}$ as an example. The $BLEU_1$ score between \#1 and \#2 is 0.8181, while the $BLEU_{1}$ score between \#1 and \#3 is 0.1738. Therefore, based on the $BLEU_{1}$ score, \#1 is closer to \#2. For LLM-based metrics, we first extract the emotion labels and compare their similarity, so \#1 is closer to \#3. This demonstrates that matching-based metrics are not suitable for evaluating emotion recognition performance.

---

> > > ### Author Response · Authors · 2024-11-20
> > > **Response to Reviewer 5yry (Part3)**
> > >
> > > **Q5:**
> > > I also suggest a deeper study on the effectiveness of multimodal fusion.
> > >
> > > **A5:**
> > > Due to page limitations, we have not fully discussed the impact of multimodal fusion. We will provide a more in-depth analysis in the appendix:
> > >
> > > In Table 3, we select the best-performing ALLMs and VLLMs and explore whether their combinations can lead to better performance. To fuse different modalities, we input prelabeled acoustic and visual clues into LLM and leverage its reasoning capabilities for multimodal integration (see Figure 4). This approach is consistent with the fusion method used during our dataset construction process (see Section 2). We observe that multimodal results generally outperform ALLM-only or VLLM-only results. **The reason lies in that emotions are conveyed through various modalities. By integrating different modalities, we can obtain a more comprehensive understanding of emotions, leading to better performance in OV-MER.**
> > >
> > >
> > > **Q6:**
> > > Did you consider human evaluation of the combined output of ALLM and VLLM? How capable do you think this combination is of handling ambiguous and contradictory cases? Furthermore, how can we ensure that our fusion is unbiased? I see that no annotation control has been used for the merging step. Do you have a control measure for this step? How can we be sure the merge is effective enough?
> > >
> > > **A6:**
> > > This is indeed an interesting point and thanks for your professional comment. Ambiguity and contradictions in multimodal fusion are inevitable in the MER task. In this paper, we mainly rely on the strong reasoning capabilities of LLM to address this issue. Specifically, as shown in Section 2, we ask LLM to integrate text, audio, and visual cues and further infer the emotional state. We observe that LLM can produce reasonable analytical results.
> > >
> > > Additionally, we would like to point out that ambiguity and contradictions in multimodal fusion are indeed a challenging and open problem. In this paper, we use a simple approach to address it, providing an implementable solution for the OV-MER task. In practice, there may be more effective strategies, such as using more complex prompts or incorporating the control measures you mentioned. In the future, we will conduct further research in this aspect. We add this discussion in Appendix E.
> > >
> > >
> > > **Q7:**
> > > Regarding the language test, I appreciate that you included it, but one question that comes to mind is the purpose of comparing the similarity of the label sets from Yce to Yee. I believe these sets are not from the same data, as the former comes from data originally in Chinese (translated to English) and the latter from data in English. Given this, I understand the importance of comparing Yec to Yee, which can ensure that the model is not biased by language in the same data, but not Yce to Yee. In any case, I believe you could include more details on language.
> > >
> > > **A7:**
> > > Thanks for your recognition of our experiments. In this section, we analyze from two perspectives: 1) **the impact of descriptive language (Clue-Multi)**, and 2) **the impact of abstract language (OV labels)**. The $Y_{EE}$ to $Y_{EC}$ (or $Y_{CC}$ to $Y_{CE}$) experiment aims to keep the descriptive language consistent to analyze the effect of abstract language, while the $Y_{CE}$ to $Y_{EE}$ (or $Y_{EC}$ to $Y_{CC}$) experiment aims to keep the abstract language consistent to analyze the effect of descriptive language. Thus, we conducted the analysis from these two perspectives. In Appendix K, we detail our experimental design.
> > >
> > >
> > > **Q8:**
> > > In a comparison of human-only and human-LLM collaboration in clue generation, I couldn't understand why you relate the longer length of clues in human-LLM collaboration to informativeness regarding minor details. I don't question the assumption, as I see this fact more clearly in the comparison of label counts; perhaps elaborate more on that.
> > >
> > > **A8:**
> > > Thanks very much for your professional feedback. We would like to further explain the logic behind our approach:
> > >
> > > As the philosopher Wittgenstein [1] pointed out, language is not necessarily better when it is longer; precise language is often more important. Therefore, we fully agree with your point that longer clues do not necessarily mean more information. However, he also mentioned that, in the absence of precise language, richer language becomes increasingly important. In the MER task, emotions are relatively abstract, and we cannot always find precise language to describe them. Therefore, we try to use richer language. This is also the reason why we use long descriptions and OV labels to describe emotions.
> > >
> > > [1] Hacker, Peter Michael Stephan. Wittgenstein: Comparisons and context. OUP Oxford, 2013.

---

> > > > ### Comment · Reviewer_5yry · 2024-11-24
> > > >
> > > > I appreciate the thorough and well-tailored responses to comments 1 through 5. However, I found the response to comment 8 less satisfactory, as I was hoping for a more experimental perspective rather than a philosophical one, given the delicate nature of the concept discussed. Regarding comment 6, I believe that, given the scale of the evaluation conducted in this work, incorporating representations of contradiction and ambiguity could significantly enhance its utility as a benchmark for more challenging emotion analysis tasks. That said, I understand if the authors prefer to leave this discussion for future work due to its inherent complexity. Accordingly,  I have no more questions and I would like to keep my previous score.

---

> > > > > ### Author Response · Authors · 2024-11-25
> > > > > **Response to Reviewer 5yry**
> > > > >
> > > > > Thanks for your professional comments. Below are some clarifications:
> > > > >
> > > > > Regarding Q8, inspired by your suggestion, we further calculate the Pearson correlation coefficient (PCC) score between the description length and the number of labels for each sample. We observe that, for the human-only strategy, the PCC score is 0.3416, and for the human-LLM collaboration strategy, the PCC score is 0.2939. These results further validate your guess. Although from the dataset level, the length of descriptions is related to the richness of labels (see Figure 5), these two metrics do not show a strong correlation at the sample level. In the revised paper, we clarify this point in Appendix O and revise our analysis part.
> > > > >
> > > > > Regarding Q6, ambiguity and contradictions in multimodal fusion are indeed a challenging and open problem. In the future, we will conduct further research in this aspect.
> > > > >
> > > > > Finally, thanks for your professional advice and comments. We believe this work is an important extension of traditional MER, and we will continue to dedicate ourselves to this work and the development of the field.

---

> > > > > > ### Comment · Reviewer_5yry · 2024-11-25
> > > > > >
> > > > > > I appreciate your dedication and clarity in reporting. I will consider increasing my score. I would like to see this work introduced in the field as I see the future need for more detailed study over understanding and representing emotions.

---

> > > > > > > ### Author Response · Authors · 2024-11-26
> > > > > > > **Appreciation to Reviewer 5yry**
> > > > > > >
> > > > > > > We sincerely appreciate your support.

---

> ### Comment · Reviewer_5yry · 2024-11-29
>
> Thank you for your detailed responses. After reviewing the Area Chair's (AC) comments and re-evaluating the feedback from other reviewers, I have a few additional questions and concerns:
>
> 1.	With updates to LLMs, would the label space change? One concern is the cost of labeling datasets using your approach, especially given the significant reliance on expert annotators, as you indicated. Do experts need to control the label space each time a new dataset or new version of LLM is used? How feasible is it in practice?
>
> 2.	How can we ensure that existing machine learning solutions can adapt to this new baseline? From your earlier response, I understand that adapting to this setup is "almost infeasible." If this is the case, how practical is it to implement this shift, particularly when the benchmark is based on a limited set of datasets (e.g., a single dataset)? My concern is whether the broader research community will accept this if it limits baseline comparisons or the range of datasets.
>
> 3.	How can methods be fairly compared in this setup to confidently conclude that your approach improves learning of emotional models? Table 7 is presented as evidence of effectiveness, but I don’t see a clear or fair comparison—lower performance is reported, but relative to what? Furthermore, is there any discussion or analysis showing that models trained with your dataset are demonstrably more emotionally intelligent compared to those trained on traditional datasets? While the variation in label count or the use of a broader label range is mentioned in response to a similar comment by Reviewer GTfB (Q2), I don’t believe this necessarily indicates greater informativeness or improved learning.
>
> 4.	How does training work in this setup? How is the model provided with a loss function, and how is the loss landscape defined and optimized in this new framework? You reported a cost of $1 per epoch, so am I correct in assuming that GPT acts as a loss function in this setup? I find this approach challenging both in terms of its cost and its potential reception in the literature, as well as concerns about its trustworthiness.
>
> 5.	I find question 3 from the AC debatable. While I agree that the approach does not necessarily introduce new emotions, as suggested, the discussion around Fig. 5 indicates that it enables the use of a broader set of emotion labels rather than expanding the label space itself. However, I do not see a clear analysis demonstrating how these additional labels are more informative or how they necessarily introduce new information. From your response to reviewer tZuA Q1, I understand that your method differs from VAD systems by being more aligned with the "general public's" understanding of emotions. However, this does not necessarily address how it contributes to improved machine learning performance, which ties back to question 3.

---

> ### Author Response · Authors · 2024-12-03
> **Response to Reviewer 5yry (Part1)**
>
> Thanks for your valuable comments. Below are some clarifications:
>
> **Q1:**
> With updates to LLMs, would the label space change? One concern is the cost of labeling datasets using your approach, especially given the significant reliance on expert annotators, as you indicated. Do experts need to control the label space each time a new dataset or new version of LLM is used? How feasible is it in practice?
>
> **A1:**
> Thanks for your insightful comments. We would like to clarify that the discussion regarding LLMs is beyond the scope of this paper. Technically, during the annotation process, we used GPT, as it is one of the most powerful LLMs. Meanwhile, we propose to use the human-LLM collaborative strategy in emotion annotation, which is a step from zero to one. The detailed impact of different LLMs will be discussed in our future work.
>
> In our annotation process, experts are not required to control the label space. *Annotators are free to use any emotional label they deem appropriate, with LLMs serving only as a reference.* Specifically, we employed eight annotators and conducted two rounds of checks to ensure that each label was confirmed by at least one annotator in each round, thereby ensuring the accuracy of the annotations.
>
>
> **Q2:**
> How can we ensure that existing machine learning solutions can adapt to this new baseline? From your earlier response, I understand that adapting to this setup is "almost infeasible." If this is the case, how practical is it to implement this shift, particularly when the benchmark is based on a limited set of datasets (e.g., a single dataset)? My concern is whether the broader research community will accept this if it limits baseline comparisons or the range of datasets.
>
> **A2:**
> Thanks for your comments. *Regarding the baseline, this paper shifts from traditional classification models to LLM-driven approaches, aligning with current research trends.* As more LLMs and MLLMs emerge, the baseline for OV-MER will become increasingly rich. We believe that this shift is practical and positive for the future of emotion recognition research.
>
> Meanwhile, introducing new concepts inevitably will face many challenges. We do not intend to replace the mainstream work focused on basic emotion recognition. *Our goal is to provide the community with a different perspective, highlighting that some subtle emotions cannot be perfectly described by basic emotions.* In other words, we believe that emotion recognition should not be limited to basic emotions but should expand to include more nuanced emotions. Our work is only laying the foundation for OV-MER, and the further development of this task will require the collective effort of many researchers. **Currently, some researchers have already started using our dataset in their works.**
>
>
> **Q3:**
> How can methods be fairly compared in this setup to confidently conclude that your approach improves learning of emotional models? Table 7 is presented as evidence of effectiveness, but I don’t see a clear or fair comparison—lower performance is reported, but relative to what? Furthermore, is there any discussion or analysis showing that models trained with your dataset are demonstrably more emotionally intelligent compared to those trained on traditional datasets? While the variation in label count or the use of a broader label range is mentioned in response to a similar comment by Reviewer GTfB (Q2), I don’t believe this necessarily indicates greater informativeness or improved learning.
>
> **A3:**
> Thanks for your comments. First, the main purpose of Table 7 is to demonstrate that basic emotions cannot fully capture the complex emotional states of characters. Therefore, we need to use more nuanced labels beyond basic emotions and provide richer labels for each sample. Here, we provide an example. In predicting the emotional labels of the instance in Figure 1(b), existing methods can only predict labels within the basic emotion categories (e.g., "surprise"), whereas our method can output more nuanced and diverse emotions (e.g., "surprise, nervous, dissatisfied"). *By identifying these more subtle emotional labels, we can enhance the model's emotional intelligence.*
>
> To address your second question, we conducted a user study to verify whether our dataset provides greater informativeness than basic emotions. Specifically, we hired 4 participants and randomly selected 10 samples from our dataset. For each sample, we provided both the basic emotion label and the OV-MER label. The following instruction was given: *Which label provides greater informativeness? The label with more information was marked as 1, and the other label as 0.* The results showed that 97.5\% of the results believed that our OV-MER provided more information. This user study also helps address the concerns raised by AC. The user study results will be submitted to the supplementary material.

---

> > ### Author Response · Authors · 2024-12-03
> > **Response to Reviewer 5yry (Part2)**
> >
> > **Q4:**
> > How does training work in this setup? How is the model provided with a loss function, and how is the loss landscape defined and optimized in this new framework? You reported a cost of \$1 per epoch, so am I correct in assuming that GPT acts as a loss function in this setup? I find this approach challenging both in terms of its cost and its potential reception in the literature, as well as concerns about its trustworthiness.
> >
> > **A4:**
> > Thanks for your comments. First, we can use other LLMs to replace GPT, such as Qwen2 and LLaMA. Secondly, during training, *we do not need to use GPT for loss calculation but instead follow the instruction tuning technique.* Instruction tuning involves fine-tuning the model on a dataset that contains various instructions and their corresponding responses, enabling the model to learn how to interpret and execute these instructions more effectively. In our OV-MER, the instruction is *"Please identify the character's emotional state and output all possible emotional labels,"* with the response set to *"surprised, nervous, dissatisfied,"* and we use the typical cross-entropy loss commonly used in LLMs for loss calculation.
> >
> >
> > **Q5:**
> > I find question 3 from the AC debatable. While I agree that the approach does not necessarily introduce new emotions, as suggested, the discussion around Fig. 5 indicates that it enables the use of a broader set of emotion labels rather than expanding the label space itself. However, I do not see a clear analysis demonstrating how these additional labels are more informative or how they necessarily introduce new information. From your response to reviewer tZuA Q1, I understand that your method differs from VAD systems by being more aligned with the "general public's" understanding of emotions. However, this does not necessarily address how it contributes to improved machine learning performance, which ties back to question 3.
> >
> > **A5:**
> > Thanks for your comments. In **A3**, we conducted a user study, and the results showed that 97.5\% of results believed our OV-MER provided more information. Therefore, we can conclude that OV-MER offers more information than basic emotions, helping the model better understand human emotions. Please refer to our **A3** for more details.
> >
> > Secondly, we would like to further clarify the necessity of these additional emotion labels. First, current works in the machine learning field addressing emotion recognition tasks primarily rely on basic emotion theory, using labels such as *anger*, *disgust*, *happiness*, *sadness*, *fear*, and *surprise*. However, considering the complexity of human emotions, using these basic labels to describe an individual's emotional state may not be entirely accurate. For example, *jealousy* and *hostility* are two distinct emotions, but under basic emotion theory, both are categorized as *anger*, leading to a loss of some emotional nuances. Therefore, in this paper, our aim is to shift the emotion recognition task from basic emotions to more nuanced emotions, enabling the model to distinguish between different emotions like *jealousy* and *hostility*, thereby enhancing the emotional intelligence of the machine.
> >
> > Finally, we would like to clarify that our goal is to offer a different perspective to the community, emphasizing that some subtle emotions cannot be perfectly described by basic emotions. Our OVL aims to offer an approach, using flexible, richer, and more nuanced labels for emotional annotation, rather than relying on basic emotions. *We hope that OVL can provide a new perspective for emotion recognition tasks, inspiring further innovation in the current field of emotional AI. Meanwhile, we hope that the AC and reviewers will notice the importance of our OV-MER task.*

---

### Official Review · Reviewer_XYyA · 2024-11-02

**Soundness:** 3
**Presentation:** 3
**Contribution:** 4
**Rating:** 8
**Confidence:** 4

**Summary:**

This paper centers on the task of multimodal emotion recognition. Unlike prior work, which primarily assigns a single label and fixes the label space, this study expands upon earlier tasks, enabling the prediction of multiple labels across various categories. This approach allows for a more nuanced representation of a person’s emotional state. In addition to establishing this new task, the paper also presents a newly constructed dataset, defines evaluation metrics and baseline systems, and undertakes foundational work within this area. Furthermore, the experimental sections offer insights and guidelines for advancing solutions to this task in future research.

**Strengths:**

This paper introduces a novel task, OV-MER, which extends traditional multimodal emotion recognition (MER) to more comprehensively capture individuals' emotional states. The authors have conducted foundational research in this area by constructing a dataset, defining evaluation metrics, and proposing initial solutions. Overall, the paper is well-structured, and this advancement may enhance the application of MER in downstream tasks such as human-computer interaction. Additionally, the paper includes discussions on ethics and reproducibility.

**Weaknesses:**

1. The color scheme in Figure 5 does not align with the main visual style of the paper. It is recommended to redraw this figure for consistency.
2. In Section 6, please provide additional elaboration on the recommendations for framework design, as this could be valuable for future researchers.
3. Traditional facial emotion recognition methods often utilize AU units and assign a single label per face. However, this paper’s OV-MER assigns multiple labels to a video. Could you clarify why videos in multimodal scenarios are suited to convey multiple emotions?
4. Table 1 includes only a limited selection of emotion datasets. Please consider expanding it to include additional datasets, such as MSP-Podcast.
5. In Table 2, please further clarify the distinctions between the labels extracted from CLUE-Multi and the final ground truth.

**Questions:**

Please refer to my comments on weaknesses.

---

> ### Author Response · Authors · 2024-11-20
> **Response to Reviewer XYyA**
>
> We sincerely appreciate your recognition of the importance of this paper and the acknowledgment of our contributions. In this paper, we introduce a new task called OV-MER, for which we further construct a dataset, define evaluation metrics, and propose solutions.  While there are some minor issues with the expression, we have made
> every effort to clarify them. We believe this work is an important extension of the traditional MER task and will contribute to the development of this field.
>
> **Q1:**
> The color scheme in Figure 5 does not align with the main visual style of the paper. It is recommended to redraw this figure for consistency.
>
> **A1:**
> Thanks for your suggestion. We update Figure 5 in the revised PDF.
>
> **Q2:**
> In Section 6, please provide additional elaboration on the recommendations for framework design, as this could be valuable for future researchers.
>
> **A2:**
> In this paper, we prove that how to integrate subtitle information and fuse multimodal inputs plays a crucial role in the final performance. We will work on these aspects in the framework design. Meanwhile, we plan to follow the current mainstream approaches in MLLMs, including incorporating more emotion-related instruction datasets and utilizing more powerful LLMs. We add this discussion in our revised paper.
>
>
> **Q3:**
> Traditional facial emotion recognition methods often utilize AU units and assign a single label per face. However, this paper’s OV-MER assigns multiple labels to a video. Could you clarify why videos in multimodal scenarios are suited to convey multiple emotions?
>
> **A3:**
> Video emotion is more complex than facial emotion. This is because, in videos, we need to capture subtle changes in the temporal dimension and integrate multimodal clues. Take Figure 1(b) as an example. In the temporal dimension, we need to infer a person's *nervousness* based on his stuttering; in the multimodal dimension, we need to combine information from different modalities to gain a more comprehensive understanding of emotion. **Due to the complexity of video emotion, using a single label is limiting, and more discrete labels are required to better describe video emotion.** We add this discussion to Appendix A.
>
>
> **Q4:**
> Table 1 includes only a limited selection of emotion datasets. Please consider expanding it to include additional datasets, such as MSP-Podcast.
>
> **A4:**
> Thanks for your suggestion. In Appendix G, we compare with more emotion datasets, including MSP-Podcast, JL-Corpus, EmoDB, EMOVO, MESD, eNTERFACE, etc,
>
> **Q5:**
> In Table 2, please further clarify the distinctions between the labels extracted from CLUE-Multi and the final ground truth.
>
> **A5:**
> We highlight this point in Figure 2(b). In our preliminary experiments, we observed that the labels extracted from different language versions of CLUE-Multi show some discrepancies. To eliminate language influence and achieve consensus labels, we merge these labels and conduct two-round manual checks. The output after manual checks is considered the final ground truth.

---

### Author Response · Authors · 2024-11-20
**Global Response**

Dear Reviewers, Area Chairs, and Program Chairs:

We would like to express our gratitude to all the reviewers for taking their valuable time to review our paper. We sincerely appreciate the reviewers' recognition of the importance of our proposed task, the novelty of our method, and the richness of our experiments. Meanwhile, we hope the reviewers can value the originality of this work. While there are some minor issues with the expression, we have made every effort to clarify them. We believe this work is an important extension of the traditional MER task and will contribute to the development of this field.

At the same time, we appreciate the reviewers for pointing out the shortcomings. Your valuable comments help us improve this paper. We kindly ask the reviewers to take the above clarifications into account when considering score adjustments. We welcome any further discussion with the reviewers.

Best regards,

Paper5870 Authors

---

### Comment · Area_Chair_ikpq · 2024-11-26
**Questions from the AC**

Dear Authors,

I have been following your paper and the discussions with interest, and would like to ask a few follow-up questions.

1. How does the proposed framework/paradigm, namely “open-vocabulary,” differ from established concepts such as “multi-label learning” and “partial label learning”? Both of these areas have been explored extensively, including in affective computing and other domains of human-machine systems/computing.

2. While the label space is more relaxed in your framework, it remains finite. This raises the question of whether the term “open” is accurate. Could you clarify your perspective on this?

3. Additionally, the claim that this paradigm “broadens” the “range of emotion labels” seems debatable. From my understanding, this approach primarily enables capturing simultaneous emotions rather than introducing a fundamentally broader range or type of emotions. Could you elaborate on this point?

4. Finally, are any of the baselines explicitly designed for multi-label learning? Given the existence of established methods in this area, comparisons to such approaches seem essential for a fair and complete evaluation.

Best regards,\
AC

---

> ### Author Response · Authors · 2024-11-28
> **Response to AC (Part1)**
>
> Thanks for your interest in our work. We are honored to have the opportunity to answer your questions.
>
> **Q1:**
> How does the proposed framework/paradigm, namely “open-vocabulary,” differ from established concepts such as “multi-label learning” and “partial label learning”? Both of these areas have been explored extensively, including in affective computing and other domains of human-machine systems/computing.
>
> **A1:**
> Thanks for your meaningful comments. We would like to clarify the differences between them from the following aspects:
>
> **1. Different Task Definition.**
> Open-vocabulary learning (OVL) differs from multi-label learning (MLL) and partial-label learning (PLL) primarily in its focus on generalizing to *unseen or novel labels*, whereas MLL and PLL operate within *a fixed label space*. In MLL, multiple labels are assigned to each sample [1], while in PLL, a set of candidate labels are assigned for each sample, with only one correct label [2]. In OVL, we aim to recognize labels outside of the predefined label space [3]. As such, the key difference lies in the ability of OVL to generalize to new labels dynamically, which extends the static capabilities of MLL and PLL.
>
> **2. Emergent Affective States in Real Worlds.** Human affective states are diverse and context-dependent, often extending beyond predefined taxonomies (e.g., beyond Ekman’s basic emotions). As psychologist Plutchik pointed out, humans can express approximately 34,000 different emotions [4]. Open-vocabulary methods allow models to understand and predict emotional states beyond these predefined taxonomies. Meanwhile, existing MLL and PLL methods have limited emotion label scope, and failed to adapt to open-world tasks.
>
> **3. Relationship Between OVL and MLL\&PLL.** Theoretically, MLL or PLL can be converted to OVL by spanning the label space of MLL or PLL to a complete set that includes all the emotion labels in the language (e.g., around 34,000 different emotions). However, this is not feasible in practice to construct such kind of dataset for MLL and PLL, i.e., annotators need to label each sample with 34,000 different emotions, and it's hard to collect sufficient samples for every emotion category, let alone multiple labels). This is why the largest emotion category number in existing datasets is only 23 (EmotioNet), while in our work, using OVL, we can easily extend the number of emotion categories to 248.
>
> **4. Evaluation Settings.** In our OVL benchmark, we adopt a zero-shot setting where the sample labels are selected as the most reliable emotional descriptors from an unconstrained vocabulary. Consequently, *the label spaces in the training and test sets are different*, meaning the test set may include *novel emotion labels* that were unseen during training. In contrast, traditional MLL and PLL approaches require the label spaces of the training and test sets to be strictly consistent, making them unsuitable for scenarios where the label space dynamically evolves (i.e., the dimensions of label vectors can be mismatched, impossible to calculate loss or similarity score). *This is why we proposed a novel set-based evaluating metric to measure the performance of emotion recognition.* This fundamental difference highlights our work as not only a novel OVL benchmark for emotion recognition but also a unique testbed for evaluating open-vocabulary learning approaches, which are critical for addressing the challenges of real-world, open-ended tasks.
>
>
> **5. Feasibility.** Previously, OV-MER was difficult to solve due to the near-infinite emotion label space. However, with the development of LLMs, we see the potential to propose a feasible OV-MER framework. The main reason is that LLMs contain an extensive vocabulary, enabling them to output highly diverse emotional labels. Ideally, since LLMs are trained on large-scale data, *the upper limit on the number of emotions is the ability of LLMs, instead of the predefined label spaces*. Specifically, we leverage LLMs in our annotation process and solutions. (1) Our annotation process relies on LLMs to provide reference labels for annotators, thereby improving the comprehensiveness of the annotation results; (2) Compared to traditional methods (such as MLL and PLL), our solution also leverages LLMs to output a richer set of emotional labels. *Meanwhile, it's almost infeasible to apply existing MLL and PLL baselines to OVL tasks, as they are all designed for fixed label space.*
>
> In summary, allowing the model to predict a richer set of emotions (such as OV-MER), rather than limiting the label space to a few basic emotions (such as PLL and MLL), offers a better solution to the complexity of human emotional expression. We believe this work is the future of MER. It will advance emotion recognition from basic emotions to more subtle emotional expressions, contributing to the development of emotional AI.

---

> > ### Author Response · Authors · 2024-11-28
> > **Response to AC (Part2)**
> >
> > **Q2:**
> > While the label space is more relaxed in your framework, it remains finite. This raises the question of whether the term “open” is accurate. Could you clarify your perspective on this?
> >
> > **A2:**
> > Thanks for your comments. We would like to clarify that our finite label space is due to the *finite dataset size* and *language upper limit*, as we cannot construct a dataset with infinite samples, and the emotion words in languages are limited (e.g., 34,000 different emotions according to [4]). As we discussed in **A1**, our OVL *theoretically* can handle infinite label space (e.g., in practice, we easily extend the emotion labels to 248, much more than all the existing datasets). If we increase the dataset size in the future, the label space can be further expanded.
> >
> > In OVL, the term "open" is typically understood from the perspective of *prediction mode*, i.e., whether the model can predict labels outside the predefined label space. Specifically, MLL and PLL cannot predict labels outside the predefined label space. In contrast, OV-MER can predict labels outside the predefined label space. Therefore, we introduce the concept from OVL and use the term "open" in this paper.
> >
> >
> > **Q3:**
> > Additionally, the claim that this paradigm “broadens” the “range of emotion labels” seems debatable. From my understanding, this approach primarily enables capturing simultaneous emotions rather than introducing a fundamentally broader range or type of emotions. Could you elaborate on this point?
> >
> > **A3:**
> > Thanks for your comments. We would like to clarify that, unlike traditional MER which focuses on basic emotion categories, OV-MER *broadens the range of emotion labels*, supporting the prediction of emotions from any category. To ensure this, we rely on LLMs in our annotation process. The main reason is that LLMs contain an extensive vocabulary, enabling them to output highly diverse emotional labels. In our annotation process, we rely on LLMs to provide reference labels for annotators, thereby improving the richness of the annotation results.
> >
> > Indeed we are not introducing a novel language/notion system to describe, instead, we propose a framework that leverages the rich language ability of LLMs to achieve vocabulary-unconstrained emotion recognition, which is the first feasible solution for OV-MER to our knowledge.
> >
> >
> > **Q4:**
> > Finally, are any of the baselines explicitly designed for multi-label learning? Given the existence of established methods in this area, comparisons to such approaches seem essential for a fair and complete evaluation.
> >
> > **A4:**
> > Thanks for your comments. The baseline methods from MLL cannot be directly applied to the OV-MER task. Specifically, MLL/PLL relies on a predefined label space, meaning that the prediction output is a fixed-size $M$-dimensional vector, where $M$ is the label space size. However, in OV-MER, we do not restrict the label space. *Thus, it's not feasible to provide MLL baselines in the OV-MER task and we provide multiple LLM-based baselines for building the benchmark.*
> >
> >
> > [1] Zhang, Min-Ling, and Zhi-Hua Zhou. "A review on multi-label learning algorithms." IEEE transactions on knowledge and data engineering 26, no. 8 (2013): 1819-1837.
> >
> > [2] Cour, Timothee, Ben Sapp, and Ben Taskar. "Learning from partial labels." The Journal of Machine Learning Research 12 (2011): 1501-1536.
> >
> > [3] Wu, Jianzong, Xiangtai Li, Shilin Xu, Haobo Yuan, Henghui Ding, Yibo Yang, Xia Li et al. "Towards open vocabulary learning: A survey." IEEE Transactions on Pattern Analysis and Machine Intelligence (2024).
> >
> > [4] Robert Plutchik. The nature of emotions: Human emotions have deep evolutionary roots, a fact that may explain their complexity and provide tools for clinical practice. American Scientist, 89(4): 344–350, 2001.

---

> > > ### Comment · Area_Chair_ikpq · 2024-11-28
> > >
> > > Thank you for your responses. If you don’t mind, I have some additional questions and thoughts that I would like to share to better understand your work. I would greatly appreciate your perspective.
> > >
> > > 1. Please correct me if I’m wrong, but my understanding is that MLL can indeed handle dynamic and expansive label spaces (e.g., using embedding-based approaches). As a result, I’m a bit unclear about the decision not to position the framework as part of MLL and the lack of direct comparisons to such methods.
> > >
> > > 2. Can you please provide your definition of "open-world" and "open vocabulary"? Do these involve introducing genuinely unseen concepts or just unseen combinations? I find this a bit confusing, especially given that the openness of the space is heavily reliant on LLMs. What proof or evidence is there that the possible outcomes are indeed "open"?
> > >
> > > 3. You stated, "*In OVL, the term 'open' is typically understood from the perspective of prediction mode, i.e., whether the model can predict labels outside the predefined label space.*" Additionally, it was claimed that the paradigm can (or at least aims to) "*recognize labels outside of the predefined label space.*" However, **emotions** are deeply rooted in human psychology and physiology. What assurance is there that unconstrained emotion predictions are indeed valid from a psychological standpoint and meaningful? The reason emotion recognition, and more broadly affective computing, often strictly adheres to predefined emotion labels (or dimensions namely arousal, valence, and dominance) is that these have been extensively studied and validated by experts in psychology.
> > >
> > > Thank you again for taking the time to answer my questions. I want to ensure that everything is clear for both the reviewers and myself before the discussion period ends.
> > >
> > > Best,\
> > > AC

---

> > > > ### Author Response · Authors · 2024-11-30
> > > > **Response to Area Chair ikpq (Part1)**
> > > >
> > > > Thanks for your interest in our work. We are honored to address your comments.
> > > >
> > > > **Q1:**
> > > > Please correct me if I’m wrong, but my understanding is that MLL can indeed handle dynamic and expansive label spaces (e.g., using embedding-based approaches). As a result, I’m a bit unclear about the decision not to position the framework as part of MLL and the lack of direct comparisons to such methods.
> > > >
> > > > **A1:**
> > > > Thanks for your comment. We would like to clarify that **the main difference** between our LLM-based method and MLL for the OVL task is that, MLL can only **passively** handle expansive label spaces, while our approach **actively** outputs and handles novel emotions. Previous methods (e.g., embedding-based MLL baselines) still rely on a *predefined* label space (though it is a different space than the training set) to calculate the similarity distance. In contrast, our approach can flexibly output novel emotion labels without any manual intervention. This fundamental difference makes it challenging to directly apply current methods to OV-MER.
> > > >
> > > > Please consider this specific scenario: we ask models to give emotion labels to a video clip, and we expect the models to give some novel labels (e.g., "rage") that are unseen in the label space. Previous embedding-based methods will not provide the label "rage" as it's not in the label space. Only when evaluators *explicitly* ask those models "Is this a rage emotion?", will the model *passively* calculate the possibility of "rage" via embedding distance. This is the so-called "zero-shot learning" setting in some of the existing works [3,4], but this does not conform to the general definition of OVL. Meanwhile, our LLM-based method has the ability to directly provide novel emotion labels that are not in the label space in the training set. It means our model could directly output the unseen label "rage" without explicit guidance from humans.
> > > >
> > > > Technically, the implementations of MLL and OVL to handle unseen label space are also quite different. Indeed, some MLL baselines can address dynamic and expanding label spaces via manipulating embeddings. These methods typically begin by mapping emotion labels to class name embeddings [2] or class description embeddings [3,4], and then use clustering or linear mapping to give the prediction. As we see, these methods typically rely on fine-tuning or explicit retraining to incorporate new labels into the learned model. This process assumes access to the new labels or requires embedding alignment that still needs to be provided by humans. Thus, instead of positioning our work as part of MLL research, we believe it's more appropriate to claim that those MLL-based methods handling unseen label space primarily fall under the domain of OVL, typically known as "zero-shot learning" (a passive implementation of OVL).
> > > >
> > > > Naturally, the evaluation protocols for OVL and MLL differ significantly due to their inherent conceptual distinctions. Current emotion recognition methods within the MLL paradigm rely on predefined label space, enabling the use of straightforward metrics such as accuracy for evaluation (*the dimension of the prediction vectors is fixed*). In contrast, our approach, operating in an open-vocabulary emotion recognition (OV-MER) setting, does not provide predefined label space and imposes no restrictions on the number of emotion labels per sample (*the dimension of the prediction vectors is not fixed, thus failed to evaluate with existing evaluating metrics*). This necessitates the design of our specialized set-based evaluation metrics tailored to capture the unique challenges of OV-MER, such as handling unseen labels and dynamically expanding label spaces.
> > > >
> > > > Finally, we would like to emphasize that, to our knowledge, this work is the first attempt made at addressing OVL in the emotion recognition task by providing a new benchmark with richer and more nuanced labels. Based on this, we conducted a series of studies around this task. Although some similar concepts (e.g., zero-shot learning in MLL baselines) are introduced, this is the first work that explicitly defines, discusses, and provides a feasible solution to the OVL task. We hope to tackle the emotion recognition problem from a different perspective and bring some new ideas to the community.
> > > >
> > > >
> > > > [1] Liu, Weiwei, Haobo Wang, Xiaobo Shen, and Ivor W. Tsang. "The emerging trends of multi-label learning." IEEE TPAMI 44, no. 11 (2021): 7955-7974.
> > > >
> > > > [2] Xu, Xinzhou, Jun Deng, Nicholas Cummins, Zixing Zhang, Li Zhao, and Björn W. Schuller. "Exploring zero-shot emotion recognition in speech using semantic-embedding prototypes." IEEE TMM 24 (2021): 2752-2765.
> > > >
> > > > [3] Foteinopoulou, Niki Maria, and Ioannis Patras. "Emoclip: A vision-language method for zero-shot video facial expression recognition." In FG, pp. 1-10. IEEE, 2024.
> > > >
> > > > [4] Zhang Sitao, Pan Yimu and Wang James Z, Learning Emotion Representations from Verbal and Nonverbal Communication, CVPR, 2023

---

> > > > > ### Author Response · Authors · 2024-11-30
> > > > > **Response to Area Chair ikpq (Part2)**
> > > > >
> > > > > **Q2:**
> > > > > Can you please provide your definition of "open-world" and "open vocabulary"? Do these involve introducing genuinely unseen concepts or just unseen combinations? I find this a bit confusing, especially given that the openness of the space is heavily reliant on LLMs. What proof or evidence is there that the possible outcomes are indeed "open"?
> > > > >
> > > > > **A2:**
> > > > > In [1], open-world learning refers to models that can identify the classes that are not available during training; in [2], open vocabulary learning refers to models that can recognize categories beyond the annotated label space. Therefore, in the field of machine learning, these concepts are closely related. Based on their definition, they do not involve introducing genuinely unseen concepts beyond language systems. In machine learning, the term "open" generally refers to the model's ability to recognize categories that are not encountered during training.
> > > > >
> > > > > **Proof or evidence of "open".** As we discussed above, our LLM-based method has the capability to *actively* generate diverse emotional expressions. LLMs have an extensive vocabulary that covers almost all existing words, including a large number of emotional words. The "open" mentioned in this paper does not imply that LLMs are expected to generate genuinely unseen emotions beyond current language systems. Rather, it refers to the model's ability to provide emotions that go beyond predefined emotion taxonomies (such as Ekman's Big Six Theory), producing more nuanced and diverse emotional expressions. Here we provide one example to demonstrate the "open" attribute of our method. When predicting the emotion labels of an instance (please see Fig 1(b)), existing methods can only predict labels within basic emotions ("surprise"), while our method can output more nuanced and diverse emotions ("surprise, nervous, dissatisfied").
> > > > >
> > > > > [1] Parmar, Jitendra, Satyendra Chouhan, Vaskar Raychoudhury, and Santosh Rathore. "Open-world machine learning: applications, challenges, and opportunities." ACM Computing Surveys 55, no. 10 (2023): 1-37.
> > > > >
> > > > > [2] Wu, Jianzong, Xiangtai Li, Shilin Xu, Haobo Yuan, Henghui Ding, Yibo Yang, Xia Li et al. "Towards open vocabulary learning: A survey." IEEE Transactions on Pattern Analysis and Machine Intelligence (2024).

---

> > > > > > ### Author Response · Authors · 2024-11-30
> > > > > > **Response to Area Chair ikpq (Part3)**
> > > > > >
> > > > > > **Q3:**
> > > > > > Emotions are deeply rooted in human psychology and physiology. What assurance is there that unconstrained emotion predictions are indeed valid from a psychological standpoint and meaningful? The reason emotion recognition, and more broadly affective computing, often strictly adheres to predefined emotion labels (or dimensions) is that these have been extensively studied and validated by experts in psychology.
> > > > > >
> > > > > > **A3:**
> > > > > > Thanks for your comment.
> > > > > >
> > > > > > First of all, we would like to clarify that, although there are multiple predefined emotion presentation systems (or dimensions) that have been extensively studied and validated by experts in psychology, there is ongoing discussion about whether discrete emotion categories (e.g., Ekman's basic emotions) or dimensional models (e.g., valence-arousal-dominance) are sufficient to fully capture the complexity and cultural variability of human emotions. Consistent efforts have been made in psychology to improve those theories with criticisms [1,2,4]. Thus, we want to emphasize that the hypothesis "strictly adhering to predefined emotion labels (or dimensions)" itself is not scientifically solid, and hard to lead to scientific breakthroughs.
> > > > > >
> > > > > > As a work in the scope of the machine learning field, we try not to touch too many psychological and cognitive terms in our manuscripts. But following the trace of our discussion, here we want to clarify that, our emotion recognition theoretical foundation goes beyond traditional classic emotion theory (Darwin and Ekman's Six Basic Emotion Theory [3,4]) and is more within the scope of appraisal theories [5]. Appraisal theories are one of the most visible cognitive approaches within emotion studies [6,7,8]. In appraisal theories, the emotion processing model is defined as a dynamic architecture for describing the effects of (1) sequential multi-level appraisal checks of a (2) event on the synchronization of the response systems of (3) motivational changes, (4) physiological response patterns, and the (5) central representation of all components (i.e., mental representation of feeling), which occasionally leads to the (6) categorization and possible (7) verbal labeling of the experience. In this paradigm, the verbal labeling step is a subjective, post-processing process where subjects try to use their own language to *mark* the whole emotional experience. Therefore, our OVL provides another alternative using, flexible, richer, and more nuanced labels for verbal labeling instead of a fixed emotion label at one time. **We wish this alternative direction of OVL to address the emotion recognition task can inspire the current emotion AI field.**
> > > > > >
> > > > > > As AC mentioned, emotions are deeply rooted in human psychology and physiology, which is an extremely challenging task. Our goal is to push this task forward from the aspect of the machining learning field. As existing work often restricts the emotion categories within predefined taxonomies (such as Ekman's Theory), we try to conduct the verbal labeling process with more nuanced emotion labels which is also supported by plenty of psychology and physiology studies [1,2,4]. For example, *jealousy* and *hostility* are two distinct emotions, but in basic emotion theory, they are both classified as *anger*, while using the VAD model, it's also challenging to differentiate the two emotions. Given the complexity of human emotions, using basic labels to describe a person's emotional state may not be accurate [1]. Providing richer and more nuanced labels can help more accurately describe an individual's emotional state.
> > > > > >
> > > > > > **To conclude, we provide a feasible solution from the machine learning perspective to address the OV-MER task, which is unique and aims to overcome traditional limitations and pave the way for more accurate and inclusive emotion recognition systems. We wish reviewers could see the potential of this work.**
> > > > > >
> > > > > > [1] Barrett, L. F. (2006). Are emotions natural kinds? Perspectives on Psychological Science, 1(1), 28–58.
> > > > > >
> > > > > > [2] Nuanced Emotions in Psychology: Shaver, P., Schwartz, J., Kirson, D., \& O'Connor, C. (1987). Emotion knowledge: Further exploration of a prototype approach. Journal of Personality and Social Psychology, 52(6),
> > > > > >
> > > > > > [3] Darwin, C. (1872). The Expression of the Emotions in Man and Animals. London: John Murray.
> > > > > >
> > > > > > [4] Ekman, P. (1992). An argument for basic emotions. Cognition \& Emotion, 6(3-4), 169–200.
> > > > > >
> > > > > > [5] Lazarus, R. S. (1991). Emotion and Adaptation. Oxford University Press.
> > > > > >
> > > > > > [6] Jokinen, Jussi PP, and Johanna Silvennoinen. "The appraisal theory of emotion in human–computer interaction." Emotions in technology design: From experience to ethics (2020): 27-39.
> > > > > >
> > > > > > [7] Moors, Agnes, Phoebe C. Ellsworth, Klaus R. Scherer, and Nico H. Frijda. "Appraisal theories of emotion: State of the art and future development." Emotion review 5, no. 2 (2013): 119-124.
> > > > > >
> > > > > > [8] Power, Mick, and Tim Dalgleish. Cognition and emotion: From order to disorder. Psychology press, 2015.

---

> > > > > > > ### Comment · Area_Chair_ikpq · 2024-12-03
> > > > > > >
> > > > > > > Thank you for providing your answers. They have been received.

---

### Author Response · Authors · 2024-12-04
**Summary of Reviewer Feedback and Rebuttal Response**

Dear Area Chair and Reviewers,

We sincerely appreciate AC and all reviewers for their valuable insights, which greatly improved the quality and clarity of our work. We are encouraged by the recognition of our contributions, which will motivate us to work continuously on this task.

In our rebuttal, we addressed the key concerns raised during the review process, including:

1. **Task clarification**: We provided a detailed explanation of how OV-MER differs from existing tasks (MLL and PLL) in terms of task definition, solutions, and evaluations.

2. **Validation of OV labels**: We conducted a *user study*, demonstrating that OV labels provide more information than basic emotions, helping to more accurately describe a person's emotional state.

3. **Clarifications**: We clarified our pre-labeling and manual checking process, added more experimental analysis, and corrected some statements.

Finally, we would like to thank the AC and reviewers for initiating insightful and in-depth discussions during this rebuttal. We believe our intensive and high-quality discourse itself already proves the huge potential of OV-MER and highlights the emerging demand for further development of the OV-MER task.

We acknowledge that OV-MER is in its nascent stages and that some challenges remain. However, we are confident that this emerging perspective offers a fresh lens for approaching emotion recognition tasks and could inspire further innovation in emotion AI. By transitioning from basic to more nuanced emotional representations, OV-MER aligns with contemporary research trends both for the machine learning field and the cognitive science field, setting the stage for the future of emotion AI.

We sincerely wish reviewers consider the potential positive impact this work could make in the community. We hope this work can be visible so that it draws more attention from other researchers, thus joint efforts in the community can be made to push forward the renovation of the current emotion AI field by refining the OVL task. We envision this work as a seed for transformative change, not just within emotion AI but across the broader landscape of human-centric technology, and we humbly invite the community to join us in nurturing its growth into something extraordinary.

Thank you once again for your support and consideration.

Best regards,

The Authors

---

### Meta-Review · Area_Chair_ikpq · 2024-12-19

**Metareview:**

The paper proposes a new paradigm for multimodal emotion recognition by expanding the label space based on LLMs. Additionally, it includes a new database and benchmarks for this new approach. The paper is quite interesting and out-of-the-box. The overall concept introduced in the paper is quite thought-provoking and could result in interesting new directions in affective computing. Additionally, the paper is easy to understand and follow. However, the paper unfortunately lacks in several areas. In particular, the validity of the newly proposed/identified labels, questionable positions given existing areas such as multi-label learning, lack of long-term stability given the rapidly varying state of LLMs, and a few others.

**Additional Comments On Reviewer Discussion:**

After the rebuttal, the paper received scores of 3, 5, 5, 6, 8, as the reviewers had a range of opinions about the work. While everyone expressed interest in the paper, the reviewers and AC acknowledged during the discussions that its critical weaknesses highlighted above, would need to be addressed before the paper can be published. Reviewer XYyA, who had initially given the paper its highest score of 8, did not participate in discussions during or after the rebuttal, leaving their alignment with the other reviewers and AC unclear.

---

### Decision · Program_Chairs · 2025-01-22

Reject